# Rethinking the Benefits of Steerable Features in 3D Equivariant Graph Neural Networks

**Shih-Hsin Wang**[1], **Yung-Chang Hsu**[2], **Justin Baker**[1],
**Andrea Bertozzi**[3], **Jack Xin**[4] **& Bao Wang**[1]*
[1]Department of Mathematics and Scientific Computing and Imaging (SCI) Institute
University of Utah, Salt Lake City, UT 84102, USA
[2]Department of Mathematics, Purdue University, West Lafayette, IN 47907, USA
[3]Department of Mathematics, UCLA, Los Angeles, CA 90095, USA
[4]Department of Mathematics, UC Irvine, Irvine, CA 92697, USA

## Abstract

Theoretical and empirical comparisons have been made to assess the expressive power and performance of invariant and equivariant GNNs. However, there is currently no theoretical result comparing the expressive power of $k$-hop invariant GNNs and equivariant GNNs. Additionally, little is understood about whether the performance of equivariant GNNs, employing steerable features up to type-$L$, increases as $L$ grows – especially when the feature dimension is held constant. In this study, we introduce a key lemma that allows us to analyze steerable features by examining their corresponding invariant features. The lemma facilitates us in understanding the limitations of $k$-hop invariant GNNs, which fail to capture the global geometric structure due to the loss of geometric information between local structures. Furthermore, we analyze the ability of steerable features to carry information by studying their corresponding invariant features. In particular, we establish that when the input spatial embedding has full rank, the information-carrying ability of steerable features is characterized by their dimension and remains independent of the feature types. This suggests that when the feature dimension is constant, increasing $L$ does not lead to essentially improved performance in equivariant GNNs employing steerable features up to type-$L$. We substantiate our theoretical insights with numerical evidence.

## 1 Introduction

Machine learning (ML) tasks have different kinds of inherent Euclidean symmetries. For instance, translating or rotating an image does not change its label (invariance) [21] while applying Euclidean isometries to a molecule results in corresponding changes in its dynamics (equivariance) [3; 5]. Enforcing Euclidean symmetry in neural network design [35; 7] significantly improves sample efficiency [16; 8], generalizability [31], and robustness [13], and preserves the principles of physics [12]. Utilizing steerable features provides a powerful framework for crafting symmetry-aware neural networks. In particular, it enables the design of neural networks with equivariance to 3D Euclidean transformations by leveraging the group representations of $O(3)$ and $SO(3)$. The reducibility of the group representations of $O(3)$ or $SO(3)$ [30; 42] allows us to decompose any steerable vector into a direct sum of steerable vectors of different types. These types of steerable vectors are defined by Wigner-D matrices [42]; in particular, type-0 and type-1 steerable vectors corresponding to scalars and 3D vectors, resp. Research in designing equivariant models, especially graph neural networks (GNNs) that update steerable features up to type $L$, has surged. For instance, SchNet [34], DimeNet [15], SphereNet [25], and ComENet [40] are invariant GNNs that propagate invariant features ($L = 0$). Equivariant GNNs, e.g., as proposed in [32; 33], update steerable features up to $L = 1$. Meanwhile, numerous architectures are designed to handle steerable features for $L > 1$; see e.g. [8; 45; 37; 42; 13; 18; 43; 22; 4; 5].

### 1.1 Remarks on some remarkable studies

Recent research, especially [20], highlights performance disparities between 1-hop message-passing invariant GNNs and equivariant GNNs. [20] introduces the geometric Weisfeiler-Lehman (GWL) test and its invariant version (IGWL) to characterize the expressive power of 1-hop equivariant and

---

*Correspond to `wangbaonj@gmail.com`

invariant GNNs, resp., and then shows that 1-hop invariant GNNs may underperform equivariant GNNs – primarily due to their limited ability to capture the global geometry of the graph. Intriguingly, [20] also reveals that 1-hop invariant GNNs can be as expressive as equivariant GNNs for fully connected input graphs. However, the theory in [20] is limited to 1-hop message passing, **leaving questions about the implications of introducing multi-hop aggregation into the message-passing of invariant GNNs.** E.g., ComENet [40] and SphereNet [25] introduce the concept of *completeness* of edge attributes – a property held by invariant edge features that enable them to uniquely determine the spatial embedding of a point cloud up to the group action. Specifically, ComENet achieves this completeness by computing dihedral angles between local structures, gathering information from up to 2 hops neighbors. Both ComENet and SphereNet can outperform equivariant GNNs in molecular modeling tasks, even when dealing with non-fully connected input graphs.

The effects of steerable features of different types have also been explored. E.g. [3; 23; 2; 28] show that the performance of equivariant GNNs – that use steerable features up to type-$L$ – can be enhanced by increasing $L$. However, these experiments typically lack the control of feature dimensions [12]. **This requires rethinking the benefits of increasing $L$ while maintaining the feature dimension.** Indeed, [4] first compares the performance of SEGNN – a particular equivariant GNN using different $L$, reporting limited improvement beyond $L = 1$ when feature dimensions are held constant.

## 1.2 Our contributions

We aim to address the theoretical questions in Section 1.1 and build new understandings of the expressive power and message-passing mechanisms of invariant and equivariant GNNs. We summarize our major theoretical contribution as follows:

- We establish Lemma 1, a key lemma showing that **any steerable feature corresponds to some invariant features**. This lemma serves as a cornerstone for our theoretical analysis, enabling us to examine steerable features by studying their corresponding invariant counterparts.
- We demonstrate that message-passing using steerable features can be interpreted as message-passing with their corresponding invariant features. This perspective reveals that both equivariant and invariant GNNs propagate invariant features across different local neighborhoods. However, **invariant GNNs, when compared to equivariant GNNs, lack the intrinsic capability to capture geometric information between local neighborhoods, even when adopting the $k$-hop message-passing.** In particular, $k$-hop invariant GNNs may struggle to capture the changing geometry between $k$-hop local structures and fail to obtain accurate global invariant features.
- We analyze the ability of steerable features to carry information by studying their corresponding invariant features. In particular, we establish that when the input spatial embedding has full rank, the information-carrying ability of steerable features is characterized by their dimension and remains independent of the feature types utilized in steerable features. **This result indicates that when preserving the feature dimension, the performance of equivariant GNNs employing steerable features up to type-$L$ may not increase as $L$ grows.**
- We provide numerical evidence that consistently echos the above theoretical insights.

## 2 Background

We recap group theory, steerable vector spaces as introduced in [4], the concepts of geometric graphs and geometric Graph Neural Networks (GNNs). We also review the Geometric Weisfeiler-Lehman test (GWL) and its relevant results from [20] in Appendix D.

**Group theory.** Let $\mathfrak{G}$ be a group, and consider a space $X$ on which $\mathfrak{G}$ acts. When an element $x \in X$ undergoes the group action of $\mathfrak{G}$, denoted as $\mathfrak{g} \cdot x$, we define the $\mathfrak{G}$-*orbit of $x$* as the set $\mathfrak{G} \cdot x := \{\mathfrak{g} \cdot x \in X \mid \mathfrak{g} \in \mathfrak{G}\} \subseteq X$. The *quotient space* $X/\mathfrak{G}$ contains all $\mathfrak{G}$-orbits. Additionally, we define the *stabilizer of $x$* as the subgroup $\mathfrak{G}_x := \{\mathfrak{g} \in \mathfrak{G} \mid \mathfrak{g} \cdot x = x\} \subseteq \mathfrak{G}$.

Assuming that $\mathfrak{G}$ acts on both spaces $X$ and $Y$, we refer to a function $f : X \to Y$ as $\mathfrak{G}$-*equivariant* if it satisfies $f(\mathfrak{g} \cdot x) = \mathfrak{g} \cdot f(x)$, and as $\mathfrak{G}$-*invariant* if it satisfies $f(\mathfrak{g} \cdot x) = f(x)$.

**Steerable vector space.** We revisit the concept of a steerable vector space following [4], where the group action is induced by a group representation. A vector space $V$ over $\mathbb{R}$ is said to be $\mathfrak{G}$-*steerable* for $\mathfrak{G}$ if a group representation $\rho : \mathfrak{G} \to \mathrm{GL}(V)$[1] of $\mathfrak{G}$ is assigned to $V$. In other words, for any vector $\boldsymbol{v} \in V$, the transformation $\mathfrak{g} \cdot \boldsymbol{v}$ is given by the matrix multiplication $\mathfrak{g} \cdot \boldsymbol{v} = \rho(\mathfrak{g})\boldsymbol{v}$.

When $\mathfrak{G}$ is $\mathrm{SO}(3)$ or $\mathrm{O}(3)$, we can decompose the representation into irreducible representations. Consequently, it is sufficient to investigate steerable vector spaces transformed by these irreducible

---

[1]$\mathrm{GL}(V)$ denotes the general linear group of the vector space $V$.

representations. It is known that the irreducible representations of SO(3) have dimensions $2l + 1$ for $l = \{0\} \cup \mathbb{N}$, and they are defined by the Wigner-D matrices $\{\boldsymbol{D}^l\}$. We refer to the vector space transformed by $\boldsymbol{D}^l$ of order $l$ as a *type-l* SO(3)-*steerable vector space*, denoted as $V_l$.

Moreover, notice that O(3) is the direct product of SO(3) and the inversion group $\mathcal{I} \coloneqq \{\boldsymbol{I}, -\boldsymbol{I}\}$, implying that any representation of O(3) can be written as the product of a representation of SO(3) and a representation of $\mathcal{I}$. The inversion group $\mathcal{I}$ has only two irreducible representations: the trivial representation $\rho_t(\boldsymbol{I}) = \rho_t(-\boldsymbol{I}) = 1$ and the sign representation $\rho_s(\boldsymbol{I}) = 1, \rho_s(-\boldsymbol{I}) = -1$. As a result, all the irreducible representation of O(3) can be expressed as $\{\rho(i) \cdot \boldsymbol{D}^l(\mathfrak{g}) | l \geq 0, \rho = \rho_t \text{ or } \rho_s\}$ by writing the element of O(3) as a product $i \cdot \mathfrak{g}$ where $i \in \mathcal{I}, \mathfrak{g} \in \text{SO}(3)$. An alternative way to represent these irreducible representations involves the determinant, as follows:

$$\boldsymbol{D}^l_{\text{ind}}(\mathfrak{g}) \coloneqq \det(\mathfrak{g})^l \cdot \boldsymbol{D}^l(\det(\mathfrak{g})\mathfrak{g}) \text{ and } \boldsymbol{D}^l_{\text{aug}}(\mathfrak{g}) \coloneqq \det(\mathfrak{g})^{l+1} \cdot \boldsymbol{D}^l(\det(\mathfrak{g})\mathfrak{g}),$$

where $g \in \text{O}(3)$ and $l \geq 0$. To avoid ambiguity, we use the terminology of $\text{ind}$ and $\text{aug}$ for clarification and ease of subsequent study. Lastly, let $V_{l,\text{ind}}$ denote the O(3)-steerable vector space acted upon by $\boldsymbol{D}^l_{\text{ind}}$ and $V_{l,\text{aug}}$ denote the O(3)-steerable vector space acted upon by $\boldsymbol{D}^l_{\text{aug}}$. We may denote $V_{0,\text{ind}}$ simply as $V_0$ since both correspond to trivial representations.

**Geometric graphs.** Let $(\mathcal{V}, \mathcal{E})$ be an attributed graph comprising $m$ nodes, where each node $i \in \mathcal{V}$ has a feature embedding $\boldsymbol{f}_i \in \mathbb{R}^n$ and a spatial embedding $\boldsymbol{x}_i \in \mathbb{R}^3$. The input embeddings can be organized as two matrices $\boldsymbol{F} = [\boldsymbol{f}_1, \ldots, \boldsymbol{f}_m] \in \mathbb{R}^{n \times m}$ and $\boldsymbol{X} = [\boldsymbol{x}_1, \ldots, \boldsymbol{x}_m] \in \mathbb{R}^{3 \times m}$. We can represent this as $\mathcal{G} = (\mathcal{V}, \mathcal{E}, \boldsymbol{F}, \boldsymbol{X})$, referring to this attributed graph as a *geometric graph*. Consider two geometric graphs $\mathcal{G}_1 = (\mathcal{V}_1, \mathcal{E}_1, \boldsymbol{F}^{\mathcal{G}_1}, \boldsymbol{X}^{\mathcal{G}_1}), \mathcal{G}_2 = (\mathcal{V}_2, \mathcal{E}_2, \boldsymbol{F}^{\mathcal{G}_2}, \boldsymbol{X}^{\mathcal{G}_2})$ where the underlying graph structures and feature embeddings are isomorphic. In other words, there is an edge-preserving bijection $b : \mathcal{V}_1 \to \mathcal{V}_2$ s.t. $\boldsymbol{f}_i^{\mathcal{G}_1} = \boldsymbol{f}_{b(i)}^{\mathcal{G}_2}$, where we do not assume the uniqueness of $b$. We say $\mathcal{G}_1$ and $\mathcal{G}_2$ are *identical* up to group action if there is a graph isomorphism $b$ such that $\boldsymbol{x}_i^{\mathcal{G}_1} = \mathfrak{g} \cdot \boldsymbol{x}_{b(i)}^{\mathcal{G}_2}$ for some $\mathfrak{g} \in \mathfrak{G}$. Let $\mathcal{N}_i^{(k)}$ represent the $k$-hop neighborhood of node $i$, the set of nodes in $\mathcal{V}$ that are reachable from $i$ through a path with $k$ edges or fewer. Then we say $\mathcal{G}_1$ and $\mathcal{G}_2$ are *k-hop identical* if there is a graph isomorphism $b$ such that for any node $i \in \mathcal{V}_1$, there exists $\mathfrak{g}_i \in \mathfrak{G}$ satisfying $\boldsymbol{x}_j^{\mathcal{G}_1} = \mathfrak{g}_i \cdot \boldsymbol{x}_{b(j)}^{\mathcal{G}_2}$ for any $j \in \mathcal{N}_i^{(k)} \cup \{i\}$. Otherwise, we say $\mathcal{G}_1$ and $\mathcal{G}_2$ are *k-hop distinct* if, for all isomorphisms $b$, there is a node $i \in \mathcal{V}_1$ such that for any $\mathfrak{g} \in \mathfrak{G}$, we have $\boldsymbol{x}_j^{\mathcal{G}_1} \neq \mathfrak{g} \cdot \boldsymbol{x}_{b(j)}^{\mathcal{G}_2}$ for some $j \in \mathcal{N}_i^{(k)} \cup \{i\}$.

**$k$-hop geometric GNNs.** We extend the framework of geometric GNNs in [20] to a $k$-hop setting. This framework can be regarded as an abstract of several existing invariant and equivariant GNNs, e.g. [34; 32; 4; 2]; see Appendix E for details. Consider a geometric graph $\mathcal{G} = (\mathcal{V}, \mathcal{E}, \boldsymbol{F}, \boldsymbol{X})$ with $\mathfrak{G}$ being SO(3) or O(3). Geometric GNNs propagate features from iteration $t$ to $t + 1$ as follows:

$$\boldsymbol{f}_i^{(t+1)} = \text{UPD}\left(\boldsymbol{f}_i^{(t)}, \text{AGG}(\{\!\!\{\boldsymbol{f}_i^{(t)}, \boldsymbol{f}_j^{(t)}, \boldsymbol{x}_{ij} \mid j \in \mathcal{N}_i^{(k)}\}\!\!\})\right), \text{ with } \boldsymbol{f}_i^{(0)} = \boldsymbol{f}_i \quad (1)$$

where $\boldsymbol{x}_{ij} \coloneqq \boldsymbol{x}_i - \boldsymbol{x}_j$, $\{\!\!\{\cdot\}\!\!\}$ denotes a multiset, and UPD and AGG are learnable update and aggregate ($\mathfrak{G}$-equivariant) functions. Notably, the equivariance of UPD and AGG ensures that this message-passing mechanism remains equivariant, that is, we have

$$\rho^{(t+1)}(\mathfrak{g})\boldsymbol{f}_i^{(t+1)} = \text{UPD}\left(\rho^{(t)}(\mathfrak{g})\boldsymbol{f}_i^{(t)}, \text{AGG}(\{\!\!\{\rho^{(t)}(\mathfrak{g})\boldsymbol{f}_i^{(t)}, \rho^{(t)}(\mathfrak{g})\boldsymbol{f}_j^{(t)}, \mathfrak{g} \cdot \boldsymbol{x}_{ij} \mid j \in \mathcal{N}_i^{(k)}\}\!\!\})\right). \quad (2)$$

where $\rho^{(t+1)}(\mathfrak{g})$ and $\rho^{(t)}(\mathfrak{g})$ are group representations acting on $\boldsymbol{f}_i^{(t+1)}$ and $\boldsymbol{f}_i^{(t)}$, resp., and $\mathfrak{g} \cdot \boldsymbol{x}_{ij}$ is the regular representation. The features $\boldsymbol{f}_i^{(t)}$ in geometric GNNs consist of steerable features up to type-$L$, where $L = 0(> 0)$ corresponds to $\mathfrak{G}$-invariant/$\mathfrak{G}$-equivariant GNNs.

## 3 MAIN THEORY

**Steerable features and invariant features.** From now on, let $\mathfrak{G}$ be a group acting on $\mathbb{R}^{3 \times m}$. We define two key concepts: *steerable features* refer to the steerable vectors that are generated equivariantly from the input spatial embedding $\boldsymbol{X}$, meaning that they can be expressed as $f(\boldsymbol{X})$ for some $\mathfrak{G}$-equivariant function $f$ mapping from $\mathbb{R}^{3 \times m}$ to a steerable vector space. Similarly, *invariant features* are characterized as the steerable vectors $f(\boldsymbol{X})$ produced by some $\mathfrak{G}$-invariant function $f$. Notice that these two concepts cover all the notions of existing terms, such as steerable feature fields [42; 41] and steerable feature vectors [37]; see Section 4 for more discussion.

**Steerable features and their corresponding invariant features.** By the axiom of choice [17], we can choose a "representative" for each orbit in $\mathbb{R}^{3 \times m}/\mathfrak{G}$. Specifically, there is a (set-theoretical) function $c : \mathbb{R}^{3 \times m}/\mathfrak{G} \to \mathbb{R}^{3 \times m}$ that maps each orbit to an element within that orbit.

Note that for any $\boldsymbol{X}, \boldsymbol{X}' \in \mathbb{R}^{3 \times m}$, they belong to the same $\mathfrak{G}$-orbit if and only if $\boldsymbol{X} = \mathfrak{g} \cdot \boldsymbol{X}'$ for some $\mathfrak{g} \in \mathfrak{G}$. Consequently, for any given $\boldsymbol{X}$, the set $\{\mathfrak{g} \in \mathfrak{G} \mid \mathfrak{g} \cdot c(\mathfrak{G} \cdot \boldsymbol{X}) = \boldsymbol{X}\}$ is not empty since $c(\mathfrak{G} \cdot \boldsymbol{X})$ and $\boldsymbol{X}$ lie in the same orbit $\mathfrak{G} \cdot \boldsymbol{X}$. Once again, applying the axiom of choice, we can select one element from this set, denoting it as $\mathfrak{g}_{\boldsymbol{X}}$. Then we can represent any $\boldsymbol{X} \in \mathbb{R}^{3 \times m}$ using the pair $\boldsymbol{X} = \big(c(\mathfrak{G} \cdot \boldsymbol{X}), \mathfrak{g}_{\boldsymbol{X}}\big)$, where $\mathfrak{g}_{\boldsymbol{X}} \cdot c(\mathfrak{G} \cdot \boldsymbol{X}) = \boldsymbol{X}$. Next, we introduce a key lemma that serves as the cornerstone of our theoretical framework; a similar finding can be found in [44].

**Lemma 1.** *Let $V$ be a $d$-dimensional $\mathfrak{G}$-steerable vector space with the assigned group representation $\rho : \mathfrak{G} \to \mathrm{GL}(V)$. If $f : \mathbb{R}^{3 \times m} \to V$ is $\mathfrak{G}$-equivariant, then there exists a unique $\mathfrak{G}$-invariant function $\lambda : \mathbb{R}^{3 \times m} \to V_0^{\oplus d}$ s.t. $f(\boldsymbol{X}) = \rho(\mathfrak{g}_{\boldsymbol{X}})\lambda(\boldsymbol{X})$, where $V_0$ denotes the 1D trivial representation of $\mathfrak{G}^2$. In particular, the following map is well-defined*

$$\{f : \mathbb{R}^{3 \times m} \to V \mid f : \mathfrak{G}\text{-equivariant}\} \to \{\lambda : \mathbb{R}^{3 \times m} \to V_0^{\oplus d} \mid \lambda : \mathfrak{G}\text{-invariant}\}. \tag{3}$$

**Remark 1.** *While we consider the group action on $\mathbb{R}^{3 \times m}$, it is important to note that the same result applies to group actions on any space. One can interpret that $\mathfrak{g}_{\boldsymbol{X}}$ absorbs the group action on $\boldsymbol{X}$ and constrains $\lambda(\boldsymbol{X})$ to be invariant. More precisely, we see that $\rho(\mathfrak{h})\rho(\mathfrak{g}_{\boldsymbol{X}})\lambda(\boldsymbol{X}) = \rho(\mathfrak{h})f(\boldsymbol{X}) = f(\mathfrak{h} \cdot \boldsymbol{X}) = \rho(\mathfrak{g}_{\mathfrak{h} \cdot \boldsymbol{X}})\lambda(\boldsymbol{X})$ for any $\mathfrak{h} \in \mathfrak{G}$. However, it is not necessary that $\mathfrak{h} \cdot \mathfrak{g}_{\boldsymbol{X}} = \mathfrak{g}_{\mathfrak{h} \cdot \boldsymbol{X}}$.*

We observe from Lemma 1 that any steerable feature with dimension $d$, denoted as $f(\boldsymbol{X})$, can be substituted with a group element $\mathfrak{g}_{\boldsymbol{X}}$ and $d$-dimensional invariant features $\lambda(\boldsymbol{X})$. However, since the selection of $\mathfrak{g}_{\boldsymbol{X}}$ remains unaffected by the function $f$, what we truly observe is that **the steerable feature $f(\boldsymbol{X})$ corresponds to a unique $d$-dimensional invariant feature $\lambda(\boldsymbol{X})$.** The correspondence of $f(\boldsymbol{X})$ to $\lambda(\boldsymbol{X})$ shares similarities with the notion of scalarization in [9] and [10]. A detailed discussion of their distinctions is provided in Appendix A.

### 3.1 Message-passing mechanisms of geometric GNNs

It has been demonstrated that 1-hop invariant GNNs underperform equivariant GNNs [20]. However, the underlying mechanisms of message passing that distinguish these two approaches remain unclear. In this section, we leverage our framework to shed some light on this issue. In particular, we will show that *$k$-hop invariant GNNs do not possess an inherent capability to capture geometric information between local structures, whereas equivariant GNNs do.*

**Steerable features propagate like invariant features.** We first treat the steerable features $\boldsymbol{f}_i^{(t)}$, in the propagation scheme defined in equation (1), as functions of $\boldsymbol{X}$ and denote them as $\boldsymbol{f}_i^{(t)} = \boldsymbol{f}_i^{(t)}(\boldsymbol{X})$. By applying Lemma 1, we can express $\boldsymbol{f}_i^{(t)}(\boldsymbol{X}) = \rho^{(t)}(\mathfrak{g})\lambda_i^{(t)}(\boldsymbol{X})$, where $\lambda_i^{(t)}$ is $\mathfrak{G}$-invariant, and for simplicity, we represent $\mathfrak{g}_{\boldsymbol{X}}$ as $\mathfrak{g}$. Additionally, since we have $\boldsymbol{X} = \mathfrak{g} \cdot c(\mathfrak{G} \cdot \boldsymbol{X})$ where $c(\mathfrak{G} \cdot \boldsymbol{X})$ is $\mathfrak{G}$-invariant, and $\mathfrak{g}^{-1}\boldsymbol{x}_{ij}$ are the invariant features corresponding to $\boldsymbol{x}_{ij} := \boldsymbol{x}_i - \boldsymbol{x}_j$ due to the uniqueness described in Lemma 1. Consequently, we arrive at the following relationship:

$$\mathrm{UPD}\left(\boldsymbol{f}_i^{(t)}(\boldsymbol{X}), \mathrm{AGG}(\{\!\!\{\boldsymbol{f}_i^{(t)}(\boldsymbol{X}), \boldsymbol{f}_j^{(t)}(\boldsymbol{X}), \boldsymbol{x}_{ij} \mid j \in \mathcal{N}_i^{(k)}\}\!\!\})\right)$$

$$= \mathrm{UPD}\left(\rho^{(t)}(\mathfrak{g})\lambda_i^{(t)}(\boldsymbol{X}), \mathrm{AGG}(\{\!\!\{\rho^{(t)}(\mathfrak{g})\lambda_i^{(t)}(\boldsymbol{X}), \rho^{(t)}(\mathfrak{g})\lambda_j^{(t)}(\boldsymbol{X}), \mathfrak{g} \cdot (\mathfrak{g}^{-1}\boldsymbol{x}_{ij}) \mid j \in \mathcal{N}_i^{(k)}\}\!\!\})\right)$$

$$= \rho^{(t+1)}(\mathfrak{g})\, \mathrm{UPD}\left(\lambda_i^{(t)}(\boldsymbol{X}), \mathrm{AGG}(\{\!\!\{\lambda_i^{(t)}(\boldsymbol{X}), \lambda_j^{(t)}(\boldsymbol{X}), \mathfrak{g}^{-1}\boldsymbol{x}_{ij} \mid j \in \mathcal{N}_i^{(k)}\}\!\!\})\right).$$

Since $\boldsymbol{f}_i^{(t+1)}(\boldsymbol{X}) = \rho^{(t+1)}(\mathfrak{g})\lambda_i^{(t+1)}(\boldsymbol{X})$, the uniqueness of corresponding invariant features implies

$$\lambda_i^{(t+1)}(\boldsymbol{X}) = \mathrm{UPD}\left(\lambda_i^{(t)}(\boldsymbol{X}), \mathrm{AGG}(\{\!\!\{\lambda_i^{(t)}(\boldsymbol{X}), \lambda_j^{(t)}(\boldsymbol{X}), \mathfrak{g}^{-1}\boldsymbol{x}_{ij} \mid j \in \mathcal{N}_i^{(k)}\}\!\!\})\right). \tag{4}$$

This reveals that the propagation of steerable features can be effectively understood as the propagation of their corresponding invariant features. Therefore, **we can analyze the message-passing mechanism by examining how the corresponding invariant features are aggregated and updated.**

**Message aggregated from multi-hop neighborhoods.** To investigate the aggregation of local features from multi-hop neighborhoods, we explicitly specify the input spatial embeddings for each steerable feature $\boldsymbol{f}_i^{(t)}$: namely, for any iteration $t$ and node $i$, we write $\boldsymbol{f}_i^{(t)} = \boldsymbol{f}_i^{(t)}(\boldsymbol{X}_i^{(t)})$ where $\boldsymbol{X}_i^{(t)} := [\boldsymbol{x}_j]_{j \in \mathcal{N}_i^{(tk)} \cup \{i\}}$ [3] represents the spatial embedding, consisting of all coordinates of node $i$ and its $tk$-hop neighbors $\mathcal{N}_i^{(tk)}$. In particular, the $tk$-hop neighbor $\mathcal{N}_i^{(tk)}$ includes all the nodes that can propagate information to node $i$ through $k$-hop aggregation $t$ times. Without loss of generality, we may assume the group representations are all the same for any iteration, i.e. $\rho = \rho^{(t)}$ for any $t$. Next,

---

[2]For $\mathfrak{G} = \mathrm{SO}(3)$, it corresponds to the type-0 steerable vector space we defined in Section 2. For simplicity, we employ the same notation here.

[3]We do not make any assumptions about the order of the coordinates or indices here.

we utilize Lemma 1 to examine how corresponding invariant features propagate through different multi-hop neighborhoods. Specifically, we express these steerable features as $\rho(\mathfrak{g}_i^{(t)})\lambda_i^{(t)}$, where $\lambda_i^{(t)} = \lambda_i^{(t)}(\boldsymbol{X}_i^{(t)})$ corresponds to the local invariant features, and $\mathfrak{g}_i^{(t)}$ denotes $\mathfrak{g}_{\boldsymbol{X}_i^{(t)}}^{(t)}$ for simplicity.

Our goal is to investigate how invariant features $\lambda_i^{(t)}$ at iteration $t$ are aggregated and updated into invariant features $\lambda_i^{(t+1)}$ at iteration $t + 1$. In particular, we have

$$
\begin{aligned}
\boldsymbol{f}_i^{(t+1)} &= \mathrm{UPD}\left(\boldsymbol{f}_i^{(t)}, \mathrm{AGG}(\{\!\!\{\boldsymbol{f}_i^{(t)}, \boldsymbol{f}_j^{(t)}, \boldsymbol{x}_{ij} \mid j \in \mathcal{N}_i^{(k)}\}\!\!\})\right) \\
&= \mathrm{UPD}\left(\rho(\mathfrak{g}_i^{(t)})\lambda_i^{(t)}, \mathrm{AGG}(\{\!\!\{\rho(\mathfrak{g}_i^{(t)})\lambda_i^{(t)}, \rho(\mathfrak{g}_j^{(t)})\lambda_j^{(t)}, \mathfrak{g}_i^{(1)} \cdot ((\mathfrak{g}_i^{(1)})^{-1}\boldsymbol{x}_{ij}) \mid j \in \mathcal{N}_i^{(k)}\}\!\!\})\right),
\end{aligned} \tag{5}
$$

Next, by leveraging the fact that $\boldsymbol{f}_i^{(t+1)} = \rho(\mathfrak{g}_i^{(t+1)})\lambda_i^{(t+1)}$ and the uniqueness of $\lambda_i^{(t+1)}$ as established in Lemma 1, we deduce the propagation of the corresponding invariant features.

$$
\begin{aligned}
\lambda_i^{(t+1)} = \mathrm{UPD}\Big(&\rho((\mathfrak{g}_i^{(t+1)})^{-1}\mathfrak{g}_i^{(t)})\lambda_i^{(t)}, \mathrm{AGG}\Big(\{\!\!\{\rho((\mathfrak{g}_i^{(t+1)})^{-1}\mathfrak{g}_i^{(t)})\lambda_i^{(t)}, \\
&\rho((\mathfrak{g}_i^{(t+1)})^{-1}\mathfrak{g}_j^{(t)})\lambda_j^{(t)}, ((\mathfrak{g}_i^{(t+1)})^{-1}\mathfrak{g}_i^{(1)})(\mathfrak{g}_i^{(1)})^{-1}\boldsymbol{x}_{ij} \mid j \in \mathcal{N}_i^{(k)}\}\!\!\}\Big)\Big).
\end{aligned} \tag{6}
$$

We assert that the collection of elements $\left\{\rho\big((\mathfrak{g}_i^{(t+1)})^{-1}\mathfrak{g}_j^{(t)}\big) \mid j \in \mathcal{N}_i^{(k)} \cup \{i\}\right\}$ plays a crucial role in enabling geometric GNNs to capture geometric information between the local structures $\mathcal{N}_j^{(tk)}$. To clarify this, we consider $k$-hop invariant GNNs where $\rho$ is trivial, resulting in the set of elements $\left\{\rho\big((\mathfrak{g}_i^{(t+1)})^{-1}\mathfrak{g}_j^{(t)}\big) \mid j \in \mathcal{N}_i^{(k)} \cup \{i\}\right\}$ being inevitably overlooked. Then we have the following:

**Theorem 1.** *If $\mathcal{G}_1$ and $\mathcal{G}_2$ are two $k$-hop identical graphs, then any iteration of $k$-hop invariant GNNs will get the same output from these two graphs. That is, there is a graph isomorphism $b$ such that $\lambda_i^{(t+1)} = \lambda_{b(i)}^{(t+1)}$ for any $i$, even though $\mathcal{G}_1$ and $\mathcal{G}_2$ may not be identical up to group action.*

**Remark 2.** *We also extend the IGWL test in [20] to a $k$-hop setting and show that (1-hop) GWL is still more powerful than $k$-hop IGWL; see Appendix D for details.*

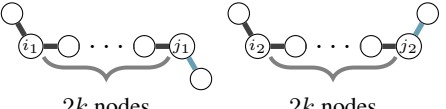

Theorem 1 implies that $k$-hop invariant GNNs may struggle to capture the changing geometry between $k$-hop local structures. To illustrate this point, let's consider the examples of $k$-chains discussed in [20]. For any $k$, let's examine a pair of graphs, $\mathcal{G}_1$ and $\mathcal{G}_2$, each consisting of $2k + 2$ nodes. In these graphs, there are $2k$ nodes arranged in a line, with differentiation based on the orientation of the two endpoints, as demonstrated in Fig. 1. By assigning the same attributes and customizing the

Figure 1: A pair of graphs each consisting of $2k + 2$ nodes. These graphs are nearly identical, differing only in the orientation of a single edge, marked in blue. Despite this minor distinction, these graphs remain $k$-hop identical.

spatial embeddings, we can make these graphs $k$-hop identical but $(k + 1)$-hop distinct. Specifically, the geometry between $\mathcal{N}_{i_1}^{(k)}$ and $\mathcal{N}_{j_1}^{(k)}$ and the geometry between $\mathcal{N}_{i_2}^{(k)}$ and $\mathcal{N}_{j_2}^{(k)}$ differ, as the unions $\mathcal{N}_{i_1}^{(k)} \cup \mathcal{N}_{j_1}^{(k)} = \mathcal{G}_1$ and $\mathcal{N}_{i_2}^{(k)} \cup \mathcal{N}_{j_2}^{(k)} = \mathcal{G}_2$ are not identical. However, since $\mathcal{G}_1$ and $\mathcal{G}_2$ are $k$-hop identical, we expect that $k$-hop invariant GNNs will likely struggle to distinguish between the distinct geometries present in these scenarios. We further analyze this example empirically in Section 5.

Let $V$ and $W$ be steerable vector spaces with the assigned faithful group representations $\rho_V$ and $\rho_W$, i.e., the group homomorphisms $\rho_V : \mathfrak{G} \to \mathrm{GL}(V)$ and $\rho_W : \mathfrak{G} \to \mathrm{GL}(W)$ are injective. The injectivity implies $\rho_V((\mathfrak{g}_i^{(t+1)})^{-1}\mathfrak{g}_j^{(t)})$ and $\rho_W((\mathfrak{g}_i^{(t+1)})^{-1}\mathfrak{g}_j^{(t)})$ come from the group element $(\mathfrak{g}_i^{(t+1)})^{-1}\mathfrak{g}_j^{(t)}$. They capture the same geometric information defined by $(\mathfrak{g}_i^{(t+1)})^{-1}\mathfrak{g}_j^{(t)}$. With assumptions on the injectivity of UPD and AGG, we can show that equivariant GNNs that learn steerable features on faithful representations can distinguish any two $k$-hop distinct geometric graphs.

**Theorem 2.** *Consider 1-hop equivariant GNNs learning features on steerable vector space $V$ where the aggregate function AGG learns features on steerable vector space $W$. Suppose $V$ and $W$ are faithful representations, and UPD and AGG satisfy certain assumptions on the injectivity outlined in Proposition 8 in the appendix. Then with $k$ iterations, these equivariant GNNs learn different multisets of node features $\{\!\!\{\boldsymbol{f}_i^{(k)}\}\!\!\}$ on two $k$-hop distinct geometric graphs.*

**Remark 3.** *It can be verified that $D^1$ and $D_{\mathrm{ind}}^1$, which correspond to type-1 steerable vector spaces $V_1$ for $\mathrm{SO}(3)$ and $V_{1,\mathrm{ind}}$ for $\mathrm{O}(3)$, are faithful representations. As a result, we conclude that equivariant GNNs defined in equation (1), that propagating steerable features up to type $L > 0$, all exhibit the*

*same capability to capture geometric information between two local neighborhoods. For additional discussion on the importance of faithfulness, refer to Appendix B.*

To mitigate the limited ability of invariant GNNs mentioned above, several approaches have emerged to encode geometric information between local structures into edge attributes. For instance, ComENet introduces the concept of *completeness* of edge attributes, a property associated with invariant edge features. Completeness enables these edge features to uniquely determine the spatial embedding $X$ up to isometries. ComENet achieves this completeness by encoding spherical coordinates (in triplets) and dihedral angles (in quadruplets) as edge features, which involves computation within 2-hop neighborhoods. The former specifies the locations of nodes within 1-hop neighborhoods, and the latter captures the angle between two 1-hop neighborhoods that share two nodes, thus addressing the issue of missing geometric information between these neighborhoods. Consequently, while ComENet employs 1-hop aggregation schemes, these edge attributes effectively contain all the necessary geometric information within and between 1-hop neighborhoods for molecular tasks [40].

However, it's important to note that dihedral angles may not always be well-defined in quadruplets[4], particularly when facing a chain structure within the graph, such as cis-trans stereoisomers or Fig. 1. Additionally, the complete edge attributes that ComENet utilizes are extracted within 2-hop neighborhoods, while Theorem 1 suggests that $k$-hop invariant GNNs may still struggle to capture the true global geometry effectively. This result for ComENet is empirically validated in Section 5.

**Some remarks.** We have pointed out that $k$-hop invariant GNNs inevitably ignore the information that captures the geometry between $k$-hop neighborhoods. Namely, they are limited to the information within $k$-hop neighborhoods. While encoding complete edge attributes emerges as a potential remedy, it remains an open question whether constructing complete edge attributes can be achieved for other tasks beyond molecular graphs, especially when dihedral angles cannot be well-defined in quadruplets. Theorem 1 further suggests that **invariant features confined to specific $k$-hop neighborhoods may not be sufficient to capture the accurate global geometry and global invariant features of geometric graphs**, thereby emphasizing the need for encoding global features as a potential solution to address this limitation. We leave this intriguing avenue for future research.

### 3.2 Comparing equivariant GNNs using different types of steerable features

Remark 3 suggests that when equivariant GNNs learn steerable features up to type $L$, they exhibit the same capability to capture geometric information between two local neighborhoods. To understand if the performance of equivariant GNNs can be enhanced by increasing $L$, we analyze the information-carrying ability of steerable features by investigating their corresponding invariant features.

Lemma 1 states that any type-$l$ steerable feature $f(X) \in V_l$ corresponds to a $(2l+1)$-dimensional invariant feature $\lambda(X)$ due to the decomposition $f(X) = D^l(\mathfrak{g}_X)\lambda(X)$. Now, we raise the question: **Does any $(2l+1)$-dimensional invariant feature $\lambda(X)$ correspond to a type-$l$ steerable feature $f(X)$?** Note that this question is essentially asking whether the space of all type-$l$ steerable features $f(X)$ has a dimension of $2l+1$ since $D^l(\mathfrak{g}_X)$ is invertible. While Lemma 1 is not affected by $\mathrm{rank}(X)$, the rank of the spatial embedding $X$, the answer to this question is contingent upon the rank. This is because equivariant functions must obey "Curie's principle" [36]. For a brief intuition, refer to Fig. 4 in Appendix G. Based on this principle, we present the following results, answering the aforementioned question. Specifically, we first examine $\mathfrak{G}$-equivariant functions without focusing on a spatial embedding and then shift our attention to a spatial embedding to obtain a precise answer.

**Theorem 3.** *Let $\mathbb{X}_r$ denote the set $\{X \in \mathbb{R}^{3\times m} \mid \mathrm{rank}(X) = r\}$. Then we have a one-to-one correspondence between* $\mathrm{O}(3)$*-equivariant functions and* $\mathrm{O}(3)$*-invariant functions:*

$$\{f : \mathbb{X}_3 \to V_{l,\mathrm{ind}} \mid f : \mathrm{O}(3)\text{-}equivariant\} \rightleftarrows \{\lambda : \mathbb{X}_3 \to V_0^{\oplus 2l+1} \mid \lambda : \mathrm{O}(3)\text{-}invariant\}, \quad (7)$$

$$\{f : \mathbb{X}_2 \to V_{l,\mathrm{ind}} \mid f : \mathrm{O}(3)\text{-}equivariant\} \rightleftarrows \{\lambda : \mathbb{X}_2 \to V_0^{\oplus l+1} \mid \lambda : \mathrm{O}(3)\text{-}invariant\},$$

$$\{f : \mathbb{X}_1 \to V_{l,\mathrm{ind}} \mid f : \mathrm{O}(3)\text{-}equivariant\} \rightleftarrows \{\lambda : \mathbb{X}_1 \to V_0^{\oplus 1} \mid \lambda : \mathrm{O}(3)\text{-}invariant\},$$

$$\{f : \mathbb{X}_0 \to V_{l,\mathrm{ind}} \mid f : \mathrm{O}(3)\text{-}equivariant\} = \{f : \mathbb{X}_0 = \{\mathbf{0}\} \to \{\mathbf{0}\}\}.$$

$$\{f : \mathbb{X}_3 \to V_{l,\mathrm{aug}} \mid f : \mathrm{O}(3)\text{-}equivariant\} \rightleftarrows \{\lambda : \mathbb{X}_3 \to V_0^{\oplus 2l+1} \mid \lambda : \mathrm{O}(3)\text{-}invariant\}, \quad (8)$$

$$\{f : \mathbb{X}_2 \to V_{l,\mathrm{aug}} \mid f : \mathrm{O}(3)\text{-}equivariant\} \rightleftarrows \{\lambda : \mathbb{X}_2 \to V_0^{\oplus l} \mid \lambda : \mathrm{O}(3)\text{-}invariant\},$$

$$\{f : \mathbb{X}_1 \to V_{l,\mathrm{aug}} \mid f : \mathrm{O}(3)\text{-}equivariant\} = \{f : \mathbb{X}_1 \to \{\mathbf{0}\}\},$$

$$\{f : \mathbb{X}_0 \to V_{l,\mathrm{aug}} \mid f : \mathrm{O}(3)\text{-}equivariant\} = \{f : \mathbb{X}_0 = \{\mathbf{0}\} \to \{\mathbf{0}\}\}.$$

---

[4]When the nodes are collinear, there are infinitely many planes containing them.

**Remark 4.** *Note that any $\mathfrak{G}$-equivariant function $f : \mathbb{R}^{3 \times m} \to V$ can be expressed as a summation: $f = \sum_{r=0}^{3} f \cdot \mathbf{1}_{\mathbb{X}_r}(\boldsymbol{X})$, where $\mathbf{1}_{\mathbb{X}_r}$ represents the indicator function, and $f \cdot \mathbf{1}_{\mathbb{X}_r}(\boldsymbol{X})$ can be considered as a $\mathfrak{G}$-equivariant function that maps from $\mathbb{X}_r$ to $V$. Consequently, the results presented above suffice to describe any $O(3)$-equivariant function that maps to a steerable vector space.*

**Corollary 1.** *Let $\boldsymbol{X} \in \mathbb{R}^{3 \times m}$ be a spatial embedding. We have the following relation between $O(3)$-steerable features and invariant features:*

1. *If $\mathrm{rank}(\boldsymbol{X}) = 3$, there is a bijection between steerable features in $V_{l,\mathrm{ind}}$ and $(2l+1)$-dimensional invariant features, as well as a bijection between steerable features in $V_{l,\mathrm{aug}}$ and $(2l+1)$-dimensional invariant features.*
2. *If $\mathrm{rank}(\boldsymbol{X}) = 2$, there is a bijection between steerable features in $V_{l,\mathrm{ind}}$ and $(l+1)$-dimensional invariant features and a bijection between steerable features in $V_{l,\mathrm{aug}}$ and $l$-dimensional invariant features.*
3. *If $\mathrm{rank}(\boldsymbol{X}) = 1$, there is a bijection between steerable features in $V_{l,\mathrm{ind}}$ and $1$-dimensional invariant features, while there is no non-trivial steerable feature lying in $V_{l,\mathrm{aug}}$.*
4. *There exist only trivial steerable feature $\mathbf{0}$ and trivial invariant feature $0$ if $\mathrm{rank}(\boldsymbol{X}) = 0$.*

Similar results for $SO(3)$ are presented in Theorem 5 and Corollary 4 in Appendix C. Due to the reducibility of $\mathfrak{G}$-steerable vector spaces, we can decompose any $\mathfrak{G}$-steerable vector space into a direct sum of steerable vector spaces of different types. Then, we have the following two corollaries:

**Corollary 2.** *Let $\mathbb{X}_3$ denote the set $\{\boldsymbol{X} \in \mathbb{R}^{3 \times m} \mid \mathrm{rank}(\boldsymbol{X}) = 3\}$. Then for any $\mathfrak{G}$-steerable vector space of dimension $d$, denoted as $V$, we have a one-to-one correspondence:*

$$\{f : \mathbb{X}_3 \to V \mid f : \mathfrak{G}\text{-equivariant}\} \rightleftarrows \{\lambda : \mathbb{X}_3 \to V_0^{\oplus d} \mid \lambda : \mathfrak{G}\text{-invariant}\}, \qquad (9)$$

*where the map between these two spaces is induced by the map defined in equation (3).*

**Corollary 3.** *Let $V$ and $W$ be two $\mathfrak{G}$-steerable vector spaces of dimension $d$. Then for any $\mathfrak{G}$-equivariant function $f_V : \mathbb{X}_3 \to V$, there is a $\mathfrak{G}$-equivariant function $f_W : \mathbb{X}_3 \to W$ such that for any $\boldsymbol{X} \in \mathbb{X}_3$, we have $f_V(\boldsymbol{X}) = \rho_V(\mathfrak{g}_{\boldsymbol{X}})\lambda(\boldsymbol{X})$ and $f_W(\boldsymbol{X}) = \rho_W(\mathfrak{g}_{\boldsymbol{X}})\lambda(\boldsymbol{X})$ for the same $\mathfrak{G}$-invariant function $\lambda$ where $\rho_V, \rho_W$ are the group representation on $V$ and $W$, resp.*

Consider learning steerable features in $V$ and $W$, resp. Corollary 3 suggests that if $\dim V = \dim W$, regardless of their irreducible decomposition, any (learnable) $\mathfrak{G}$-equivariant function $f_V : \mathbb{X}_3 \to V$ can be replaced by a $\mathfrak{G}$-equivariant function $f_W : \mathbb{X}_3 \to W$ where they learn the same corresponding invariant features. That is, **the invariant features carried by steerable features is primarily characterized by the feature dimension – independent of the highest type utilized.**

Specifically, we have the following equivalence among geometric GNNs, detailed in Appendix C:

**Theorem 4.** *Consider two geometric GNNs learning features on steerable vector spaces $V$ and $W$ of the same dimension, resp. Denote their update and aggregation functions at iteration $t$ as $\mathrm{UPD}_V^{(t)}, \mathrm{UPD}_W^{(t)}$ and $\mathrm{AGG}_V^{(t)}, \mathrm{AGG}_W^{(t)}$. Then for any collection $\{(\mathrm{UPD}_V^{(t)}, \mathrm{AGG}_V^{(t)})\}_t$, there exists a collection $\{(\mathrm{UPD}_W^{(t)}, \mathrm{AGG}_W^{(t)})\}_t$ such that for any fully connected graph, they learn the same corresponding invariant features $\lambda_i^{(t)}$ for any iteration $t \geq 0$ on each node $i$.*

Theorem 4 holds for any representation, especially non-faithful representations, but relies on the assumption of fully connected graphs. This result establishes the equivalence of geometric GNNs on fully connected graphs; this is similar to the equivalence of IGWL and GWL tests on fully connected graphs [20] but without strong assumptions on the injectivity of update functions and aggregate functions. Additionally, when dealing with non-fully connected graphs, our earlier exploration in Section 3.1 highlighted that learning features on faithful representations versus non-faithful representations results in different expressive powers. This discrepancy arises primarily because each node in a non-fully connected graph can only capture the global geometry through message passing.

**Some remarks.** Our proof of Theorem 4 relies on a precise understanding of how these models capture global geometry through message passing, making it challenging to ascertain its validity for non-fully connected graphs under the assumption of faithfulness of representations. While we believe it may be more feasible to demonstrate this by considering specific architectures and gaining a better understanding of how they obtain global geometry from local information, we consider exploring this aspect further as a future work. Nevertheless, Theorem 2 and Remark 3 suggest that equivariant

GNNs learning steerable features up to type-$L$ exhibit the same capacity to capture global geometry. We propose that when the feature dimension remains constant, the performance of equivariant GNNs employing steerable features up to type-$L$ may not increase as $L$ grows. However, we cannot assert that using $L = 1$ is sufficient. The concept of expressiveness includes two key aspects: the capacity of features to carry information and the ability of a model to extract it – the latter is commonly referred to as universality [11]. We intended to focus on the former, as the latter is subject to the architecture of the given model, while the former is not. Moreover, due to the lack of regularity of functions appearing in Lemma 1, we decided to defer discussions on universality to future works. More detailed discussions and potential methods are available in Appendix B.

## 4 ADDITIONAL RELATED WORKS

**Numerical comparisons in steerable feature types.** In a recent experiment in [20], a comparison was made regarding using different types of steerable features. However, this experiment was not specifically designed as an invariant classification task. Therefore, the conclusion that higher-type steerable features are superior may not directly

| $L$ | $c$ | # Param. | Feat. Dim. |
|---|---|---|---|
| 2 | 256 | 39M | 2304 |
| 2 | 824 | 113M | 7416 |
| 6 | 256 | 107M | 7424 |

Table 1: Total parameter count (# Param.) and steerable feature dimension (Steer. Dim.) for eSCN models with varying order of steerable features $L$ and steerable channels $c$.

apply to this context. Additionally, in experiments conducted in [3; 23; 2; 28], it was indeed observed that higher-type steerable features improved performance. Nevertheless, it should be noted that these experiments do not maintain fixed dimensions for hidden features, making direct comparisons challenging. Table 1 illustrates the difference in the number of parameters and the steerable feature dimension for the eSCN model [28] with varying feature type $L$ and channels $c$. The ablation study in [28] compares rows 1&3, while rows 2&3 provide a more suitable comparison.

## 5 EXPERIMENTS

In this section, we empirically verify our theory on several benchmark tasks. First, we verify the limitations of invariant models using the synthetic $k$-chain dataset. Second, we perform an ablation study over the steerable feature dimension when training steerable models on the large-scale OC20 IS2RE and S2EF datasets [6]. The invariant models considered are SchNet [34], DimeNet++ [14], SphereNet [25], and ComENet [40]; the equivariant models include EGNN [32], ClofNet [9], and GVP [19]; the steerable models are eSCN [28], EquiformerV2 [24] and MACE [2]. We provide details on each model, training procedure and hyperparameters in Appendix E, F.3, and F.4 resp.

$k$**-Chain.** In the $k$-chain task, motivated by [20] and illustrated in Figure 1, we aim to differentiate chains using the orientation of the terminal node. Each chain is made of $k + 2$ nodes, all possessing constant features. The connecting edges are undirected with uniform distance $r_{ij} = 5$. Chains are assigned binary labels based on the orientation of their final node. The dataset consists of pairs of chains, one of each label, which undergoes the same random rotation/reflection and translation. 50 transformed graph pairs are split into 50%/30%/20% train/validation/test splits with balanced labels.

| Layers | 1 | 2 | 3 | 1 | 2 | 3 | 4 |
|---|---|---|---|---|---|---|---|
| $k$-hop chain | $k = 2$ | $k = 2$ | $k = 2$ | $k = 3$ | $k = 3$ | $k = 3$ | $k = 3$ |
| | | | $L = 0$ | | | | |
| SchNet | $50.0 \pm 0.0$ | $50.0 \pm 0.0$ | $50.0 \pm 0.0$ | $50.0 \pm 0.0$ | $50.1 \pm 0.2$ | $50.0 \pm 0.0$ | $50.0 \pm 0.0$ |
| DimeNet++ | $50.0 \pm 0.0$ | $50.0 \pm 0.0$ | $50.0 \pm 0.0$ | $50.0 \pm 0.0$ | $50.0 \pm 0.0$ | $50.0 \pm 0.0$ | $50.0 \pm 0.0$ |
| SphereNet | $50.0 \pm 0.0$ | $50.0 \pm 0.0$ | $50.0 \pm 0.0$ | $50.0 \pm 0.0$ | $50.0 \pm 0.0$ | $50.0 \pm 0.0$ | $50.0 \pm 0.0$ |
| ComENet | $55.0 \pm 4.5$ | $59.0 \pm 11.6$ | $53.0 \pm 6.4$ | $54.0 \pm 6.2$ | $50.0 \pm 0.0$ | $46.5 \pm 5.0$ | $51.0 \pm 2.0$ |
| EquiformerV2 | $71.0 \pm 3.0$ | $76.0 \pm 8.0$ | $83.0 \pm 6.4$ | $43.0 \pm 9.0$ | $67.0 \pm 4.6$ | $67.9 \pm 9.0$ | $61.0 \pm 5.4$ |
| | | | $L = 1$ | | | | |
| EGNN | $50.0 \pm 0.0$ | $100.0 \pm 0.0$ | $95.0 \pm 15.0$ | $50.0 \pm 0.0$ | $50.0 \pm 0.0$ | $90.0 \pm 20.0$ | $100.0 \pm 0.0$ |
| GVP | $50.0 \pm 0.0$ | $100.0 \pm 0.0$ | $100.0 \pm 0.0$ | $50.0 \pm 0.0$ | $92.5 \pm 16.0$ | $91.5 \pm 17.3$ | $95.0 \pm 15.0$ |
| ClofNet | $50.0 \pm 0.0$ | $50.0 \pm 0.0$ | $100.0 \pm 0.0$ | $50.0 \pm 0.0$ | $50.0 \pm 0.0$ | $100.0 \pm 0.0$ | $100.0 \pm 0.0$ |
| MACE | $50.0 \pm 0.0$ | $100.0 \pm 0.0$ | $100.0 \pm 0.0$ | $50.0 \pm 0.0$ | $100.0 \pm 0.0$ | $100.0 \pm 0.0$ | $100.0 \pm 0.0$ |
| eSCN | $64.0 \pm 8.0$ | $60.5 \pm 10.0$ | $64.3 \pm 18.2$ | $53.0 \pm 4.6$ | $63.0 \pm 9.0$ | $60.0 \pm 13.4$ | $56.0 \pm 10.2$ |
| EquiformerV2 | $90.0 \pm 0.0$ | $95.0 \pm 5.0$ | $96.0 \pm 4.9$ | $76.0 \pm 6.6$ | $84.0 \pm 6.6$ | $92.0 \pm 6.0$ | $98.0 \pm 4.0$ |
| | | | $L = 2$ | | | | |
| MACE | $50.0 \pm 0.0$ | $100.0 \pm 0.0$ | $100.0 \pm 0.0$ | $50.0 \pm 0.0$ | $100.0 \pm 0.0$ | $100.0 \pm 0.0$ | $100.0 \pm 0.0$ |
| eSCN | $62.0 \pm 7.5$ | $61.0 \pm 9.4$ | $52.0 \pm 4.0$ | $62.0 \pm 10.8$ | $59.0 \pm 9.4$ | $56.0 \pm 10.2$ | $54.0 \pm 6.6$ |
| EquiformerV2 | $73.0 \pm 4.6$ | $88.0 \pm 4.0$ | $86.0 \pm 4.9$ | $86.0 \pm 4.9$ | $89.0 \pm 3.0$ | $88.0 \pm 4.0$ | $83.0 \pm 9.0$ |

Table 2: Test accuracy for the $k$-chain dataset with different $k$s. Models are further distinguished by their use of type-$L$ features. Cell shading is based on two standard deviations above or below the expected value. Unit:%.

Table 2 reports the 10-fold cross-validation mean test accuracy and standard deviation for varying chain length $k$ and model depth. The values are shaded based on the expected value from Section 3. In general, we observe that models perform as expected. The outliers can be categorized as equivariant models EGNN and ClofNet and quasi-equivariant models eSCN and EquiformerV2. The consistent underperformance of the equivariant models can be attributed to over-squashing as discussed in [20].

The abnormal performance of the quasi-equivariant models eSCN and EquiformerV2 can be attributed to the error in their equivariance. More discussions on models with outlying performance are provided in Appendix F.1. Table 2 shows that invariant features $L = 0$ are insufficient to distinguish the geometry regardless of the model depth. Also, architectures with $L = 1$ or 2 and finite $k$-hops can learn with sufficient depth. We show further comparisons up to $k = 4$ in Appendix F.1.

**IS2RE.** The OC20 IS2RE dataset [6] is a large-scale molecular property prediction task that uses a molecule's initial structure to predict its adsorption energy. In this task, we perform an ablation study on the order of the steerable features $L$ and the size of the steerable feature dimension $c$. In particular, we select a value $c$ to roughly fix the steerable feature dimension when varying $L$.

| Model | $L$ | $c$ | Feat. Dim. | # Param. | Loss ↓ | Energy MAE [meV] ↓ | EwT [%] ↑ |
|---|---|---|---|---|---|---|---|
| eSCN | 2 | 206 | 1854 | 10M | **0.369 ± 0.006** | 842 ± 13 | **1.94 ± 0.12** |
| eSCN | 4 | 98 | 1862 | 9M | 0.408 ± 0.006 | 929 ± 15 | 1.74 ± 0.12 |
| eSCN | 6 | 64 | 1856 | 8M | 0.3836 ± 0.003 | 872 ± 6 | 1.91 ± 0.19 |
| EquiformerV2 | 2 | 34 | 306 | 9M | 0.369 ± 0.009 | 841 ± 21 | 2.02 ± .14 |
| EquiformerV2 | 4 | 16 | 304 | 15M | **0.364 ± 0.005** | **832 ± 11** | **2.03 ± 0.14** |

Table 3: Validation results of the steerable model ablation study on $L$ and $c$ over 4-folds of the IS2RE dataset with 10k training molecules. We observe that higher type-$L$ steerable models may not perform best.

We list the ablation study in Table 3, showing that when the steerable feature dimension is fixed, lower order type-$L$ steerable features may provide improved results. These results are further validated in Figure 2, which illustrates the validation curves over 4-fold cross-validation with plotted mean and shaded standard deviation. We include additional results for EquiformerV2 in Appendix F.2.

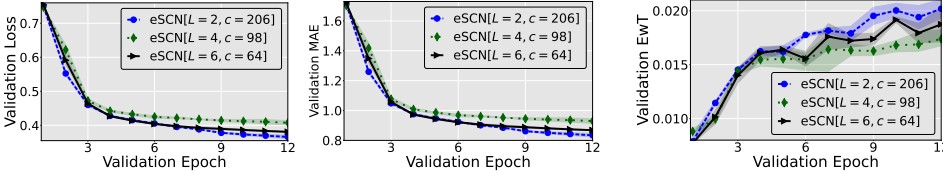

Figure 2: Validation results for the ablation study on eSCN for the IR2SE dataset with 10k training molecules. Depicted is the loss (left), energy MAE (middle), and energy within the threshold(EwT) (right) over 12 validation epochs. We plot the mean over four runs and shade the standard deviation.

**S2EF.** The OC20 S2EF dataset [6] uses the molecular structure to predict molecular adsorption energies and per-atom forces. Table 1 lists the size and the steerable feature dimension for each model. Table 4 reports the results of our training procedure against the reported results from [28]. We observe that with a fixed feature dimension and fewer training

| Config of eSCN | Energy MAE [meV] ↓ | Force MAE [meV/Å] ↓ | Force Cos ↑ | EFwT [%] ↑ |
|---|---|---|---|---|
| $L = 2, c = 256^*$ | 307 | 26.7 | 0.577 | 0.94 |
| $L = 6, c = 256^*$ | 294 | **21.3** | **0.653** | 1.45 |
| $L = 2, c = 824$ | **246** | 23.1 | 0.596 | **1.77** |

Table 4: Validation results after 8 training epochs for the eSCN model on the S2EF dataset with 2M training molecules. Results marked with * are reported from [28] and average over four runs after 12 training epochs.

epochs, the eSCN[$L = 2, c = 824$] consistently outperforms its lower dimensional counterpart eSCN[$L = 2, c = 256$] and outperforms eSCN[$L = 6, c = 256$] on energy MAE and EFwT. These tasks do not rule out confounding factors like over-squashing [1], but strongly support our results.

## 6 CONCLUDING REMARKS

We analyze the advantages granted by steerable features in GNNs. Specifically, we highlight two key findings: (1) Propagating steerable features of type-$L \geq 1$ equips geometric GNNs with inherent ability to automatically capture the geometry between local structures and obtain global invariant features from local ones. However, relying solely on propagating invariant features confined to specific $k$-hop neighborhoods is insufficient for enabling geometric GNNs to capture global information of all geometric graphs precisely. (2) When keeping the feature dimension constant, increasing using steerable features up to type-$L$ cannot essentially improve the performance of equivariant GNNs. In sum, our findings highlight the necessity of incorporating global features, extending beyond $k$-hop neighborhoods for a fixed $k$, to achieve the same expressiveness in invariant GNNs as in equivariant GNNs. Additionally, the traditional trade-off between performance and computational cost of using steerable features in equivariant GNNs should be reevaluated. Specifically, when maintaining a constant feature dimension, achieving better performance may not be guaranteed by using higher-type steerable features in equivariant GNNs, and this choice may come with a computational overhead.

## ACKNOWLEDGEMENT

This material is based on research sponsored by NSF grants DMS-1952339, DMS-1952644, DMS-2151235, DMS-2152762, DMS-2152717, DMS-2219956, DMS-2219904, and DMS-2208361 and DOE grant DE-SC0023490.

## ETHICS STATEMENT

Our paper focuses on developing new theoretical understandings of the benefits of steerable features of different types in 3D equivariant graph neural networks. The paper is mainly theoretical and we do not see any potential ethical issues in our research.

## REPRODUCIBILITY STATEMENT

We are dedicated to upholding the principles of reproducible research. In pursuit of this commitment, we have taken several measures. For the theoretical proofs, we have included comprehensive derivations to enhance accessibility for a wide range of readers. Additionally, we have submitted the code, along with detailed documentation, to facilitate the reproduction of the numerical results presented in our paper.

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

## A  ADDITIONAL RELATED WORKS

**Characterization of equivariant functions.**  The characterization of some equivariant functions has been explored in [39]. Specifically, Lemma 1 is analogous to Proposition 4 in [39], as it can be interpreted as expressing $f(\boldsymbol{X})$ as a linear combination of the column vectors in $\rho(\mathfrak{g}_{\boldsymbol{X}})$ with $\lambda(\boldsymbol{X})$ as the coefficients. However, a significant distinction lies in the fact that the number of vectors involved in our linear combination is the same as the dimension of $f(\boldsymbol{X})$, not the number of column vectors in $\boldsymbol{X}$. Additionally, the vectors in our linear combination truly form a basis, ensuring the uniqueness of the coefficients $\lambda(\boldsymbol{X})$. Meanwhile, the vectors in [39]'s linear combination do not necessarily form a basis and therefore cannot guarantee the uniqueness of the coefficients. Furthermore, our results cover cases of higher-type steerable features ($l > 1$), whereas [39] only studied $l = 0$ and 1.

**Equivariant moving frames and scalarization with local frames.** The existing concepts, such as equivariant moving frames [27; 29] and scalarization with local frames [9; 10], exhibit similarities with Lemma 1. However, we emphasize the distinctions as follows: While equivariant moving frames in [27] and our $\mathfrak{g}_{\boldsymbol{X}}$ both assign $\boldsymbol{X}$ to a single element in $\mathfrak{G}$, the distinction lies in the fact that $\mathfrak{g}_{\boldsymbol{X}}$ is well-defined even in cases where the action of $\mathfrak{G}$ is not free, making it not necessarily equivariant. The scalarization proposed in [9; 10] shares similarities with how we define $\lambda(\boldsymbol{X}) \coloneqq \rho(\mathfrak{g}_{\boldsymbol{X}})^{-1} \cdot f(\boldsymbol{X})$. Nevertheless, it differs because the local frames used for scalarization are generated from two coordinates rather than all the coordinates in $\boldsymbol{X}$. In contrast, our $\mathfrak{g}_{\boldsymbol{X}}$ requires consideration of the entire spatial embedding $\boldsymbol{X}$.

**Alignment.** Lemma 1 indicates that steerable features correspond to certain invariant features. However, implementing this in practice may pose challenges. The selection of the group element $\mathfrak{g}_{\boldsymbol{X}}$ and the representative $c(\mathfrak{G} \cdot \boldsymbol{X})$ is closely related to the notion of alignment in [46; 44]. In [46], the Kabsch alignment algorithm is adopted to optimally find $c(\mathfrak{G} \cdot \boldsymbol{X})$, while [44] utilizes an encoder to learn it. Nevertheless, it is worth mentioning that $\lambda(\boldsymbol{X}) \coloneqq f(c(\mathfrak{G} \cdot \boldsymbol{X}))$ might not be well-behaved. The topological properties of the Lie group $\mathfrak{G} = \mathrm{SO}(3)$ or $\mathrm{O}(3)$ prevent us from constructing a continuous function $\boldsymbol{X} \mapsto \mathfrak{g}_{\boldsymbol{X}}$, resulting in poor regularity  of $\boldsymbol{X} \mapsto \mathfrak{g}_{\boldsymbol{X}}$ and $\lambda(\boldsymbol{X})$.

**Various notions of steerable features.** Several related concepts surround steerable features in the literature, and it's beneficial to briefly recap them. The notion of a "steerable vector space" [4] refers to a vector space transformed by a group $\mathfrak{G}$ through a group representation of $\mathfrak{G}$. "Steerable feature fields" [42; 41] pertain to assigning each point a steerable vector. On the other hand, "steerable feature vectors" [37] refer to the actual outputs of these steerable feature fields. In the context of this paper, we use the term "steerable feature" to specifically refer to the steerable vectors generated by an equivariant map of the spatial embedding. Our notion aligns closely with the steerable feature fields concept but emphasizes including all relevant spatial information as input. This distinction helps clarify the focus and scope of our work in geometric graph neural networks.

## B  ADDITIONAL REMARKS AND DISCUSSION

**Significance of faithful representations in learning steerable features.** It's worth noting that [20] suggests the extensibility of their setup to higher-order tensors. However, our finding indicates that similar results for this extension may not hold if the geometric objects lack faithfulness. One can see that faithfulness is necessary to guarantee the injectivity in the proof of Proposition 8 and Theorem 2.

To illustrate this issue, we can straightforwardly extend the argument from Theorem 1. Let's consider a 1-hop equivariant GNN learning features on non-faithful representations, such as $\boldsymbol{D}_{\mathrm{aug}}^0$ or $\boldsymbol{D}_{\mathrm{aug}}^1$. Our aim is to demonstrate the existence of two 2-hop distinct geometric graphs for which the considered 1-hop equivariant GNN produces the same feature representation for any iteration.

While our discussion here is somewhat informal, the details we provided can be used to establish a solid proof easily. Notice that the non-faithfulness of representations implies that features may remain unchanged even when the input undergoes a transformation by some group element $\mathfrak{g}$. This aligns with the challenge of using invariant features, which are expected to remain unchanged under any transformation.

In fact, it is sufficient to construct two 2-hop distinct geometric graphs, where their 1-hop subgraphs are identical up to certain group actions (either $\mathfrak{g}$ or the identity). This pair of graphs can be obtained by adjusting the pair illustrated in Fig. 1. The analogous argument in the proof of Theorem 1 will

then demonstrate the desired results. The underlying rationale for this approach lies in the fact that aggregating features on non-faithful representations, due to their nature, fail to capture the locally changed geometry induced by the group element $\mathfrak{g}$.

We did not explicitly delve into this in the main paper, as existing models typically focus on learning features involving faithful representations ($l = 1$). However, a crucial determining factor for achieving the maximal expressive power in equivariant GNNs is whether the steerable features lie on faithful representations.

**Limitations: universality and the lack of regularity.** In the final part of Section 3.2, we refrain from asserting that using $L = 1$ is sufficient due to the unaddressed consideration of universality. Here, we delve into the challenges of studying the intricate relationship between universality and internal representation, specifically the representation involved in steerable features. Additionally, we present a potential method to address them.

Let us formally articulate the questions pivotal to the investigation of the relationship between universality and internal representation.

Consider a $\mathfrak{G}$-steerable vector space $V$, and let $\mathbb{X}_3$ denote the set $\{\boldsymbol{X} \in \mathbb{R}^{3 \times m} \mid \operatorname{rank}(\boldsymbol{X}) = 3\}$. For a specific architecture of geometric GNNs capable of learning features on any $\mathfrak{G}$-steerable vector space, we denote the family of all possible continuous $\mathfrak{G}$-equivariant functions from $\mathbb{X}_3$ to $V$ that this architecture can parameterize (with a sufficient number of layers) as $\mathcal{A}_V$. The family consisting of all continuous $\mathfrak{G}$-equivariant functions from $\mathbb{X}_3$ to $V$ is denoted as $\mathcal{F}_V$.

The first question to address is as follows:

**Question 1.** *Can any continuous $\mathfrak{G}$-equivariant function in $\mathcal{F}_V$ be uniformly approximated on compact sets by functions in $\mathcal{A}_V$?*

Now, let $W$ be another $\mathfrak{G}$-steerable vector space with the same dimension as $V$. Let $f_V : \mathbb{X}_3 \to V$ be an arbitrary continuous $\mathfrak{G}$-equivariant function in $\mathcal{F}_V$. According to Corollary 3, there exists a $\mathfrak{G}$-equivariant function $f_W : \mathbb{X}_3 \to W$ such that for any $\boldsymbol{X} \in \mathbb{X}_3$, we have $f_V(\boldsymbol{X}) = \rho_V(\mathfrak{g}_{\boldsymbol{X}})\lambda(\boldsymbol{X})$ and $f_W(\boldsymbol{X}) = \rho_W(\mathfrak{g}_{\boldsymbol{X}})\lambda(\boldsymbol{X})$ for the same $\mathfrak{G}$-invariant function $\lambda$ where $\rho_V, \rho_W$ are the group representation on $V$ and $W$, resp. However, $f_W$ is not necessarily continuous, i.e. $f_W \notin \mathcal{F}_W$, implying that it may not be approximated by functions in $\mathcal{A}_W$. In particular, the topological properties of the Lie group $\mathfrak{G} = \operatorname{SO}(3)$ or $\operatorname{O}(3)$ prevent us from constructing a continuous function $\boldsymbol{X} \mapsto \mathfrak{g}_{\boldsymbol{X}}$, resulting in poor regularity of $\boldsymbol{X} \mapsto \mathfrak{g}_{\boldsymbol{X}}$ and $\lambda(\boldsymbol{X})$. Consequently, the continuity of $f_W$ cannot be guaranteed.

As a result, we present the following question:

**Question 2.** *Let $\mu$ denote the Lebesgue measure on $\mathbb{R}^{3 \times m}$. For any $\epsilon > 0$ and any compact set $K \subset \mathbb{X}_3$, does there exist a continuous $\mathfrak{G}$-equivariant function $f'_W : \mathbb{X}_3 \to W$ and a measurable subset $E \subset K$ with $\mu(K \setminus E) < \epsilon$ such that $f'_W(\boldsymbol{X}) = f_W(\boldsymbol{X})$ for all $\boldsymbol{X} \in E$?*

Suppose we obtain affirmative answers to both Question 1 and 2. This implies that for any $\epsilon > 0$ and any compact set $K \subset \mathbb{X}_3$, there exists a function $f'_W \in \mathcal{F}_W$ that can be uniformly approximated on compact sets by functions in $\mathcal{A}_W$. Additionally, its corresponding invariant function $\lambda'(\boldsymbol{X}) := \rho_W(\mathfrak{g}_{\boldsymbol{X}})^{-1} f'_W(\boldsymbol{X})$ is identical to $\lambda(\boldsymbol{X})$, the corresponding invariant function of $f_V(\boldsymbol{X})$, on some measurable set $E$ with $\mu(K \setminus E) < \epsilon$.

By restricting $f_V$ to a function in $\mathcal{A}_V$, we observe that the given architecture is able to learn a function in $\mathcal{A}_W$ such that its corresponding invariant function is identical to the corresponding invariant function of $f_V$ on any compact set except an $\epsilon$-small measurable subset. In essence, the architecture can successfully learn approximated corresponding invariant functions on steerable vector spaces of the same dimension.

While the response to Question 1 depends on the architectures, the answer to Question 2 is independent of them. In the following context, we propose a potential approach to address Question 2 by constructing $\boldsymbol{X} \mapsto \mathfrak{g}_{\boldsymbol{X}}$ with a better regularity[5]. More precisely, we aim to remedy the regularity issue of $\boldsymbol{X} \mapsto \mathfrak{g}_{\boldsymbol{X}}$ by extending the domain $\mathbb{X}_3$ to its covering space $\widetilde{\mathbb{X}_3}$. We proceed similarly to [29] to construct the covering space for the set of all generic spatial embeddings in

---

[5]This will not impact the validity of Lemma 1 and Corollary 3.

$\mathbb{X}_3$. Namely, given the spatial embedding $\boldsymbol{X} \in \mathbb{X}_3$, Principle Component Analysis (PCA) suggests that there are three vectors $\boldsymbol{v}_1, \boldsymbol{v}_2, \boldsymbol{v}_3 \in \mathbb{R}^3$ uniquely determined up to sign corresponding to three distinct singular values $\sigma_1 > \sigma_2 > \sigma_3$. As a result, one obtains the equivariant moving frame $\boldsymbol{X} \mapsto \mathcal{F}(\boldsymbol{X}) := \{[\alpha_1 \boldsymbol{v}_1 \quad \alpha_2 \boldsymbol{v}_2 \quad \alpha_3 \boldsymbol{v}_3] \in \mathrm{O}(3) | \alpha_i \in \{1, -1\}\}$ as a set-valued function. In our situation, we take an alternative viewpoint. Notice that any generic $\boldsymbol{X} \in \mathbb{X}_3$, there's a small enough neighborhood $U_{\boldsymbol{X}}$ of $\boldsymbol{X}$ such that any choice of sign $(\alpha_1, \alpha_2, \alpha_3)$ gives a homeomorphism $\boldsymbol{X} \mapsto (\boldsymbol{X}, [\alpha_1 \boldsymbol{v}_1 \quad \alpha_2 \boldsymbol{v}_2 \quad \alpha_3 \boldsymbol{v}_3])$ from $U_{\boldsymbol{X}} \subset \mathbb{X}_3$ to $U_{\boldsymbol{X}, \alpha_1, \alpha_2, \alpha_3} \subset \mathbb{X}_3 \times \mathrm{O}(3)$. One may check that the neighborhood compatibility allows us to patch $U_{\boldsymbol{X}, \alpha_1, \alpha_2, \alpha_3}$s into a topological space $\widetilde{\mathbb{X}_3}$. Intuitively speaking, we obtain $\widetilde{\mathbb{X}_3}$ via locally embedding multiple copies of small neighborhoods into $\mathbb{X}_3 \times \mathrm{O}(3)$. Each neighborhood $U_{\boldsymbol{X}}$ corresponds to $2^3$ copies of themselves in $\mathbb{X}_3 \times \mathrm{O}(3)$. It becomes clear that $\widetilde{\mathbb{X}_3} \hookrightarrow \mathbb{X}_3 \times \mathrm{O}(3) \to \mathrm{O}(3)$ defines a continuous equivariant function $\widetilde{\boldsymbol{X}} := (\boldsymbol{X}, \mathfrak{g}) \mapsto \mathfrak{g}_{\widetilde{\boldsymbol{X}}} := \mathfrak{g}$ and so does $\widetilde{\boldsymbol{X}} \mapsto \rho_W(\mathfrak{g}_{\widetilde{\boldsymbol{X}}})$. Now, for a given $f_W$ with the corresponding invariant function $\lambda$, we lift $\lambda$ to an invariant function $\widetilde{\boldsymbol{X}} \mapsto \widetilde{\lambda}(\widetilde{\boldsymbol{X}}) := \lambda\left(\pi_1(\widetilde{\boldsymbol{X}})\right)$ on the covering space. Indeed, $\widetilde{\boldsymbol{X}} \mapsto \rho_W(\mathfrak{g}_{\widetilde{\boldsymbol{X}}}) \widetilde{\lambda}(\widetilde{\boldsymbol{X}})$ produces a $\mathfrak{G}$-equivariant continuous function, but the function is defined on the covering space $\widetilde{\mathbb{X}_3}$. To produce a function on $\mathbb{X}_3$, one need to modify $\lambda$ on an $\epsilon$-small set and also $\widetilde{\lambda}$ so that the modified invariant function $\widetilde{\lambda}'$ remains continuous and $f_W'(\boldsymbol{X}) := \rho_W(\mathfrak{g}_{\widetilde{\boldsymbol{X}}}) \widetilde{\lambda}'(\widetilde{\boldsymbol{X}})$ actually defines a function on $\mathbb{X}_3$. In particular, $f_W'$ and $f_W$ coincide on a given compact set except an $\epsilon$-small measurable subset.

Although addressing Question 1 and 2 may suggest that learning on steerable vector spaces of the same dimension demonstrates consistent expressive power, we cannot deny the possibility that certain choices of representation could benefit the model, allowing it to approximate the target function with fewer layers. Additionally, our exploration in Section 3.1 indicates that learning on non-faithful representations might restrict the model's expressive power and, consequently, its universality. Nevertheless, we have chosen to reserve the exploration of Question 1 and 2 for future work.

While our theory reveals this limitation, it is noteworthy that empirical results surprisingly align with the claim that learning lower types of steerable features can perform as effectively as learning higher types of steerable features.

## C  MISSING PROOF

**Lemma 1.** *Let $V$ be a $d$-dimensional $\mathfrak{G}$-steerable vector space with the assigned group representation $\rho : \mathfrak{G} \to \mathrm{GL}(V)$. If $f : \mathbb{R}^{3 \times m} \to V$ is $\mathfrak{G}$-equivariant, then there exists a unique $\mathfrak{G}$-invariant function $\lambda : \mathbb{R}^{3 \times m} \to V_0^{\oplus d}$ s.t. $f(\boldsymbol{X}) = \rho(\mathfrak{g}_{\boldsymbol{X}}) \lambda(\boldsymbol{X})$, where $V_0$ denotes the 1D trivial representation of $\mathfrak{G}$[6]. In particular, the following map is well-defined*

$$\{f : \mathbb{R}^{3 \times m} \to V \mid f : \mathfrak{G}\text{-equivariant}\} \to \{\lambda : \mathbb{R}^{3 \times m} \to V_0^{\oplus d} \mid \lambda : \mathfrak{G}\text{-invariant}\}. \tag{3}$$

*Proof of Lemma 1.* The equivariance follows that,

$$f(\boldsymbol{X}) = f\left(\mathfrak{g}_{\boldsymbol{X}} \cdot c(\mathfrak{G} \cdot \boldsymbol{X})\right) = \rho(\mathfrak{g}_{\boldsymbol{X}}) f\left(c(\mathfrak{G} \cdot \boldsymbol{X})\right).$$

It suffices to show that $f\left(c(\mathfrak{G} \cdot \boldsymbol{X})\right)$ induces a $\mathfrak{G}$-invariant map $\lambda : \mathbb{R}^{3 \times m} \to V_0^{\oplus d}$. First, sending $\boldsymbol{X}$ to $f\left(c(\mathfrak{G} \cdot \boldsymbol{X})\right)$ defines a map from $\mathbb{R}^{3 \times m}$ to a $d$-dimensional vector space. Since $\mathfrak{G} \cdot (\mathfrak{g} \cdot \boldsymbol{X}) = \mathfrak{G} \cdot \boldsymbol{X}$ for any $\mathfrak{g} \in \mathfrak{G}$, we see that $c(\mathfrak{G} \cdot \boldsymbol{X})$ is invariant, and hence $f\left(c(\mathfrak{G} \cdot \boldsymbol{X})\right)$ is invariant to the group action. Therefore, $\lambda(\boldsymbol{X}) := f\left(c(\mathfrak{G} \cdot \boldsymbol{X})\right)$ is $\mathfrak{G}$-invariant. One can check that the uniqueness of this function directly stems from the linear independence of the columns of $\rho(\mathfrak{g}_{\boldsymbol{X}})$. Alternatively, the invertibility of $\rho(\mathfrak{g}_{\boldsymbol{X}})$ can be utilized to conclude that $\lambda(\boldsymbol{X}) = \rho(\mathfrak{g}_{\boldsymbol{X}})^{-1} f(\boldsymbol{X})$. □

For the ease of reading, we restate Theorem 1 below

**Theorem 1.** *If $\mathcal{G}_1$ and $\mathcal{G}_2$ are two $k$-hop identical graphs, then any iteration of $k$-hop invariant GNNs will get the same output from these two graphs. That is, there is a graph isomorphism $b$ such that $\lambda_i^{(t+1)} = \lambda_{b(i)}^{(t+1)}$ for any $i$, even though $\mathcal{G}_1$ and $\mathcal{G}_2$ may not be identical up to group action.*

---

[6]For $\mathfrak{G} = \mathrm{SO}(3)$, it corresponds to the type-0 steerable vector space we defined in Section 2. For simplicity, we employ the same notation here.

*Proof.* For $k$-hop invariant GNNs, the propagation of the (corresponding) invariant features in equation (6)[7] can be expressed as follows:

$$\lambda_i^{(t+1)} = \text{UPD}\left(\lambda_i^{(t)}, \text{AGG}\left(\{\!\{\lambda_i^{(t)}, \lambda_j^{(t)}, ((\mathfrak{g}_i^{(t+1)})^{-1}\mathfrak{g}_i^{(1)})(\mathfrak{g}_i^{(1)})^{-1}\boldsymbol{x}_{ij} \mid j \in N_i^{(k)}\}\!\}\right)\right)$$

$$= \text{UPD}\left(\lambda_i^{(t)}, \text{AGG}\left(\{\!\{\lambda_i^{(t)}, \lambda_j^{(t)}, \boldsymbol{x}_{ij} \mid j \in N_i^{(k)}\}\!\}\right)\right),$$

(10)

where we use $\rho(\mathfrak{g}) = \boldsymbol{I}$ for any $\mathfrak{g} \in \mathfrak{G}$ and the invariance property of AGG.

Let $\mathcal{G}_1 = (\mathcal{V}_1, \mathcal{E}_1, \boldsymbol{F}^{\mathcal{G}_1}, \boldsymbol{X}^{\mathcal{G}_1}), \mathcal{G}_2 = (\mathcal{V}_2, \mathcal{E}_2, \boldsymbol{F}^{\mathcal{G}_2}, \boldsymbol{X}^{\mathcal{G}_2})$ be two $k$-hop identical geometric graphs. By definition, there exists a graph isomorphism $b$ such that for any node $i \in \mathcal{V}_1$, we can find a group element $\mathfrak{g}_i \in \mathfrak{G}$ satisfies $\boldsymbol{f}_i^{\mathcal{G}_1} = \boldsymbol{f}_{b(i)}^{\mathcal{G}_2}, \boldsymbol{x}_i^{\mathcal{G}_1} = \mathfrak{g}_i \cdot \boldsymbol{x}_{b(i)}^{\mathcal{G}_2}$ and $\boldsymbol{f}_j^{\mathcal{G}_1} = \boldsymbol{f}_{b(j)}^{\mathcal{G}_2}, \boldsymbol{x}_j^{\mathcal{G}_1} = \mathfrak{g}_i \cdot \boldsymbol{x}_{b(j)}^{\mathcal{G}_2}$ whenever $j \in \mathcal{N}_i^{(k)}$.

Then, utilizing the invariance of AGG once more, we observe that:

$$\lambda_i^{(1)} = \text{UPD}\left(\boldsymbol{f}_i^{\mathcal{G}_1}, \text{AGG}\left(\{\!\{\boldsymbol{f}_i^{\mathcal{G}_1}, \boldsymbol{f}_j^{\mathcal{G}_1}, \boldsymbol{x}_{ij}^{\mathcal{G}_1} \mid j \in N_i^{(k)}\}\!\}\right)\right)$$

$$= \text{UPD}\left(\boldsymbol{f}_{b(i)}^{\mathcal{G}_2}, \text{AGG}\left(\{\!\{\boldsymbol{f}_{b(i)}^{\mathcal{G}_2}, \boldsymbol{f}_{b(j)}^{\mathcal{G}_2}, \mathfrak{g}_i \cdot \boldsymbol{x}_{b(i)b(j)}^{\mathcal{G}_2} \mid b(j) \in N_{b(i)}^{(k)}\}\!\}\right)\right)$$

$$= \text{UPD}\left(\boldsymbol{f}_{b(i)}^{\mathcal{G}_2}, \text{AGG}\left(\{\!\{\boldsymbol{f}_{b(i)}^{\mathcal{G}_2}, \boldsymbol{f}_{b(j)}^{\mathcal{G}_2}, \boldsymbol{x}_{b(i)b(j)}^{\mathcal{G}_2} \mid b(j) \in N_{b(i)}^{(k)}\}\!\}\right)\right)$$

$$= \lambda_{b(i)}^{(1)}.$$

Hence, it follows that $\lambda_i^{(1)} = \lambda_{b(i)}^{(1)}$ for any $i$. Following this step inductively, we can demonstrate that $\lambda_i^{(t+1)} = \lambda_{b(i)}^{(t+1)}$ for any node $i$. $\qquad\square$

For the ease of reading, we restate Theorem 4 below:

**Theorem 4.** *Consider two geometric GNNs learning features on steerable vector spaces $V$ and $W$ of the same dimension, resp. Denote their update and aggregation functions at iteration $t$ as $\text{UPD}_V^{(t)}, \text{UPD}_W^{(t)}$ and $\text{AGG}_V^{(t)}, \text{AGG}_W^{(t)}$. Then for any collection $\{(\text{UPD}_V^{(t)}, \text{AGG}_V^{(t)})\}_t$, there exists a collection $\{(\text{UPD}_W^{(t)}, \text{AGG}_W^{(t)})\}_t$ such that for any fully connected graph, they learn the same corresponding invariant features $\lambda_i^{(t)}$ for any iteration $t \geq 0$ on each node $i$.*

Let us first provide more details about what we are going to show. We consider the following two geometric GNNs:

$$\boldsymbol{f}_{i,*}^{(t+1)} = \text{UPD}_*\left(\boldsymbol{f}_{i,*}^{(t)}, \text{AGG}_*(\{\!\{\boldsymbol{f}_{i,*}^{(t)}, \boldsymbol{f}_{j,*}^{(t)}, \boldsymbol{x}_{ij} \mid j \in \mathcal{N}_i\}\!\})\right)$$

(11)

where $*$ represents the steerable vector space $V$ or $W$ and $\boldsymbol{f}_{i,*}^{(t)}$ denotes the $i$-th node feature learn on $*$ at iteration $t$. Let $\rho_V$ and $\rho_W$ denote the group representations on $V$ and $W$, resp. Remark that the corresponding invariant features are given by:

$$\lambda_{i,*}^{(t)} = \rho_*(\mathfrak{g})^{-1}\boldsymbol{f}_{i,*}^{(t)}$$

(12)

where $\mathfrak{g} = \mathfrak{g}_{\boldsymbol{X}}$ is the group element associated with input spatial embedding $\boldsymbol{X}$.

We aim to show that for any collection $\{(\text{UPD}_V^{(t)}, \text{AGG}_V^{(t)})\}_t$ of $\mathfrak{G}$-equivariant functions, there exists a collection $\{(\text{UPD}_W^{(t)}, \text{AGG}_W^{(t)})\}_t$ of $\mathfrak{G}$-equivariant functions such that $\lambda_{i,V}^{(t)} = \lambda_{i,W}^{(t)}$ for any $t \geq 0$ and $i$.

*Proof.* We proceed with this proof by induction. First, $\lambda_{i,V}^{(t)} = \lambda_{i,W}^{(t)}$ holds for $t = 0$ since

$$\lambda_{i,*}^{(0)} = \boldsymbol{f}_{i,*}^{(0)} = \boldsymbol{f}_i,$$

---

[7]One can also directly use equation (5).

where $\boldsymbol{f}_i$ denotes the input node feature of node $i$. Suppose $\lambda_{i,V}^{(t)} = \lambda_{i,W}^{(t)}$ holds for any node $i$ at iteration $t$. At iteration $t + 1$, we have

$$
\begin{aligned}
\lambda_{i,V}^{(t+1)} &= \rho_V(\mathfrak{g})^{-1} \boldsymbol{f}_{i,V}^{(t)} \\
&= \rho_V(\mathfrak{g})^{-1} \operatorname{UPD}_V \left( \boldsymbol{f}_{i,V}^{(t)}, \operatorname{AGG}_V(\{\!\!\{ \boldsymbol{f}_{i,V}^{(t)}, \boldsymbol{f}_{j,V}^{(t)}, \boldsymbol{x}_{ij} \mid j \in \mathcal{N}_i \}\!\!\}) \right) \\
&= \operatorname{UPD}_V \left( \rho_V(\mathfrak{g})^{-1} \boldsymbol{f}_{i,V}^{(t)}, \operatorname{AGG}_V(\{\!\!\{ \rho_V(\mathfrak{g})^{-1} \boldsymbol{f}_{i,V}^{(t)}, \rho_V(\mathfrak{g})^{-1} \boldsymbol{f}_{j,V}^{(t)}, \mathfrak{g}^{-1} \boldsymbol{x}_{ij} \mid j \in \mathcal{N}_i \}\!\!\}) \right) \\
&= \operatorname{UPD}_V \left( \lambda_{i,V}^{(t)}, \operatorname{AGG}_V(\{\!\!\{ \lambda_{i,V}^{(t)}, \lambda_{j,V}^{(t)}, \mathfrak{g}^{-1} \boldsymbol{x}_{ij} \mid j \in \mathcal{N}_i \}\!\!\}) \right).
\end{aligned}
$$

Consider the following construction of $(\operatorname{UPD}_W^{(t)}, \operatorname{AGG}_W^{(t)})$:

$$
\begin{aligned}
&\operatorname{UPD}_W \left( \boldsymbol{f}_{i,W}^{(t)}, \operatorname{AGG}_W(\{\!\!\{ \boldsymbol{f}_{i,W}^{(t)}, \boldsymbol{f}_{j,W}^{(t)}, \boldsymbol{x}_{ij} \mid j \in \mathcal{N}_i \}\!\!\}) \right) := \\
&\rho_W(\mathfrak{g}) \operatorname{UPD}_V \left( \rho_W(\mathfrak{g})^{-1} \boldsymbol{f}_{i,W}^{(t)}, \operatorname{AGG}_V(\{\!\!\{ \rho_W(\mathfrak{g})^{-1} \boldsymbol{f}_{i,W}^{(t)}, \rho_W(\mathfrak{g})^{-1} \boldsymbol{f}_{j,W}^{(t)}, \mathfrak{g}^{-1} \boldsymbol{x}_{ij} \mid j \in \mathcal{N}_i \}\!\!\}) \right)
\end{aligned}
\tag{13}
$$

More precisely, to guarantee $\mathfrak{G}$-equivariance, we define $\operatorname{UPD}_W^{(t)}$ and $\operatorname{AGG}_W^{(t)}$ to be: $(\operatorname{UPD}_W^{(t)}, \operatorname{AGG}_W^{(t)})$:

$$
\begin{aligned}
&\operatorname{AGG}_W \left( \{\!\!\{ \boldsymbol{f}_{i,W}^{(t)}, \boldsymbol{f}_{j,W}^{(t)}, \boldsymbol{x}_{ij} \mid j \in \mathcal{N}_i \}\!\!\} \right) \\
&:= \left( \rho_W(\mathfrak{g}) \operatorname{AGG}_V(\{\!\!\{ \rho_W(\mathfrak{g})^{-1} \boldsymbol{f}_{i,W}^{(t)}, \rho_W(\mathfrak{g})^{-1} \boldsymbol{f}_{j,W}^{(t)}, \mathfrak{g}^{-1} \boldsymbol{x}_{ij} \mid j \in \mathcal{N}_i \}\!\!\}), \mathfrak{g} \right) \\
&\operatorname{UPD}_W \left( \boldsymbol{f}_{i,W}^{(t)}, (\boldsymbol{m}_i, \mathfrak{g}) \right) := \rho_W(\mathfrak{g}) \operatorname{UPD}_V \left( \rho_W(\mathfrak{g})^{-1} \boldsymbol{f}_{i,W}^{(t)}, \rho_W(\mathfrak{g})^{-1} \cdot \boldsymbol{m}_i \right)
\end{aligned}
\tag{14}
$$

where $\boldsymbol{m}_i$ denotes the first component of $\operatorname{AGG}_W^{(t)}$)'s output. The $\mathfrak{G}$-equivariance follows from the sense in Remark 1.

It is well-defined since we can construct $\mathfrak{g} = \mathfrak{g}_{\boldsymbol{X}}$ from $\boldsymbol{x}_{ij}$ by using the assumption of fully connected graphs. Indeed, since we have $\sum_k \boldsymbol{x}_k = 0$ (from eliminating the effects of translations), then $\sum_{j \neq i} \boldsymbol{x}_{ij} = m\boldsymbol{x}_i - \sum_{j \neq i} \boldsymbol{x}_j = (m+1)\boldsymbol{x}_i$ where $m$ is the number of nodes. Therefore, $\frac{1}{m+1} \sum_{k \neq i} \boldsymbol{x}_{ik} = \boldsymbol{x}_i$ and $\left( \frac{1}{m+1} \sum_{k \neq i} \boldsymbol{x}_{ik} \right) - \boldsymbol{x}_{ij} = \boldsymbol{x}_j$ for any $j \neq i$. These coordinates then determine $\boldsymbol{X}$ and hence determine $\mathfrak{g}_{\boldsymbol{X}}$.

Then using the inductive assumption $\lambda_{i,V}^{(t)} = \lambda_{i,W}^{(t)}$ and $\lambda_{i,*}^{(t)} = \rho_*(\mathfrak{g})^{-1} \boldsymbol{f}_{i,*}^{(t)}$, we obtain

$$
\begin{aligned}
\lambda_{i,W}^{(t+1)} &= \rho_W(\mathfrak{g})^{-1} \boldsymbol{f}_{i,W}^{(t)} \\
&= \rho_W(\mathfrak{g})^{-1} \operatorname{UPD}_W \left( \boldsymbol{f}_{i,W}^{(t)}, \operatorname{AGG}_W(\{\!\!\{ \boldsymbol{f}_{i,W}^{(t)}, \boldsymbol{f}_{j,W}^{(t)}, \boldsymbol{x}_{ij} \mid j \in \mathcal{N}_i \}\!\!\}) \right) \\
&= \operatorname{UPD}_V \left( \rho_W(\mathfrak{g})^{-1} \boldsymbol{f}_{i,W}^{(t)}, \operatorname{AGG}_V(\{\!\!\{ \rho_W(\mathfrak{g})^{-1} \boldsymbol{f}_{i,W}^{(t)}, \rho_W(\mathfrak{g})^{-1} \boldsymbol{f}_{j,W}^{(t)}, \mathfrak{g}^{-1} \boldsymbol{x}_{ij} \mid j \in \mathcal{N}_i \}\!\!\}) \right) \\
&= \operatorname{UPD}_V \left( \lambda_{i,W}^{(t)}, \operatorname{AGG}_V(\{\!\!\{ \lambda_{i,W}^{(t)}, \lambda_{j,W}^{(t)}, \mathfrak{g}^{-1} \boldsymbol{x}_{ij} \mid j \in \mathcal{N}_i \}\!\!\}) \right) \\
&= \operatorname{UPD}_V \left( \lambda_{i,V}^{(t)}, \operatorname{AGG}_V(\{\!\!\{ \lambda_{i,V}^{(t)}, \lambda_{j,V}^{(t)}, \mathfrak{g}^{-1} \boldsymbol{x}_{ij} \mid j \in \mathcal{N}_i \}\!\!\}) \right) \\
&= \lambda_{i,V}^{(t+1)}.
\end{aligned}
$$

This shows the construction satisfies the desired result.

$\square$

**Proofs and Additional Details for Section 3.2.** All the theoretical results in Section 3.2 stem from the following theorem:

**Theorem 3.** *Let $\mathbb{X}_r$ denote the set $\{ \boldsymbol{X} \in \mathbb{R}^{3 \times m} \mid \operatorname{rank}(\boldsymbol{X}) = r \}$. Then we have a one-to-one correspondence between $\mathrm{O}(3)$-equivariant functions and $\mathrm{O}(3)$-invariant functions:*

$$
\{ f : \mathbb{X}_3 \to V_{l,\mathrm{ind}} \mid f : \mathrm{O}(3)\text{-}equivariant \} \rightleftarrows \{ \lambda : \mathbb{X}_3 \to V_0^{\oplus 2l+1} \mid \lambda : \mathrm{O}(3)\text{-}invariant \}, \tag{7}
$$

$$
\{ f : \mathbb{X}_2 \to V_{l,\mathrm{ind}} \mid f : \mathrm{O}(3)\text{-}equivariant \} \rightleftarrows \{ \lambda : \mathbb{X}_2 \to V_0^{\oplus l+1} \mid \lambda : \mathrm{O}(3)\text{-}invariant \},
$$

$$
\{ f : \mathbb{X}_1 \to V_{l,\mathrm{ind}} \mid f : \mathrm{O}(3)\text{-}equivariant \} \rightleftarrows \{ \lambda : \mathbb{X}_1 \to V_0^{\oplus 1} \mid \lambda : \mathrm{O}(3)\text{-}invariant \},
$$

$$
\{ f : \mathbb{X}_0 \to V_{l,\mathrm{ind}} \mid f : \mathrm{O}(3)\text{-}equivariant \} = \{ f : \mathbb{X}_0 = \{\boldsymbol{0}\} \to \{\boldsymbol{0}\} \}.
$$

$$\{f : \mathbb{X}_3 \to V_{l,\text{aug}} \mid f : \text{O}(3)\text{-}equivariant\} \rightleftarrows \{\lambda : \mathbb{X}_3 \to V_0^{\oplus 2l+1} \mid \lambda : \text{O}(3)\text{-}invariant\}, \quad (8)$$

$$\{f : \mathbb{X}_2 \to V_{l,\text{aug}} \mid f : \text{O}(3)\text{-}equivariant\} \rightleftarrows \{\lambda : \mathbb{X}_2 \to V_0^{\oplus l} \mid \lambda : \text{O}(3)\text{-}invariant\},$$

$$\{f : \mathbb{X}_1 \to V_{l,\text{aug}} \mid f : \text{O}(3)\text{-}equivariant\} = \{f : \mathbb{X}_1 \to \{\mathbf{0}\}\},$$

$$\{f : \mathbb{X}_0 \to V_{l,\text{aug}} \mid f : \text{O}(3)\text{-}equivariant\} = \{f : \mathbb{X}_0 = \{\mathbf{0}\} \to \{\mathbf{0}\}\}.$$

**Theorem 5.** *Similarly, we have a one-to-one correspondence for* $\text{SO}(3)$*-equivariance and* $\text{SO}(3)$*-invariance:*

$$\{f : \mathbb{X}_3 \to V_l \mid f : \text{SO}(3)\text{-}equivariant\} \rightleftarrows \{\lambda : \mathbb{X}_3 \to V_0^{\oplus 2l+1} \mid \lambda : \text{SO}(3)\text{-}invariant\}, \quad (15)$$

$$\{f : \mathbb{X}_2 \to V_l \mid f : \text{SO}(3)\text{-}equivariant\} \rightleftarrows \{\lambda : \mathbb{X}_2 \to V_0^{\oplus 2l+1} \mid \lambda : \text{SO}(3)\text{-}invariant\},$$

$$\{f : \mathbb{X}_1 \to V_l \mid f : \text{SO}(3)\text{-}equivariant\} \rightleftarrows \{\lambda : \mathbb{X}_1 \to V_0^{\oplus 1} \mid \lambda : \text{SO}(3)\text{-}invariant\},$$

$$\{f : \mathbb{X}_0 \to V_l \mid f : \text{SO}(3)\text{-}equivariant\} = \{f : \mathbb{X}_0 = \{\mathbf{0}\} \to \{\mathbf{0}\}\}.$$

Consider $\mathfrak{G}$ as either $\text{O}(3)$ or $\text{SO}(3)$. Given the reducibility of $\mathfrak{G}$-steerable vector spaces, we can decompose any $\mathfrak{G}$-steerable vector space $V$ into a direct sum of steerable vector spaces of different types. Then equation (7), equation (8) in Theorem 3, and equation (15) in Theorem 5 imply the following result:

**Corollary 2.** *Let* $\mathbb{X}_3$ *denote the set* $\{\boldsymbol{X} \in \mathbb{R}^{3 \times m} \mid \text{rank}(\boldsymbol{X}) = 3\}$. *Then for any* $\mathfrak{G}$*-steerable vector space of dimension* $d$, *denoted as* $V$, *we have a one-to-one correspondence:*

$$\{f : \mathbb{X}_3 \to V \mid f : \mathfrak{G}\text{-}equivariant\} \rightleftarrows \{\lambda : \mathbb{X}_3 \to V_0^{\oplus d} \mid \lambda : \mathfrak{G}\text{-}invariant\}, \quad (9)$$

*where the map between these two spaces is induced by the map defined in equation (3).*

**Corollary 3.** *Let* $V$ *and* $W$ *be two* $\mathfrak{G}$*-steerable vector spaces of dimension* $d$. *Then for any* $\mathfrak{G}$*-equivariant function* $f_V : \mathbb{X}_3 \to V$, *there is a* $\mathfrak{G}$*-equivariant function* $f_W : \mathbb{X}_3 \to W$ *such that for any* $\boldsymbol{X} \in \mathbb{X}_3$, *we have* $f_V(\boldsymbol{X}) = \rho_V(\mathfrak{g}_{\boldsymbol{X}})\lambda(\boldsymbol{X})$ *and* $f_W(\boldsymbol{X}) = \rho_W(\mathfrak{g}_{\boldsymbol{X}})\lambda(\boldsymbol{X})$ *for the same* $\mathfrak{G}$*-invariant function* $\lambda$ *where* $\rho_V, \rho_W$ *are the group representation on* $V$ *and* $W$, *resp.*

*Proof.* According to Lemma 1, there is a $\mathfrak{G}$-invariant function $\lambda : \mathbb{X}_3 \to V_0^{\oplus d}$ such that $f_V(\boldsymbol{X}) = \rho_V(\mathfrak{g}_{\boldsymbol{X}})\lambda(\boldsymbol{X})$ for any $\boldsymbol{X} \in \mathbb{X}_3$. Then applying Theorem 2 to $W$, there exist a $\mathfrak{G}$-equivariant function $f_W : \mathbb{X}_3 \to W$ such that $f_W(\boldsymbol{X}) = \rho_W(\mathfrak{g}_{\boldsymbol{X}})\lambda(\boldsymbol{X})$ for any $\boldsymbol{X} \in \mathbb{X}_3$, which shows the desired result. $\square$

The following two theorems follow from Theorem 3 and 5 by focusing on a given spatial embedding.

**Corollary 1.** *Let* $\boldsymbol{X} \in \mathbb{R}^{3 \times m}$ *be a spatial embedding. We have the following relation between* $\text{O}(3)$*-steerable features and invariant features:*

1. *If* $\text{rank}(\boldsymbol{X}) = 3$, *there is a bijection between steerable features in* $V_{l,\text{ind}}$ *and* $(2l + 1)$*-dimensional invariant features, as well as a bijection between steerable features in* $V_{l,\text{aug}}$ *and* $(2l + 1)$*-dimensional invariant features.*
2. *If* $\text{rank}(\boldsymbol{X}) = 2$, *there is a bijection between steerable features in* $V_{l,\text{ind}}$ *and* $(l + 1)$*-dimensional invariant features and a bijection between steerable features in* $V_{l,\text{aug}}$ *and* $l$*-dimensional invariant features.*
3. *If* $\text{rank}(\boldsymbol{X}) = 1$, *there is a bijection between steerable features in* $V_{l,\text{ind}}$ *and* 1*-dimensional invariant features, while there is no non-trivial steerable feature lying in* $V_{l,\text{aug}}$.
4. *There exist only trivial steerable feature* $\mathbf{0}$ *and trivial invariant feature* 0 *if* $\text{rank}(\boldsymbol{X}) = 0$.

**Corollary 4.** *Let* $\boldsymbol{X} \in \mathbb{R}^{3 \times m}$ *be a spatial embedding. We have the following relation between* $\text{SO}(3)$*-steerable features and invariant features:*

1. *If* $\text{rank}(\boldsymbol{X}) = 2$ *or* 3, *there is a bijection between type-*$l$ *steerable features and* $(2l + 1)$*-dimensional invariant features.*

2. *If* $\text{rank}(\boldsymbol{X}) = 1$, *there is a bijection between type-*$l$ *steerable features and* 1*-dimensional invariant features.*

3. *If* $\text{rank}(\boldsymbol{X}) = 0$, *there exist only trivial steerable feature* $\mathbf{0}$ *and trivial invariant feature* 0.

**Remark 5.** *It is worth mentioning that Corollary 1 and 4 suggests that type-*$l$ *steerable features* $(l > 0)$ *is more sensitive to the rank of the spatial embedding* $\boldsymbol{X}$ *than invariant features* $(l = 0)$.

Before proving Theorem 3 and 5, we introduce the following lemma:

**Lemma 2.** *For any fixed $l \geq 0$, any $\mathrm{SO}(3)$-equivariant function $f^{(l)} : \mathbb{R}^{3 \times m} \to V_l$ has a unique decomposition: $f^{(l)} = f^{(l)}_{\mathrm{ind}} + f^{(l)}_{\mathrm{aug}}$ into a sum of $\mathrm{O}(3)$-equivariant functions $f^{(l)}_{\mathrm{ind}} : \mathbb{R}^{3 \times m} \to V_{l,\mathrm{ind}}$ and $f^{(l)}_{\mathrm{aug}} : \mathbb{R}^{3 \times m} \to V_{l,\mathrm{aug}}$, where*

$$f^{(l)}_{\mathrm{ind}}(\boldsymbol{X}) := \frac{f^{(l)}(\boldsymbol{X}) + (-1)^l \cdot f^{(l)}(-\boldsymbol{X})}{2} \quad and \quad f^{(l)}_{\mathrm{aug}}(\boldsymbol{X}) := \frac{f^{(l)}(\boldsymbol{X}) - (-1)^l \cdot f^{(l)}(-\boldsymbol{X})}{2}.$$

*Proof of **Lemma** 2.* Upon routine verification, it becomes apparent that:

$$f^{(l)}_{\mathrm{ind}}(-\boldsymbol{X}) = (-1)^l \cdot f^{(l)}_{\mathrm{ind}}(\boldsymbol{X}) \quad \text{and} \quad f^{(l)}_{\mathrm{aug}}(-\boldsymbol{X}) = (-1)^{l+1} \cdot f^{(l)}_{\mathrm{aug}}(\boldsymbol{X}).$$

Since both $f^{(l)}(\boldsymbol{X})$ and $f^{(l)}(-\boldsymbol{X})$ are $\mathrm{SO}(3)$-equivariant, so are both $f^{(l)}_{\mathrm{ind}}$ and $f^{(l)}_{\mathrm{aug}}$. Combined with the sign-change property, we conclude that $f^{(l)}_{\mathrm{ind}}$ and $f^{(l)}_{\mathrm{aug}}$ are $\mathrm{O}(3)$-equivariant. To establish uniqueness, let's consider any decomposition of $f^{(l)} = g^{(l)}_{\mathrm{ind}} + h^{(l)}_{\mathrm{aug}}$ into the sum of a $\mathrm{O}(3)$-equivariant function $g^{(l)}_{\mathrm{ind}} : \mathbb{R}^{3 \times m} \to V_{l,\mathrm{ind}}$ and a $\mathrm{O}(3)$-equivariant function $h^{(l)}_{\mathrm{aug}} : \mathbb{R}^{3 \times m} \to V_{l,\mathrm{aug}}$. Now, observe that the functions on both sides of the following equation,

$$g^{(l)}_{\mathrm{ind}} - f^{(l)}_{\mathrm{ind}} = f^{(l)}_{\mathrm{aug}} - h^{(l)}_{\mathrm{aug}}$$

are $\mathrm{O}(3)$-equivariant, which, due to sign-change property:

$$(-1)^l \cdot \left( g^{(l)}_{\mathrm{ind}}(\boldsymbol{X}) - f^{(l)}_{\mathrm{ind}}(\boldsymbol{X}) \right) = g^{(l)}_{\mathrm{ind}}(-\boldsymbol{X}) - f^{(l)}_{\mathrm{ind}}(-\boldsymbol{X})$$

$$= f^{(l)}_{\mathrm{aug}}(-\boldsymbol{X}) - h^{(l)}_{\mathrm{aug}}(-\boldsymbol{X}) = (-1)^{l+1} \cdot \left( f^{(l)}_{\mathrm{aug}}(\boldsymbol{X}) - h^{(l)}_{\mathrm{aug}}(\boldsymbol{X}) \right)$$

$$= -(-1)^l \cdot \left( g^{(l)}_{\mathrm{ind}}(\boldsymbol{X}) - f^{(l)}_{\mathrm{ind}}(\boldsymbol{X}) \right)$$

can only be 0. In other words, $g^{(l)}_{\mathrm{ind}} = f^{(l)}_{\mathrm{ind}}$ and $h^{(l)}_{\mathrm{aug}} = f^{(l)}_{\mathrm{aug}}$. $\qquad\square$

*Proof of Theorem 3 and 5.* We can observe that Theorem 3 imply Theorem 5 by applying Lemma 2. Therefore, it remains to prove all the correspondences for $\mathrm{O}(3)$. The proof strategy here is similar to that of Lemma 1 with additional care on the design of the choice function $c : \mathbb{R}^{3 \times m}/\mathfrak{G} \to \mathbb{R}^{3 \times m}$. We recall the following decomposition:

$$\mathbb{R}^{3 \times m} = \bigsqcup_{r=0}^{3} \mathbb{X}_r, \quad \text{where} \quad \mathbb{X}_r := \left\{ \boldsymbol{X} \in \mathbb{R}^{3 \times m} \big| \mathrm{rank}\,(\boldsymbol{X}) = r \right\}.$$

Since $\mathfrak{G} := \mathbf{O}(3)$ action does not affect the rank of a spatial embedding $\boldsymbol{X}$, the decomposition is preserved under quotient:

$$\mathbb{R}^{3 \times m}/\mathfrak{G} = \bigsqcup_{r=0}^{3} \left( \mathbb{X}_r/\mathfrak{G} \right).$$

This allows us to define the choice function in ways that respect the geometry arising from different rank $\boldsymbol{X} = r$ conditions:

$(r = 3)$ We shall see that the argument here is a special case of Remark 1 applied on the subset $\mathbb{X}_3 \subset \mathbb{R}^{3 \times m}$. Invoking axiom of choice, we obtain a choice function:

$$c_3 : \mathbb{X}_3/\mathfrak{G} \to \mathbb{X}_3 \quad \text{with} \quad c_3 \left( \mathfrak{G} \cdot \boldsymbol{X} \right) \in \mathfrak{G} \cdot \boldsymbol{X}.$$

Notice that in the full-rank setting, all stabilizers are the same; in particular, they are trivial:

$$\mathfrak{G}_{\boldsymbol{X}} = \mathfrak{G}_{c_3(\mathfrak{G} \cdot \boldsymbol{X})} = \mathfrak{G}_3 = \{\boldsymbol{I}\}, \quad \forall \boldsymbol{X} \in \mathbb{X}_3.$$

$(r = 2)$ Here is where the novelty of our argument comes in. We observe that

$$(\mathfrak{G} \cdot \boldsymbol{X}) \cap \overset{x-y-\text{plane}}{\left( \mathbb{R}^2 \times \{0\} \right)^m} \neq \varnothing, \quad \forall \boldsymbol{X} \in \mathbb{X}_2.$$

In other words, we can take the following special choice function:

$$c_2 : \mathbb{X}_2/\mathfrak{G} \to \mathbb{X}_2 \cap \left(\mathbb{R}^2 \times \{0\}\right)^m \quad \text{with} \quad c_2\left(\mathfrak{G} \cdot \boldsymbol{X}\right) \in \mathfrak{G} \cdot \boldsymbol{X}.$$

Notably, all the choices here share the same stabilizer:

$$\mathfrak{G}_{c_2(\mathfrak{G} \cdot \boldsymbol{X})} = \mathfrak{G}_2 := \left\{ \boldsymbol{I}, \boldsymbol{R}_z := \begin{bmatrix} 1 & 0 & 0 \\ 0 & 1 & 0 \\ 0 & 0 & -1 \end{bmatrix} \right\}, \quad \forall \boldsymbol{X} \in \mathbb{X}_2.$$

This will be useful later on.

$(r = 1)$ We modify the argument in the previous $(r = 2)$ case to reflect the rank $\boldsymbol{X} = 1$ geometry and apply to the $(r = 1)$ case. Again, we start with the following observation:

$$(\mathfrak{G} \cdot \boldsymbol{X}) \cap \left(\{0\}^2 \overset{z-\text{axis}}{\times} \mathbb{R}\right)^m \neq \varnothing, \quad \forall \boldsymbol{X} \in \mathbb{X}_1.$$

We may, thus, take the following special choice function:

$$c_1 : \mathbb{X}_1/\mathfrak{G} \to \mathbb{X}_1 \cap \left(\{0\}^2 \times \mathbb{R}\right)^m \quad \text{with} \quad c_1\left(\mathfrak{G} \cdot \boldsymbol{X}\right) \in \mathfrak{G} \cdot \boldsymbol{X}.$$

Similarly, all the choices here share the same stabilizer:

$$\mathfrak{G}_{c_1(\mathfrak{G} \cdot \boldsymbol{X})} = \mathfrak{G}_1 := \left\{ \begin{bmatrix} \boldsymbol{R} & 0 \\ 0 & 1 \end{bmatrix} \in \mathfrak{G} \,\middle|\, \boldsymbol{R} \in \mathbf{O}(2) \right\}, \quad \forall \boldsymbol{X} \in \mathbb{X}_1.$$

$(r = 0)$ This is the trivial case $\mathbb{X}_0 = \{\boldsymbol{0}\}$, and thus the choice function is defined uniquely:

$$c_0 : \mathbb{X}_0/\mathfrak{G} = \{\mathfrak{G} \cdot \boldsymbol{0}\} \to \{\boldsymbol{0}\}.$$

And obviously, $\mathfrak{G}_{c_0(\mathfrak{G} \cdot \boldsymbol{0})} = \mathfrak{G}_0 = \mathfrak{G}$.

For convenience, we define the total choice function:

$$c := \sum_{r=0}^{3} c_r \cdot \mathbf{1}_{\mathbb{X}_r/\mathfrak{G}}.$$

Via axiom of choice again, we can find a $\mathfrak{G}$ valued function:

$$\mathfrak{g}_{(\cdot)} : \mathbb{R}^{3 \times m} \to \mathfrak{G} \quad \text{such that} \quad \boldsymbol{X} = \mathfrak{g}_{\boldsymbol{X}} \cdot c\left(\mathfrak{G} \cdot \boldsymbol{X}\right).$$

As a direct consequence, given $\mathbf{O}(3)$–equivariant $f : \mathbb{X}_r \to V_{l,*}$, we have the following formula:

$$f\left(\boldsymbol{X}\right) = \boldsymbol{D}_*^l\left(\mathfrak{g}_{\boldsymbol{X}}\right) \cdot f\left(c_r\left(\mathfrak{G} \cdot \boldsymbol{X}\right)\right), \quad \forall \boldsymbol{X} \in \mathbb{X}_r.$$

By design, the formula suggests a way to relate a $V_{l,*}$ steerable feature to a $(2l + 1)$–dimensional invariant features:

$$\lambda\left(\boldsymbol{X}\right) := f\left(c_r\left(\mathfrak{G} \cdot \boldsymbol{X}\right)\right) = \boldsymbol{D}_*^l\left(\mathfrak{g}_{\boldsymbol{X}}\right)^{-1} \cdot f\left(\boldsymbol{X}\right), \quad \forall \boldsymbol{X} \in \mathbb{X}_r.$$

Yet, upon further inspection, $\lambda\left(\boldsymbol{X}\right)$ has some hidden structure relating to the stabilizer $\mathfrak{G}_r$. To be precise, the $\mathbf{O}(3)$–equivariance of $f$ implies that

$$\boldsymbol{D}_*^l\left(g\right) \cdot \lambda\left(\boldsymbol{X}\right) = f\left(g \cdot c_r\left(\mathfrak{G} \cdot \boldsymbol{X}\right)\right) = f\left(c_r\left(\mathfrak{G} \cdot \boldsymbol{X}\right)\right) = \lambda\left(\boldsymbol{X}\right), \quad \forall g \in \mathfrak{G}_r.$$

In other words, $\lambda\left(\boldsymbol{X}\right)$ could lie in a proper subspace.

$$\lambda\left(\boldsymbol{X}\right) \in \bigcap_{g \in \mathfrak{G}_r} \ker\left(\boldsymbol{D}_*^l\left(g\right) - \boldsymbol{I}\right) =: \mathbb{V}_{r,*}.$$

We first go through the two trivial cases:

$$\begin{cases} \mathbb{V}_{3,*} = \ker\left(0\right) = \bigoplus_{-l \leq m \leq l} \mathbb{F} \cdot \boldsymbol{e}_m, \\ \mathbb{V}_{0,*} = \bigcap_{g \in \mathfrak{G}} \ker\left(\boldsymbol{D}_*^l\left(g\right) - \boldsymbol{I}\right) = \{\boldsymbol{0}\}. \end{cases}$$

For $r = 2$, $\mathbb{V}_{2,*}$ is exactly an eigenspace of $\boldsymbol{D}_*^l(\boldsymbol{R}_z)$:

$$\mathbb{V}_{2,*} = \ker\left(\boldsymbol{D}_*^l(\boldsymbol{R}_z) - \boldsymbol{I}\right).$$

A direct calculation shows the following:

$$\boldsymbol{D}_*^l(\boldsymbol{R}_z) = (-1)^{l+\mathbf{1}_{\mathrm{aug}}(*)} \boldsymbol{D}^l(\pi, 0, 0) = \left[\cdots (-1)^{m+l+\mathbf{1}_{\mathrm{aug}}(*)} \boldsymbol{e}_m \cdots\right]_{-l \le m \le l}.$$

Therefore, we have:

$$\mathbb{V}_{2,*} = \bigoplus_{\substack{-l \le m \le l, \\ m \underset{\mathrm{mod}2}{\equiv} l+\mathbf{1}_{\mathrm{aug}}(*)}} \mathbb{F} \cdot \boldsymbol{e}_m.$$

In particular, we have $\dim \mathbb{V}_{2,\mathrm{ind}} = l+1$, $\dim \mathbb{V}_{2,\mathrm{aug}} = l$, and $\mathbb{V}_{3,*} = \mathbb{V}_{2,\mathrm{ind}} \oplus \mathbb{V}_{2,\mathrm{aug}}$. To deal with the ($r = 1$) case, we first notice that $\mathfrak{G}_1$ can be generated by the following:

$$\boldsymbol{R}_y := \begin{bmatrix} 1 & 0 & 0 \\ 0 & -1 & 0 \\ 0 & 0 & 1 \end{bmatrix}, \quad \text{and} \quad \mathcal{R}_\alpha := \begin{bmatrix} \cos\alpha & -\sin\alpha & 0 \\ \sin\alpha & \cos\alpha & 0 \\ 0 & 0 & 1 \end{bmatrix}, \quad \alpha \in [0, 2\pi).$$

This simplifies our problem,

$$\mathbb{V}_{1,*} = \ker\left(\boldsymbol{D}_*^l(\boldsymbol{R}_y) - \boldsymbol{I}\right) \cap \bigcap_\alpha \ker\left(\boldsymbol{D}_*^l(\mathcal{R}_\alpha) - \boldsymbol{I}\right).$$

Direct calculation gives:

$$\boldsymbol{D}_*^l(\mathcal{R}_\alpha) = \boldsymbol{D}^l(\alpha, 0, 0) = \left[\cdots e^{-i\alpha m} \boldsymbol{e}_m \cdots\right]_{-l \le m \le l}.$$

Therefore, we must have:

$$\bigcap_\alpha \ker\left(\boldsymbol{D}_*^l(\mathcal{R}_\alpha) - \boldsymbol{I}\right) = \mathbb{F} \cdot \boldsymbol{e}_0.$$

It remains to check whether $\boldsymbol{e}_0$ lies in the eigenspace $\ker\left(\boldsymbol{D}_*^l(\boldsymbol{R}_y) - \boldsymbol{I}\right)$, or not. We perform the following calculation:

$$\boldsymbol{D}_*^l(\boldsymbol{R}_y) \cdot \boldsymbol{e}_0 = (-1)^{l+\mathbf{1}_{\mathrm{aug}}(*)} \cdot \boldsymbol{D}^l(0, \pi, 0) \cdot \boldsymbol{e}_0 = (-1)^{\mathbf{1}_{\mathrm{aug}}(*)} \cdot \boldsymbol{e}_0.$$

We may now conclude that $\mathbb{V}_{1,\mathrm{ind}} = \mathbb{F} \cdot \boldsymbol{e}_0$ and $\mathbb{V}_{1,\mathrm{aug}} = \{\boldsymbol{0}\}$. Combining what we have, we obtain the following picture:

$$f(\boldsymbol{X}) \xmapsto{\boldsymbol{D}_*^l(\mathfrak{g}\boldsymbol{x})^{-1}} f(c(\mathfrak{G} \cdot \boldsymbol{X})) \xxlongequal{\boldsymbol{X} \in \mathbb{X}_r} \lambda(\boldsymbol{X}) \quad \in \quad \mathbb{V}_{r,*},$$

with each space being characterized as follows:

$$\begin{cases} \mathbb{V}_{3,\mathrm{ind}}, \mathbb{V}_{3,\mathrm{aug}} = \displaystyle\bigoplus_{-l \le m \le l} \mathbb{F} \cdot \boldsymbol{e}_m & \simeq V_0^{\oplus 2l+1} \\[2ex] \mathbb{V}_{2,\mathrm{ind}} = \displaystyle\bigoplus_{\substack{-l \le m \le l, \\ m \underset{\mathrm{mod}2}{\equiv} l}} \mathbb{F} \cdot \boldsymbol{e}_m & \simeq V_0^{\oplus l+1} \\[3ex] \mathbb{V}_{2,\mathrm{aug}} = \displaystyle\bigoplus_{\substack{-l \le m \le l, \\ m \underset{\mathrm{mod}2}{\not\equiv} l}} \mathbb{F} \cdot \boldsymbol{e}_m & \simeq V_0^{\oplus l} \\[3ex] \mathbb{V}_{1,\mathrm{ind}} = \mathbb{F} \cdot \boldsymbol{e}_0 & \simeq V_0 \\[2ex] \mathbb{V}_{1,\mathrm{aug}}, \mathbb{V}_{0,\mathrm{ind}}, \mathbb{V}_{0,\mathrm{aug}} = \{\boldsymbol{0}\}. \end{cases}$$

We now argue that the following space of the invariant features

$$\Lambda_{r,*} := \{\lambda : \mathbb{X}_r \to \mathbb{V}_{r,*} | \lambda : O(3)\text{-invariant}\}$$

has 1–to–1 correspondence to the following space of steerable features

$$F_{r,*} \coloneqq \{f : \mathbb{X}_r \to V_{l,*} | f : \mathrm{O}(3)\text{-equivariant}\},$$

and the correspondence is exactly given by the following formula:

$$f\left(\boldsymbol{X}\right) = \boldsymbol{D}_*^l\left(\mathfrak{g}_{\boldsymbol{X}}\right) \cdot \lambda\left(\boldsymbol{X}\right).$$

Indeed, for $f \in F_{r,*}$, the invertibility of $\boldsymbol{D}_*^l\left(\mathfrak{g}_{\boldsymbol{X}}\right)$ defines a unique $\lambda$. Moreover, we've established that such $\lambda$ is $\mathrm{O}(3)$–invariant and has its images contained in $\mathbb{V}_{r,*}$ and thus, $\lambda \in \Lambda_{r,*}$. On the other hand, given $\lambda \in \Lambda_{r,*}$ and any $\boldsymbol{X} \in \mathbb{X}_r$, we have the following,

$$\boldsymbol{D}_*^l\left(\mathfrak{g}_{\boldsymbol{X}}\right) \cdot \lambda\left(X\right) = \boldsymbol{D}_*^l\left(\mathfrak{g}_{\boldsymbol{X}}\right) \cdot p_{\mathbb{V}_{r,*}} \cdot \lambda\left(X\right),$$

where $p_{\mathbb{V}_{r,*}}$ is the matrix that represent the orthogonal projection onto $\mathbb{V}_{r,*}$. Since

$$g \cdot \boldsymbol{X} = \begin{cases} \mathfrak{g}_{g\cdot\boldsymbol{X}} \cdot c_r\big(\underbrace{\mathfrak{G} \cdot g \cdot \boldsymbol{X}}_{=\mathfrak{G}\cdot\boldsymbol{X}}\big) \\ g \cdot \mathfrak{g}_{\boldsymbol{X}} \cdot c_r\left(\mathfrak{G} \cdot \boldsymbol{X}\right) \end{cases} \implies \mathfrak{g}_{g\cdot\boldsymbol{X}}^{-1} \cdot g \cdot \mathfrak{g}_{\boldsymbol{X}} \in \mathfrak{G}_r, \quad \forall \boldsymbol{X} \in \mathbb{X}_r,$$

we obtain

$$\left(\boldsymbol{D}_*^l\left(\mathfrak{g}_{g\cdot\boldsymbol{X}}^{-1} \cdot g \cdot \mathfrak{g}_{\boldsymbol{X}}\right) - \boldsymbol{I}\right) \cdot p_{\mathbb{V}_{r,*}} = \boldsymbol{0},$$

as a direct consequence. After some algebraic manipulation, we derive the following equation:

$$\boldsymbol{D}_*^l\left(\mathfrak{g}_{g\cdot\boldsymbol{X}}\right) \cdot p_{\mathbb{V}_{r,*}} = \boldsymbol{D}_*^l\left(g\right) \cdot \boldsymbol{D}_*^l\left(\mathfrak{g}_{\boldsymbol{X}}\right) \cdot p_{\mathbb{V}_{r,*}}, \quad \forall \boldsymbol{X} \in \mathbb{X}_r.$$

In other words, the formula

$$\lambda\left(\boldsymbol{X}\right) \longmapsto f\left(\boldsymbol{X}\right) \coloneqq \boldsymbol{D}_*^l\left(\mathfrak{g}_{\boldsymbol{X}}\right) \cdot \lambda\left(X\right) \xrightarrow{\boldsymbol{X}\in\mathbb{X}_r} \boldsymbol{D}_*^l\left(\mathfrak{g}_{\boldsymbol{X}}\right) \cdot p_{\mathbb{V}_{r,*}} \cdot \lambda\left(X\right),$$

defines an $\mathrm{O}(3)$–equivariant function $f \in F_{r,*}$. With that, we establish the 1–to–1 correspondence. $\qquad\square$

# D   GEOMETRIC WEISFEILER-LEHMAN TEST (GWL)

In this section, we will provide an overview of the geometric Weisfeiler-Lehman test (GWL) and its invariant version (IGWL), as originally introduced in [20]; however, for simplicity, we exclude input vector features. Additionally, we will extend IGWL to the $k$-hop setting and demonstrate that $k$-hop IGWL cannot distinguish any two $k$-hop identical graphs. As a corollary, we will establish that GWL remains strictly more powerful than $k$-hop IGWL. We will also prove the conditions under which $k$-hop invariant GNNs achieve the same expressive power as $k$-hop IGWL.

Before delving into the details, we introduce a graph-level readout, a $\mathfrak{G}$-invariant multiset function, at the final layer of geometric GNNs. This function maps multisets $\{\boldsymbol{f}_i^{(t)} \mid i \in \mathcal{V}\}$ to invariant features in $V_0^{\oplus d'}$ for some $d' > 0$. This inclusion aids in understanding when geometric GNNs attain the maximum expressive power characterized by the GWL test.

**GWL.** Consider a geometric graph $\mathcal{G} = (\mathcal{V}, \mathcal{E}, \boldsymbol{F}, \boldsymbol{X})$. Let $C$ denote a countable space of colors. Initially, we assign a scalar color $c_i^{(0)} \in C$ to each node $i \in \mathcal{V}$ through an injective mapping function HASH based on their input features $\boldsymbol{f}_i$:

$$c_i^{(0)} \coloneqq \mathrm{HASH}(\boldsymbol{f}_i). \tag{16}$$

Additionally, we assign an extra geometric object $g_i^{(0)}$ to each node $i \in \mathcal{V}$ by $g_i^{(0)} = c_i^{(0)}$.

Then we define the inductive step. Assuming we have all the colors $c_i^{(t-1)}$ and geometric objects $g_i^{(t-1)}$ at iteration $t-1$, for each node $i$, we aggregate the geometric information from its neighborhood $\mathcal{N}_i$ into a new geometric object:

$$g_i^{(t)} \coloneqq \left((c_i^{(t-1)}, g_i^{(t-1)}), \{\!\!\{(c_j^{(t-1)}, g_j^{(t-1)}, \boldsymbol{x}_{ij}) \mid j \in \mathcal{N}_i\}\!\!\}\right). \tag{17}$$

Notice that the geometric objects are acted on by the group $\mathfrak{G}$:

$$\mathfrak{g} \cdot g_i^{(t)} := \left( (c_i^{(t-1)}, \mathfrak{g} \cdot g_i^{(t-1)}), \{\!\{ (c_j^{(t-1)}, \mathfrak{g} \cdot g_j^{(t-1)}, \mathfrak{g} \cdot \boldsymbol{x}_{ij}) \mid j \in \mathcal{N}_i \}\!\} \right). \tag{18}$$

One can check that the process of creating $g_i^{(t)}$ is injective and $\mathfrak{G}$-equivariant. We then assign the color $c_i^{(t)}$ at iteration $t$ through a $\mathfrak{G}$-invariant and $\mathfrak{G}$-orbit injective map, denoted as I-HASH$^{(t)}$,

$$c_i^{(t)} := \text{I-HASH}^{(t)}(g_i^{(t)}). \tag{19}$$

In other words, I-HASH$^{(t)}(g) = $ I-HASH$^{(t)}(g')$ if and only if $g = \mathfrak{g} \cdot g'$ for some $\mathfrak{g} \in \mathfrak{G}$.

The iteration terminates when the colors induce the same partitions of nodes. Then, given two attributed graphs $\mathcal{G}_1 = (\mathcal{V}_1, \mathcal{E}_1), \mathcal{G}_2 = (\mathcal{V}_2, \mathcal{E}_2)$, if there is some iteration $t$ s.t. $\{\!\{ c_i^{(t)} \mid i \in \mathcal{V}_1 \}\!\} \neq \{\!\{ c_j^{(t)} \mid j \in \mathcal{V}_2 \}\!\}$, then GWL determines that these two graphs are not geometrically isomorphic.

**Invariant GWL.** For the invariant version of GWL, we do not consider the equivariant geometric object. Thus, the iteration becomes:

$$c_i^{(t)} := \text{I-HASH} \left( c_i^{(t-1)}, \{\!\{ (c_j^{(t-1)}, \boldsymbol{x}_{ij}) \mid j \in \mathcal{N}_i \}\!\} \right), \tag{20}$$

where the initialization remains the same $c_i^{(0)} := \text{HASH}(\boldsymbol{f}_i)$.

$k$**-hop IGWL.** To extend the 1-hop aggregation in IGWL to $k$-hop aggregation, we replace the 1-hop neighborhood $\mathcal{N}_i$ with the $k$-hop neighborhood $\mathcal{N}_i^{(k)}$:

$$c_i^{(t)} := \text{I-HASH} \left( c_i^{(t-1)}, \{\!\{ (c_j^{(t-1)}, \boldsymbol{x}_{ij}) \mid j \in \mathcal{N}_i^{(k)} \}\!\} \right). \tag{21}$$

Now we extend the results in [20] for IGWL to $k$-hop IGWL:

**Proposition 1.** *$k$-hop IGWL can distinguish $k$-hop distinct geometric graphs with just one iteration, but it cannot differentiate $k$-hop identical geometric graphs no matter how many iterations are used.*

*Proof of Proposition 1.* Let $\mathcal{G}_1 = (\mathcal{V}_1, \mathcal{E}_1, \boldsymbol{F}^{\mathcal{G}_1}, \boldsymbol{X}^{\mathcal{G}_1}), \mathcal{G}_2 = (\mathcal{V}_2, \mathcal{E}_2, \boldsymbol{F}^{\mathcal{G}_2}, \boldsymbol{X}^{\mathcal{G}_2})$ be two $k$-hop identical geometric graphs. By definition, there exists a graph isomorphism $b$ such that for any node $i \in \mathcal{V}_1$, we can find a group element $\mathfrak{g}_i \in \mathfrak{G}$ satisfies $\boldsymbol{f}_i^{\mathcal{G}_1} = \boldsymbol{f}_{b(i)}^{\mathcal{G}_2}, \boldsymbol{x}_i^{\mathcal{G}_1} = \mathfrak{g}_i \cdot \boldsymbol{x}_{b(i)}^{\mathcal{G}_2}$ and $\boldsymbol{f}_j^{\mathcal{G}_1} = \boldsymbol{f}_{b(j)}^{\mathcal{G}_2}, \boldsymbol{x}_j^{\mathcal{G}_1} = \mathfrak{g}_i \cdot \boldsymbol{x}_{b(j)}^{\mathcal{G}_2}$ whenever $j \in \mathcal{N}_i^{(k)}$. This implies for any $i$, we have $c_i^{(0)} = c_{b(i)}^{(0)}$, and hence the multisets $\{\!\{ (c_j^{(0)}, \boldsymbol{x}_{ij}^{\mathcal{G}_1}) \mid j \in \mathcal{N}_i^{(k)} \}\!\}$ and $\{\!\{ (c_{j'}^{(0)}, \boldsymbol{x}_{b(i)j'}^{\mathcal{G}_2}) \mid j' \in \mathcal{N}_{b(i)}^{(k)} \}\!\}$ are identical up to group action. Based on the definition of $k$-hop IGWL iterations defined in 20, we can then conclude that $c_i^{(1)} = c_{b(i)}^{(1)}$ for any $i$. By induction, it follows that any number of $k$-hop IGWL iterations cannot distinguish $\mathcal{G}_1$ and $\mathcal{G}_2$.

Now, let $\mathcal{G}_1 = (\mathcal{V}_1, \mathcal{E}_1, \boldsymbol{F}^{\mathcal{G}_1}, \boldsymbol{X}^{\mathcal{G}_1}), \mathcal{G}_2 = (\mathcal{V}_2, \mathcal{E}_2, \boldsymbol{F}^{\mathcal{G}_2}, \boldsymbol{X}^{\mathcal{G}_2})$ be two $k$-hop distinct geometric graphs. By definition, for any graph isomorphism $b$, there is a node $i \in \mathcal{V}_1$ such that the corresponding $k$-hop subgraphs forming by $\mathcal{N}_i^{(k)} \cup \{i\}$ and $\mathcal{N}_{b(i)}^{(k)} \cup \{b(i)\}$ are distinct under the group action. This implies $c_i^{(1)} \neq c_{b(i)}^{(1)}$. Since $b$ is arbitrary, we conclude that $\{\!\{ c_i^{(1)} \mid i \in \mathcal{V}_1 \}\!\} \neq \{\!\{ c_j^{(1)} \mid j \in \mathcal{V}_2 \}\!\}$ and hence 1 iteration of $k$-hop IGWL is sufficient to distinguish $\mathcal{G}_1$ and $\mathcal{G}_2$. $\qquad \square$

We recap two key results regarding the expressive power of GWL from [20]:

**Proposition 2.** *GWL can distinguish any two $k$-hop distinct geometric graphs, and $k$ iteration is sufficient.*

**Proposition 3.** *Up to $k$ iteration, GWL cannot distinguish any two $k$-hop identical geometric graphs.*

Applying Proposition 2 and 3, along with Proposition 1, we derive the following theorem. Notably, the case for $k = 1$ corresponds to Theorem 8 in [20]:

**Theorem 6.** *GWL is strictly more powerful than $k$-hop IGWL for any $k$, while they have the same expressive power when applied to fully connected graphs.*

*Proof.* We have demonstrated that $k$-hop IGWL can distinguish $k$-hop distinct geometric graphs, which are distinguishable by GWL using $k$ iterations.

However, $k$-hop IGWL cannot distinguish $k$-hop identical geometric graphs. As illustrated in Figure 1, we provide an example where $k$-hop IGWL fails to distinguish such graphs, whereas GWL succeeds. $\square$

**Proposition 4.** *Any pair of geometric graphs is distinguished by $k$-hop invariant GNNs is also distinguished by $k$-hop IGWL.*

*Proof.* The proof is the same as the proof of Theorem 24 in [20] by replacing the 1-hop neighborhoods with $k$-hop neighborhoods. $\square$

**Proposition 5.** *$k$-hop invariant GNNs have the same expressive power as $k$-hop IGWL if the following conditions hold: (1) The aggregate function AGG and update function UPD are $\mathfrak{G}$-orbit injective and $\mathfrak{G}$-invariant multiset functions (2) The graph-level readout function is an injective multiset function.*

*Proof.* The proof is the same as the proof of Proposition 25 in [20] by replacing the 1-hop neighborhoods with $k$-hop neighborhoods. $\square$

**Significance of faithful representations in learning steerable features.** It has been demonstrated in [20] that the expressive power of equivariant GNNs, when learning steerable features up to type 1, is bounded by the GWL test, with equality under certain assumptions on the injectivity of UPD, AGG, and the graph-level readout function. In particular, they prove the following results:

**Proposition 6.** *Any pair of geometric graphs is distinguished by 1-hop equivariant GNNs is also distinguished by GWL.*

**Proposition 7.** *1-hop equivariant GNNs learning steerable features up to type 1 have the same expressive power as GWL if the following conditions hold: (1) The aggregate function AGG and update function UPD are $\mathfrak{G}$-orbit injective and $\mathfrak{G}$-equivariant multiset functions (2) The graph-level readout function is a $\mathfrak{G}$-orbit injective and $\mathfrak{G}$-invariant multiset function.*

**Remark 6.** *As we merge scalar and vector features into steerable features $\boldsymbol{f}_i$, certain conditions in Proposition 7 have been adjusted to accommodate our setup.*

In this work, we extend Proposition 7 to cover cases where equivariant GNNs learn steerable features on any representations. Notably, we highlight that achieving equality in this extended scenario necessitates an additional condition: the steerable features must lie on faithful representations.

**Proposition 8.** *Consider 1-hop equivariant GNNs learning features on steerable vector space $V$ where the aggregate function AGG learns features on steerable vector space $W$. Then these equivariant GNNs have the same expressive power as GWL if the following conditions hold: (1) The aggregate function AGG and update function UPD are $\mathfrak{G}$-orbit injective and $\mathfrak{G}$-equivariant multiset functions (2) The graph-level readout function is a $\mathfrak{G}$-orbit injective and $\mathfrak{G}$-invariant multiset function. (3) $V, W$ are faithful representations.*

*Proof.* We employ a similar strategy to the proof presented in Proposition 7 in [20] to establish our result. In this proof, we use $[\ldots]$ to denote the equivalence class generated by the actions of $\mathfrak{G}$. Then, any $\mathfrak{G}$-orbit injective function can be expressed as an injective function over the equivalence classes $[\ldots]$.

The GWL test updates the node color $c_i^{(t)}$ and geometric object $g_i^{(t)}$ as:

$$g_i^{(t)} = h_v \left( \left( c_i^{(t-1)}, g_i^{(t-1)} \right), \left\{\!\!\left\{ \left( c_j^{(t-1)}, g_j^{(t-1)}, \boldsymbol{x}_{ij} \right) \mid j \in \mathcal{N}_i \right\}\!\!\right\} \right), \quad c_i^{(t)} = h_s \left( \left[ g_i^{(t)} \right] \right),$$

where $h_s$ is a $\mathfrak{G}$-invariant and $\mathfrak{G}$-orbit injective map and $h_v$ is a $\mathfrak{G}$-equivariant and injective operation, such as expanding the geometric multiset by copying, as shown in equation (17).

Consider an equivariant GNN that satisfies the conditions outlined in the theorem statement. We will show by induction that at any iteration $t$, there always exist $\mathfrak{G}$-equivariant and injective functions $\varphi^{(t)}$

such that $\boldsymbol{f}_i^{(t)} = \varphi^{(t)}\left(g_i^{(t)}\right)$ for any $t$. Let $h$ denote the graph-level readout function. Since $h$ maps different multisets of node features to unique invariant features, $\boldsymbol{f}_i^{(t)} = \varphi^{(t)}\left(g_i^{(t)}\right)$ implies that there exists an injective function $\varphi_c^{(t)}$ such that $h\left(\{\!\{\boldsymbol{f}_i^{(t)} \mid i \in \mathcal{V}\}\!\}\right) = \varphi_c^{(t)}\left(\{\!\{c_i^{(t)} \mid i \in \mathcal{V}\}\!\}\right)$ where $\mathcal{V}$ denotes the set of nodes.

First, we observe that $\boldsymbol{f}_i^{(t)} = \varphi^{(t)}\left(g_i^{(t)}\right)$ holds for $t = 0$ because $g_i^{(0)} = c_i^{(0)} = \text{HASH}(\boldsymbol{f}_i)$ for all $i \in \mathcal{V}$. Now, suppose this holds for iteration $t$. At iteration $t + 1$, substituting $\boldsymbol{f}_i^{(t)}$ with $\varphi^{(t)}\left(g_i^{(t)}\right)$ implies that.

$$
\begin{aligned}
\boldsymbol{f}_i^{(t+1)} &= \text{UPD}\left(\boldsymbol{f}_i^{(t)}, \text{AGG}(\{\!\{\boldsymbol{f}_i^{(t)}, \boldsymbol{f}_j^{(t)}, \boldsymbol{x}_{ij} \mid j \in \mathcal{N}_i\}\!\})\right) \\
&= \text{UPD}\left(\varphi^{(t)}\left(g_i^{(t)}\right), \text{AGG}\left(\{\!\{\varphi^{(t)}\left(g_i^{(t)}\right), \varphi^{(t)}\left(g_j^{(t)}\right), \boldsymbol{x}_{ij} \mid j \in \mathcal{N}_i\}\!\}\right)\right).
\end{aligned}
\tag{22}
$$

Consider the function $\phi(c_i^{(t)}, g_i^{(t)}) := \varphi^{(t)}(g_i^{(t)})$. Suppose $\phi(c_i^{(t)}, g_i^{(t)}) = \phi(c_j^{(t)}, g_j^{(t)})$. The injectivity of $\varphi^{(t)}$ then implies that $g_i^{(t)} = g_j^{(t)}$. As $h_s$ is $\mathfrak{G}$-invariant and $\mathfrak{G}$-orbit injective, we deduce that $c_i^{(t)} = c_j^{(t)}$, and therefore, $\phi$ is injective. The $\mathfrak{G}$-equivariance of $\phi$ is inherited directly from $\varphi^{(t)}$. By substituting $\varphi^{(t)}(g_i^{(t)})$ with $\phi(c_i^{(t)}, g_i^{(t)})$, we obtain

$$
\boldsymbol{f}_i^{(t+1)} = \text{UPD}\left(\phi\left(c_i^{(t)}, g_i^{(t)}\right), \text{AGG}\left(\{\!\{\phi\left(c_i^{(t)}, g_i^{(t)}\right), \phi\left(c_j^{(t)}, g_j^{(t)}\right), \boldsymbol{x}_{ij} \mid j \in \mathcal{N}_i\}\!\}\right)\right). \tag{23}
$$

It remains to show that there exists an injective function $\phi'$ such that:

$$
\boldsymbol{f}_i^{(t+1)} = \phi'\left(\left(c_i^{(t)}, g_i^{(t)}\right), \{\!\{\left(c_j^{(t)}, g_j^{(t)}\right), \boldsymbol{x}_{ij} \mid j \in \mathcal{N}_i\}\!\}\right).
$$

Indeed, we can define $\varphi^{(t+1)} = \phi' \circ h_v^{-1}$ and then we have

$$
\begin{aligned}
\boldsymbol{f}_i^{(t+1)} &= \phi' \circ h_v^{-1} h_v\left(\left(c_i^{(t)}, g_i^{(t)}\right), \{\!\{\left(c_j^{(t)}, g_j^{(t)}\right), \boldsymbol{x}_{ij} \mid j \in \mathcal{N}_i\}\!\}\right) \\
&= \varphi^{(t+1)} \circ h_v\left(\left(c_i^{(t)}, g_i^{(t)}\right), \{\!\{\left(c_j^{(t)}, g_j^{(t)}\right), \boldsymbol{x}_{ij} \mid j \in \mathcal{N}_i\}\!\}\right) \\
&= \varphi^{(t+1)}(g_i^{(t+1)}).
\end{aligned}
\tag{24}
$$

Now consider the following construction of $\phi'$,

$$
\begin{aligned}
&\phi'\left(\left(c_i^{(t)}, g_i^{(t)}\right), \{\!\{\left(c_j^{(t)}, g_j^{(t)}\right), \boldsymbol{x}_{ij} \mid j \in \mathcal{N}_i\}\!\}\right) \\
&:= \text{UPD}\left(\phi\left(c_i^{(t)}, g_i^{(t)}\right), \text{AGG}\left(\{\!\{\phi\left(c_i^{(t)}, g_i^{(t)}\right), \phi\left(c_j^{(t)}, g_j^{(t)}\right), \boldsymbol{x}_{ij} \mid j \in \mathcal{N}_i\}\!\}\right)\right)
\end{aligned}
$$

To demonstrate $\phi'$ is injective, we will first show that UPD and AGG are injective. **This step requires the faithfulness of $V$ and $W$.** More precisely, the $\mathfrak{G}$-orbit injectivity and $\mathfrak{G}$-equivariance of UPD and AGG with the faithfulness of $V$ and $W$ imply that they are injective. Suppose $\text{UPD}(\boldsymbol{f}_i, \boldsymbol{m}_i) = \text{UPD}(\boldsymbol{f}_j, \boldsymbol{m}_j)$ where $\boldsymbol{m}_i$ denotes the output of AGG. Due to the $\mathfrak{G}$-orbit injectivity, we have $[\boldsymbol{f}_i, \boldsymbol{m}_i] = [\boldsymbol{f}_j, \boldsymbol{m}_j]$. Therefore, $(\boldsymbol{f}_i, \boldsymbol{m}_i) = \mathfrak{g} \cdot (\boldsymbol{f}_j, \boldsymbol{m}_j) = (\mathfrak{g} \cdot \boldsymbol{f}_j, \mathfrak{g} \cdot \boldsymbol{m}_j)$ for some $\mathfrak{g} \in \mathfrak{G}$. Applying the $\mathfrak{G}$-equivariance, we obtain

$$
\begin{aligned}
\text{UPD}(\boldsymbol{f}_j, \boldsymbol{m}_j) &= \text{UPD}(\boldsymbol{f}_i, \boldsymbol{m}_i) \tag{25} \\
&= \text{UPD}(\mathfrak{g} \cdot \boldsymbol{f}_j, \mathfrak{g} \cdot \boldsymbol{m}_j) \tag{26} \\
&= \mathfrak{g} \cdot \text{UPD}(\boldsymbol{f}_j, \boldsymbol{m}_j) \tag{27}
\end{aligned}
$$

Faithfulness implies $\mathfrak{g}$ is the identity, proving the injectivity of UPD. Similarly, we can prove the injectivity of AGG.

Now applying the injectivity of UPD and AGG, we see that:

$$
\phi\left(c_i^{(t)}, g_i^{(t)}\right) = \phi\left(c_j^{(t)}, g_j^{(t)}\right), \tag{28}
$$

$$
\{\!\{\phi\left(c_i^{(t)}, g_i^{(t)}\right), \phi\left(c_k^{(t)}, g_k^{(t)}\right), \boldsymbol{x}_{ik} \mid k \in \mathcal{N}_i\}\!\} = \{\!\{\phi\left(c_j^{(t)}, g_j^{(t)}\right), \phi\left(c_k^{(t)}, g_k^{(t)}\right), \boldsymbol{x}_{jk} \mid k \in \mathcal{N}_j\}\!\}. \tag{29}
$$

The injectivity of $\phi$ then implies that

$$\left(c_i^{(t)}, g_i^{(t)}\right) = \left(c_j^{(t)}, g_j^{(t)}\right), \tag{30}$$

$$\left\{\!\!\left\{\left(c_k^{(t)}, g_k^{(t)}\right), \boldsymbol{x}_{ik} \mid k \in \mathcal{N}_i\right\}\!\!\right\} = \left\{\!\!\left\{\left(c_k^{(t)}, g_k^{(t)}\right), \boldsymbol{x}_{jk} \mid k \in \mathcal{N}_j\right\}\!\!\right\}. \tag{31}$$

This shows the injectivity of $\phi'$ and thus completes the proof. $\qquad\square$

**Theorem 2.** *Consider* 1*-hop equivariant GNNs learning features on steerable vector space $V$ where the aggregate function* AGG *learns features on steerable vector space $W$. Suppose $V$ and $W$ are faithful representations, and* UPD *and* AGG *satisfy certain assumptions on the injectivity outlined in Proposition 8 in the appendix. Then with $k$ iterations, these equivariant GNNs learn different multisets of node features $\{\!\!\{\boldsymbol{f}_i^{(k)}\}\!\!\}$ on two $k$-hop distinct geometric graphs.*

*Proof.* This can be derived from the proof of Proposition 8 and 2. $\qquad\square$

# E AN OVERVIEW OF EXISTING INVARIANT AND EQUIVARIANT GNNS

In this section, we discuss the invariant and equivariant architectures considered in the work. However our framework covers models beyond those mentioned here, for instance it includes the models discussed in Joshi et al. [20]. It's worth noting that this framework is not confined to using 2-body aggregation; it can also employ multi-body aggregation methods.

## E.1 $\mathfrak{G}$-INVARIANT GNNS

**SchNet** The SchNet model [34] is a 2-body $\mathfrak{G}$-invariant architecture which propagates type-0 features in 1-hop using relative distances:

$$\boldsymbol{f}_i^{(t+1)} = \mathrm{UPD}\left(\boldsymbol{f}_i^{(t)}, \mathrm{AGG}(\{\!\!\{\boldsymbol{f}_i^{(t)}, \boldsymbol{f}_j^{(t)}, \|\boldsymbol{x}_{ij}\| \mid j \in \mathcal{N}_i^{(1)}\}\!\!\})\right)$$

**DimeNet++** The DimeNet++ model [14] is a 3-body $\mathfrak{G}$-invariant architecture which propagates type-0 features in 1-hop using relative distances and angles.

$$\boldsymbol{f}_i^{(t+1)} = \mathrm{UPD}\left(\boldsymbol{f}_i^{(t)}, \mathrm{AGG}(\{\!\!\{\boldsymbol{f}_i^{(t)}, \boldsymbol{f}_j^{(t)}, \|\boldsymbol{x}_{ij}\|, \boldsymbol{x}_{ij} \cdot \boldsymbol{x}_{ik} \mid j, k \in \mathcal{N}_i^{(1)}, k \neq j\}\!\!\})\right)$$

**ComENet** The ComENet model [40] is a 4-body $\mathfrak{G}$-invariant architecture which propagates type-0 features in 1-hop with complete edge attributes ($\|\boldsymbol{x}_{ij}\|, \theta_{ij}, \phi_{ij}, \tau_{ij}$):

$$\boldsymbol{f}_i^{(t+1)} = \mathrm{UPD}\left(\boldsymbol{f}_i^{(t)}, \mathrm{AGG}(\{\!\!\{\boldsymbol{f}_j^{(t)}, \|\boldsymbol{x}_{ij}\|, \theta_{ij}, \phi_{ij}, \tau_{ij} \mid j \in \mathcal{N}_i^{(1)}\}\!\!\})\right),$$

where $\theta_{ij}$, $\phi_{ij}$, and $\tau_{ij}$ are computed in quadruplet (within a 2-hop neighborhood).

## E.2 $\mathfrak{G}$-EQUIVARIANT GNNS

**EGNN and GVP** The EGNN [32] and GVP [19] models are a 3-body $\mathfrak{G}$-equivariant architecture which propagate type-0 and type-1 features in 1-hop using relative distances.

$$\boldsymbol{f}_i^{(t+1)} = \mathrm{UPD}\left(\boldsymbol{f}_i^{(t)}, \mathrm{AGG}(\{\!\!\{\boldsymbol{f}_i^{(t)}, \boldsymbol{f}_j^{(t)}, \|\boldsymbol{x}_{ij}\| \mid j \in \mathcal{N}_i^{(1)}\}\!\!\})\right)$$

**MACE and eSCN** The MACE [2] and eSCN [28] models are 2-body $\mathfrak{G}$-equivariant architecture which propagate up to type-$L$ features in 1-hop using spherical harmonics.

$$\boldsymbol{f}_i^{(t+1)} = \mathrm{UPD}\left(\boldsymbol{f}_i^{(t)}, \mathrm{AGG}(\{\!\!\{\boldsymbol{f}_i^{(t)}, \boldsymbol{f}_j^{(t)}, Y(\hat{\boldsymbol{x}}_{ij}), \|\boldsymbol{x}_{ij}\| \mid j \in \mathcal{N}_i^{(1)}\}\!\!\})\right)$$

**EquiformerV2** The EquiformerV2 model [24] is a $\mathfrak{G}$-equivariant architecture, similar to the one mentioned above. It propagates up to type-$L$ features in 1-hop by fully utilizing all neighbors within the neighborhood to create the attention weights.

# F ADDITIONAL EXPERIMENTAL DETAILS

## F.1 ADDITIONAL RESULTS FOR $k$-CHAINS

| Layers | 2 | 3 | 4 | 5 | 6 |
|---|---|---|---|---|---|
| | | | $L = 0$ | | |
| SchNet | $50.0 \pm 0.0$ | $50.0 \pm 0.0$ | $50.0 \pm 0.0$ | $50.0 \pm 0.0$ | $50.0 \pm 0.0$ |
| DimeNet++ | $50.0 \pm 0.0$ | $50.0 \pm 0.0$ | $50.0 \pm 0.0$ | $50.0 \pm 0.0$ | $50.0 \pm 0.0$ |
| SphereNet | $50.0 \pm 0.0$ | $50.0 \pm 0.0$ | $50.0 \pm 0.0$ | $50.0 \pm 0.0$ | $50.0 \pm 0.0$ |
| ComENet | $50.0 \pm 0.0$ | $50.0 \pm 4.5$ | $53.0 \pm 5.1$ | $49.5 \pm 1.5$ | $49.5 \pm 2.7$ |
| EquiformerV2 | $55.0 \pm 8.1$ | $61.0 \pm 12.2$ | $68.0 \pm 8.7$ | $80.0 \pm 7.7$ | $75.0 \pm 11.2$ |
| | | | $L = 1$ | | |
| EGNN | $50.0 \pm 0.0$ | $50.0 \pm 0.0$ | $95.0 \pm 15.0$ | $100.0 \pm 0.0$ | $90.0 \pm 20.0$ |
| GVP | $50.0 \pm 0.0$ | $93.0 \pm 15.5$ | $90.5 \pm 19.0$ | $99.0 \pm 2.0$ | $100.0 \pm 0.0$ |
| ClofNet | $50.0 \pm 0.0$ | $100.0 \pm 0.0$ | $95.0 \pm 15.0$ | $95.0 \pm 15.0$ | $95.0 \pm 15.0$ |
| LEFTNet | $50.0 \pm 0.0$ | $50.0 \pm 0.0$ | $50.0 \pm 0.0$ | $50.0 \pm 0.0$ | $50.0 \pm 0.0$ |
| MACE | $50.0 \pm 0.0$ | $100.0 \pm 0.0$ | $100.0 \pm 0.0$ | $100.0 \pm 0.0$ | $100.0 \pm 0.0$ |
| eSCN | $57.0 \pm 7.8$ | $58.0 \pm 9.8$ | $62.0 \pm 9.8$ | $52.0 \pm 4.0$ | $53.0 \pm 6.4$ |
| EquiformerV2 | $53.0 \pm 10.0$ | $52.0 \pm 7.5$ | $100.0 \pm 0.0$ | $100.0 \pm 0.0$ | $100.0 \pm 0.0$ |
| | | | $L = 2$ | | |
| MACE | $50.0 \pm 0.0$ | $100.0 \pm 0.0$ | $100.0 \pm 0.0$ | $100.0 \pm 0.0$ | $100.0 \pm 0.0$ |
| eSCN | $57.0 \pm 10.0$ | $54.0 \pm 8.0$ | $60.0 \pm 10.0$ | $55.0 \pm 6.7$ | $60.0 \pm 7.7$ |
| EquiformerV2 | $53.0 \pm 10.0$ | $46.0 \pm 9.2$ | $45.0 \pm 6.7$ | $75.0 \pm 6.7$ | $77.0 \pm 7.8$ |

Table 5: Test accuracy for the $k$-hop chain dataset with $k = 4$. Cell shading is based on two standard deviations above or below the expected value. Unit:%.

Table 5 reports the 10-fold cross validation mean test accuracy and standard deviation for chain length $k = 4$ and varying model depth. The values are shaded based on the expected value from Section 3 . We remark that EGNN, ClofNet, LEFTNet [10], eSCN and EquiformerV2 do not perform as expected on this particular task. As discussed in [20] the under performance of EGNN and ClofNet is likely due to oversquashing [38; 1]. For LEFTNet, the underlying cause is attributed to its invariant design. Consequently, this architecture can be categorized as a 2-hop invariant GNN, leading to expected challenges in capturing changes in global geometry. The consistent underperformance of eSCN may be due to the task being O(3) equivariant but not SO(3) equivariant. Additionally, eSCN and EquiformerV2 are quasi-equivariant methods which introduce error into the equivariance due to their spherical activation function, see Appendix D of [28] for details. This additional error explains the significant difference in results for eSCN and EquiformerV2 as well as why they may over perform as well as under perform.

## F.2 ADDITIONAL RESULTS FOR IS2RE

| Model | $L$ | $c$ | Feat. Dim. | # Param. | Loss ↓ | Energy MAE [meV] ↓ | EwT [%] ↑ |
|---|---|---|---|---|---|---|---|
| eSCN | 1 | 464 | 1856 | 11M | $0.380 \pm 0.006$ | $865 \pm 14$ | $1.91 \pm 0.09$ |
| eSCN | 2 | 206 | 1854 | 10M | $\mathbf{0.369 \pm 0.006}$ | $\mathbf{842 \pm 13}$ | $\mathbf{1.94 \pm 0.12}$ |
| eSCN | 3 | 133 | 1862 | 9M | $0.397 \pm 0.001$ | $904 \pm 3$ | $1.85 \pm .12$ |
| eSCN | 4 | 98 | 1862 | 9M | $0.408 \pm 0.006$ | $929 \pm 15$ | $1.74 \pm 0.12$ |
| eSCN | 5 | 77 | 1848 | 8M | $0.409 \pm 0.003$ | $933 \pm 7$ | $1.61 \pm .12$ |
| eSCN | 6 | 64 | 1856 | 8M | $0.3836 \pm 0.003$ | $872 \pm 6$ | $1.91 \pm 0.19$ |
| EquiformerV2 | 1 | 77 | 304 | 7M | OOM | OOM | OOM |
| EquiformerV2 | 2 | 34 | 306 | 9M | $0.369 \pm 0.009$ | $841 \pm 21$ | $2.02 \pm 0.14$ |
| EquiformerV2 | 3 | 22 | 306 | 12M | $\mathbf{0.363 \pm 0.009}$ | $\mathbf{828 \pm 21}$ | $1.94 \pm 0.08$ |
| EquiformerV2 | 4 | 16 | 304 | 15M | $0.364 \pm 0.005$ | $832 \pm 11$ | $\mathbf{2.03 \pm 0.14}$ |

Table 6: Validation results of the steerable model ablation study on $L$ and $c$ over 4-folds of the IS2RE dataset with 10k training molecules. We observe that higher type-$L$ steerable models may not perform best. OOM denotes models that run out of memory during training.

Note that for EquiformerV2 L=1, the model will fit onto the GPU. However, during training the data and the model will exceed the GPU memory capabilities. We denote the out of memory phenomenon OOM in the Table 6 and Table 7 and report the model parameters and size.

| Model | $L$ | $c$ | Run Time (min) | Memory |
|---|---|---|---|---|
| eSCN | 1 | 464 | $151 \pm 1$ | 12.3GB |
| eSCN | 2 | 206 | $297 \pm 5$ | 9.1GB |
| eSCN | 3 | 133 | $207 \pm 3$ | 9.0GB |
| eSCN | 4 | 98 | $347 \pm 4$ | 9.2GB |
| eSCN | 5 | 77 | $246 \pm 3$ | 10.9GB |
| eSCN | 6 | 64 | $429 \pm 8$ | 11.5GB |
| EquiformerV2 | 1 | 77 | OOM | 18.6GB |
| EquiformerV2 | 2 | 34 | $284 \pm 6$ | 12.6GB |
| EquiformerV2 | 3 | 22 | $256 \pm 1$ | 11.1GB |
| EquiformerV2 | 4 | 16 | $298 \pm 2$ | 11.5GB |

Table 7: Run time and memory footprint for the steerable model ablation study on $L$ and $c$ over 4-folds of the IS2RE dataset with 10k training molecules. OOM denotes models which run out of memory during training.

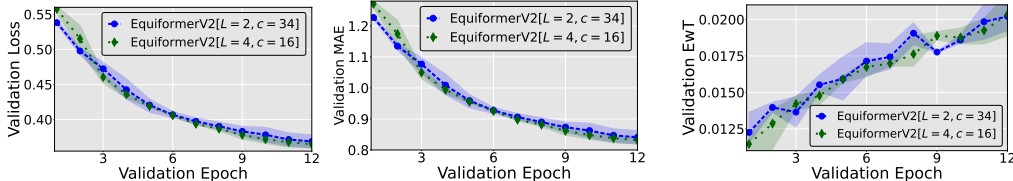

Figure 3: Validation results for the ablation study on EquiformerV2 for the IR2SE dataset with 10k training molecules. Depicted is the loss (left) energy MAE (middle) and energy within the threshold(EwT) (right) over 12 validation epochs. We plot the mean over four runs and shade the standard deviation. We observe no difference between the values for different type-$L$ models.

## F.3 EXPERIMENTAL DETAILS

$k$-**Chain** The reported optimal hyperparameters are used except in the case of steerable models eSCN and EquiformerV2 where the steerable feature dimensions are reduced due to memory constraints. Steerable models using type $L = 0, 1$, and 2 are used with adjusted steerable feature dimension to preserve the dimension of the steerable convolution. The experiment is implemented in Google Colab [26] with invariant and equivariant models using the 16GB NVIDIA T4 GPUs and the steerable models using the 40GB NVIDIA A100 GPUs. To ensure the connectivity of the graph is respected, we adjust the cutoff radius of all models to $5.1$ units.

The training procedure of [20] is modified to apply a softmax function to model outputs, stabilizing the classification training. The cross entropy is minimized over $1000$ training epochs using the Adam optimizer. The learning rate is scheduled based on the validation accuracy with an initial learning rate of $1e$-4, 0.9 learning rate decay, 25 epochs of patience, and a minimum learning rate of $1e$-5. We use fixed data splits and report the mean and standard deviation of the test accuracy from 10-fold cross-validation of random model weight initializations.

**IS2RE** The task is implemented using the OC20 framework using the baseline IS2RE training procedure with 12 training epochs for all models and the reported optimal model hyperparameters. Due to computational constraints, we consider the 10k molecule training data split and reduce the steerable feature dimension to train each model on a single 24GB NVIDIA RTX3090 GPU.

The training procedure of [6] is used for training. In particular we use the AdamW optimizer with a learning rate of $8e$-4, a maximum epoch of 12 and gradient clipping if the gradient norm is greater than 20.

**S2EF** The task is implemented in the OC20 framework using the baseline S2EF training procedure and the reported optimal hyperparameters for each model. Training is performed in parallel on two 24GB NVIDIA RTX4090 GPUs using the 2M molecule training data split. Due to the limited computational resources, we are only able to train a single model for 8 epochs over the span of 8 weeks. We note that we compare our results to [28] which trains 4-fold cross validation on 16GPUs with 32GB of RAM and does not report the standard deviation.

The training procedure of [6] is used. In particular we use the AdamW optimizer with a weighted decay of $1e$-3, a cosine learning rate scheduler and an initial learning rate of $4e$-4. The maximum number of epochs are 12 and gradient clipping is applied if the gradient norm greater than 100.

## F.4 Hyperparameter Details

In this section we provide details on the adjusted hyperparameters for each task. In general we implement the reported optimal hyperparameters for each model.

### F.4.1 $k$-Chain

For all models the cutoff hyperparameter (cutoff, max_radius, r_max) is set to $5.1$. Table 8 lists the hyperparameters for ComENet, eSCN and EquiformerV2. For all other models, we use the hyperparameters outlined in Joshi et al. [20].

| ComeENet | Value | eSCN | Value | EquiformerV2 | Value |
|---|---|---|---|---|---|
| cutoff | 5.0 | cutoff | 5.1 | max_radius | 5.1 |
| hidden_channels | 256 | hidden_channels | [256,113] | attn_hidden_channels | [64,16,7] |
| middle_channels | 64 | lmax_list | [1,2] | lmax_list | [0,1,2] |
| num_radial | 3 | mmax_list | [1,2] | mmax_list | [0,1,2] |
| num_spherical | 2 | regress_forces | False | regress_forces | False |
| num_output_layers | 3 | | | | |

Table 8: Hyperparameters for the ComeENet, eSCN and EquiformerV2 architectures trained on $k$-Chain dataset. In brackets we provide the values for each type-$L$ model.

### F.4.2 IS2RE

| eSCN | Value | EquiformerV2 | Value |
|---|---|---|---|
| num_layers | 12 | num_layers | 8 |
| max_neighbors | 20 | sphere_channels | 128 |
| cutoff | 12.0 | attn_hidden_channels | [34/16,16] |
| sphere_channels | 128 | num_heads | 8 |
| hidden_channels | [206/64,98/64, 64] | attn_alpha_channels | 64 |
| lmax_list | [2,4,6] | lmax_list | [2,4] |
| mmax_list | [2,2,2] | mmax_list | [2,2] |
| num_sphere_samples | 128 | norm_type | 'layer_norm_sh' |
| distance_function | "gaussian" | otf_graph | True |
| regress_forces | False | regress_forces | False |
| use_pbc | True | attn_value_channels | 16 |
| basis_width_scalar | 2.0 | ffn_hidden_channels | 32 |
| otf_graph | True | | |

Table 9: Hyperparameters for the eSCN and EquiformerV2 architectures trained on IS2RE with 10k training molecules. In brackets, we provide the values for the ablation study on the steerable feature dimension.

F.4.3 S2EF

| eSCN | Value |
|---|---|
| num_layers | 12 |
| max_neighbors | 20 |
| cutoff | 12.0 |
| sphere_channels | 128 |
| hidden_channels | [256,824] |
| lmax_list | [6,2] |
| mmax_list | 2 |
| num_sphere_samples | 128 |
| distance_function | "gaussian" |
| regress_forces | True |
| use_pbc | True |
| basis_width_scalar | 2.0 |
| otf_graph | True |

Table 10: Hyperparameters for the eSCN architectures trained on S2EF with 2M training molecules. In brackets we provide the values for eSCN[$L = 6, c = 256$] and eSCN[$L = 2, c = 824$] resp.

## G AN ILLUSTRATIVE EXAMPLE

While Lemma 1 is not affected by $\mathrm{rank}(\boldsymbol{X})$, the rank of the spatial embedding $\boldsymbol{X}$, the answer to this question is contingent upon the rank. In Fig. 4, we present an example to gain intuition into this phenomenon. The graph comprises two distinct points $\boldsymbol{x}_1, \boldsymbol{x}_2$. We assert that all type-1 features must lie on the line passing through $\boldsymbol{x}_1$ and $\boldsymbol{x}_2$, which corresponds to a rank-1 space. Suppose there exists a type-1 steerable feature $\boldsymbol{v} \neq 0$ that does not lie on this line. In such a case, we can select a rotation $\boldsymbol{R}$ around the line, which preserves $\boldsymbol{x}_1$ and $\boldsymbol{x}_2$ but changes the direction of $\boldsymbol{v}$ to $\boldsymbol{R}\boldsymbol{v} \neq \boldsymbol{v}$. Consequently, $\boldsymbol{v}$ cannot be generated equivariantly from $\boldsymbol{x}_1$ and $\boldsymbol{x}_2$, leading to a contradiction.

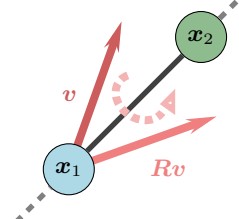

Figure 4: If $\boldsymbol{v}$ doesn't lie on the line passing through $\boldsymbol{x}_1$ and $\boldsymbol{x}_2$, it can be transformed into another vector by some rotation $\boldsymbol{R}$ around this line. Therefore, $\boldsymbol{v}$ cannot be generated equivariantly from $\boldsymbol{x}_1$ and $\boldsymbol{x}_2$.

The rationale behind this is that different ranks of $\boldsymbol{X}$ correspond to distinct underlying symmetries inherent to $\boldsymbol{X}$. Specifically, when $\mathrm{rank}(\boldsymbol{X}) = 3$, there exists no non-trivial group action capable of preserving the spatial embedding $\boldsymbol{X}$, meaning that the stabilizer $\mathfrak{G}_{\boldsymbol{X}}$ is empty. In contrast, in cases where $\mathrm{rank}(\boldsymbol{X}) = 1$ or $2$ – the coordinates are confined to a line or a plane – certain group actions can be applied without affecting $\boldsymbol{X}$. That is, $\exists \mathfrak{g} \in \mathfrak{G}$ s.t. $\mathfrak{g} \cdot \boldsymbol{X} = \boldsymbol{X}$. Since $\mathfrak{g} \cdot f(\boldsymbol{X}) = f(\mathfrak{g} \cdot \boldsymbol{X}) = f(\boldsymbol{X})$, we observe that any steerable feature $f(\boldsymbol{X})$ must be preserved under the action of $\mathfrak{g}$. This limitation inherently constrains the complexity of steerable features.

