# OpenReview forum: "Rethinking the Benefits of Steerable Features in 3D Equivariant Graph Neural Networks"
_ICLR.cc/2024/Conference — ICLR 2024 poster_

### Official Review · Reviewer_3HqL · 2023-10-25

**Soundness:** 2 fair
**Presentation:** 3 good
**Contribution:** 2 fair
**Rating:** 6
**Confidence:** 3

**Summary:**

This paper compares the expressive power of invariant and equivariant GNNs, from a theoretical perspective, backed by a couple of experiments (1 toy experiment and 2 real datasets).
It is an extension of what was recently done in [16] to the case of k-hop GNNs.
The main theoretical contribution is Lemma 1, which states that there is a unique invariant feature corresponding to any equiavariant one. It is made more concrete by theorem 2 and corrolary 1.
The experiments investigate the interplay between type order L (order of the Spherical Harmonics expansion, e.g. L=1 is vectors, L=0 scalars, L>1 are high order tensors), depth, and channel number, in the context of fixed feature dimension. This allows the authors to formulate the cautious claim "preserving the feature dimension, the expressiveness of equivariant GNNs employing steerable features up to type-L may not increase as L grows."

I update my rating to 6, marginally above acceptance threshold.

**Strengths:**

The idea of studying equivariant GNNs in the context of k-hop aggregation is new.
The formalism the authors use to formulate proofs is general and does not rely on Spherical Harmonics explictly (this is also a weakness though).
The idea of working at fixed budget to compare models, albeit probably not new, is to be saluted.

**Weaknesses:**

See my questions below for details. The main Weaknesses are:

Lack of novelty compared to what was done in [16]. See my points 2d., 4., 7a., 8.

Overall clarity: although wording is clear, some definitions are not given (as k-hop, see questions below) and the fact that theorems prove existence but do not provide an example of such function (like lambda, in some concrete example) make it hard to follow. See questions.

Also, it is not clear whether all claims are correct (or, some of them may be correct but trivial if properly re-phrased -- but I may be wrong !).

I may be misunderstood in my assessment, hence my numerous questions.

**Questions:**

1. In the definition of k-hop geometric GNNs, k is not defined (its role is not defined).
Actually, the definition of k-hop GNN is a bit overlooked (although it's rather intuitive what the meaning of k-hop is), and more importantly, it is not clear to me how these are not essentially equivalent to regular GNNs with more layers (since a single layer of a 2-hop GNN aggregates information with a similar receptive field as a 1-hop GNN with 2 layers).
Probably authors should elaborate on that or cite appropriate references.

2a. About Lemma 1, page 4. When you define the function c, why can't it simply be always the identity? One could always choose the neutral element from the group G.
I do understand why the set {g ∈ G | g · c(G · X) = X} is not empty.
I do not understand what is non trivial about c(G · X)

2b. Also, V_0^{⊕d} is not explicitly defined early enough. What is d ? Does it relate to the maximal rotation order L ?

2c. You proove that the decomposition of Lemma 1 is unique. But, concretely, what is lambda ? Isn't it almost always simply the norm of the irrep ? Like, for a vector, its norm, or for a higher-order tensor, also its norm ? And then \rho(g_X) is just the usual (matrix) representation of g_X.
Can it be otherwise ? Or is it more complicated ? Am I missing the point ?
If in practice things are simple, in most of the usual settings, saying so is helpful to the reader.
OR, maybe g_X simply encodes the orientation of X (say if it's a vector)?


2d. In ref [16], page 3, it says:
|At initial-
|isation, we assign to each node i ∈ V a scalar node colour
|c i ∈ C ′ and an auxiliary object g i containing the geometric
|information associated to it
|i.e. they already separate (factorize) the scalar (feature only) contributions from the geometric ones (vectorial one v + previous color).
In this respect it is not obvious how contribution (1) (your lemma 1) is new.
Furthermore, I lack an intuitive view of what may concretely go into your scalar function lambda(X).


3. gothic{g} appears in the r.h.s. of Eq (4) or in the last line of the previous est of equations (let me call this last line Eq. 3bis) (actually the gothic-style has been forgotten in Eq. (4), this is a typo I believe).
How can we identify the right term in the r.h.s. of Eq. 3bis as a lambda (invariant), when g^-1 appears in it ? I don't see why g^-1\cdot x_ij should be invariant, on the contrary, it seems to me like an equivariant feature.


In Eq. 7, first line, why isn't it 2l+2 instead of 2l+1 (in the r.h.s.) ? I understood that representing O(3) required one additional component ? Could you provide an intuitive explanation?

4. You write, in page 6:
|Does any (2l + 1)-dimensional invariant feature λ(X) correspond to a type-l steerable feature f (X)?
Which sounds like an interesting question (although ideally I'd like to know the mapping, not just know about its existence)
|Note that this question is essentially asking whether the space of all type-l steerable features f (X) has a dimension of 2l + 1 since D l (g X ) is invertible.
But that sounds like a known fact: using Spherical Harmonics, it is known that  type-l steerable features  need components $h_m$ with  $m\in[-l,+l]$ to be represented. That is, they need 2l+1 numbers (dimensions)

5. The remark below corrolary 3 is a very strong, original conclusion.
I think I understand corrolary 3 (altohough I feel like in practice lambda is the norm of the representation, and in that case it's kind of trivial..).
In any case I do not see how the remark  "the expressiveness of learning steerable features is primarily characterized by the feature dimension – independent of the highest type utilized"  follows from corrolary 3.


6. In table I, I can do the maths for line 1 and 2, but not line 3.
(5+3+1)*256 = 2304
(5+3+1)*824 = 7416
but then,
(1+3+5+7+9+11+13)*256 =12544 , not 7424


7. experiments are a bit disappointing.
7a. First, Table 2 shows a number of red (intersting) cases, but they turn out to be related to eSCN being SO(3)-eqivariant, when the task is O(3). This is not making the point of the paper, and instead is rather confusing, at first sight.
Most importantly, it's not clear to me in which sense table 2 is different from the table 1 of ref [16] (which by the way seemed more readable and dense).
Please clarify this.

7b. IS2RE. Table 3.
Here I enjoyed a lot the idea of comparing models with comparable feature dimension.
However, several points:
- why not report also the total number of parameters ? They do not grow as fast as feature dimension since weights are shared for a given L (type order)
- although I understand it's somewhat hardware dependent, please also report the training time for these models. Maybe even the memory footprint (on GPU).
- Figure 3 gives an idea of how things perform over a reasonable number of epochs (and I salute the attempt to not go to an exceedingly large number of epochs), but it seems that a more thorough study of variations with L and c, reporting only the epoch=12 validation performances, would be useful to the reader (I did not have time to look at the content of ref [5])

7c. S2EF. Table 4 is commented, but no conclusion stands out. What is the point of that table ?


8. Conclusion. Two key findings are claimed:
| (1) Propagating steerable features of type ≥ 1
| equips geometric GNNs with the inherent ability to automatically capture the geometry between local
| structures. (2) When the feature dimension remains constant, increasing the use of steerable features
| up to type-L cannot essentially improve the performance of equivariant GNNs
But I think that  (1) is vague and already known. For instance from (16), which clearly distinguishes invariant (L=0) from equivariant L>=1
networks.
And that (2) is known as oversquashing and was known before.
I note that the term oversquashing is absent from the manuscript, although it is mentionned several times in [16].
I think the term should appear. If it sounds too buzzwordy to the authors, they should mention it at least once, and explain why they don't like it.

---

> ### Author Response · Authors · 2023-11-16
> **Response to Reviewer 3HqL (1/6)**
>
> Thank you for your thoughtful review and valuable feedback. We fear the reviewer has some misunderstandings of our work which we have worked to clarify in this rebuttal and in our revision.
>
> Regarding your concern about the novelty, we would like to clarify there are some misunderstandings. In particular, our results cover $k$-hop geometric GNN, such as ComENet, which is mentioned in the limitation section in Joshi et al.’s work. Moreover, we investigate whether, when the feature dimension remains constant, the performance of equivariant GNNs employing steerable features up to type-$L$ increases as $L$ grows.
>
> In what follows, we provide point-by-point responses to your comments.
>
> ---
>
> **Q1. In the definition of $k$-hop geometric GNNs, $k$ is not defined (its role is not defined). Actually, the definition of $k$-hop GNN is a bit overlooked (although it's rather intuitive what the meaning of $k$-hop is), and more importantly, it is not clear to me how these are not essentially equivalent to regular GNNs with more layers (since a single layer of a 2-hop GNN aggregates information with a similar receptive field as a 1-hop GNN with 2 layers). Probably authors should elaborate on that or cite appropriate references.**
>
> **Reply:** We present the architecture of $k$-hop geometric GNNs in Eq (1), where $k$ denotes the size of neighborhoods from which a node aggregates information -- or equivalently, the maximum distance over which it aggregates information in a single message-passing step. Additionally, we reference ComENet as an instance of 2-hop invariant GNNs in Section 1.1.
>
> Moreover, it is crucial to note that $k$-hop geometric GNNs are not necessarily equivalent to regular 1-hop geometric GNNs with $k$ layers. As proved in Appendix D of the revised version, under some assumptions on UPD and AGG, k-hop invariant GNNs can distinguish any two $k$-hop distinct geometric graphs but fail to distinguish any two $k$-hop identical geometric graphs. ***This indicates that 1-hop invariant GNNs might be less powerful than their 2-hop counterparts, and this trend continues.***
>
> The reason lies in the fact that the value of $k$ influences the size of local neighborhoods, consequently affecting the input of the aggregation process. Utilizing a smaller $k$ may cause $k$-hop invariant GNNs to overlook changes in geometry between small local neighborhoods, whereas using a larger $k$ can capture these variations.
>
> However, it's noteworthy that, for any finite $k$, we can always construct two $k$-hop identical but $(k+1)$-hop distinct geometric graphs. In this scenario, $k$-hop invariant GNNs cannot differentiate them even with thousands of layers, while 1-hop equivariant GNNs, due to the faithfulness of features, can achieve this with $k+1$ layers. This finding sheds light on the advantages of learning steerable features up to type-$L\geq1$. We have substantiated this point through empirical results for ComENet. In the revision, we have bolstered these observations by considering the LEFTNet model in Appendix F.1 Table 5 of the revision.
>
> To clarify this implication, we have refined the concluding remarks. In particular, we have added the following remark: relying solely on propagating invariant features confined to specific k-hop neighborhoods is insufficient for enabling geometric GNNs to capture the global information of geometric graphs precisely.
>
>
> ---
>
> **Q2. About Lemma 1, page 4. When you define the function $c$, why can't it simply be always the identity? One could always choose the neutral element from the group $G$. I do understand why the set {$g\in G | g\cdot c(G\cdot {\bf X})={\bf X}$} is not empty. I do not understand what is non trivial about $c(G\cdot X)$**
>
> **Reply:** The function $c$ is defined on the quotient space $\mathbb{R}^{3\times m}/G$ and maps to $\mathbb{R}^{3\times m}$. Therefore, it cannot be a trivial identity, which is a function mapping from a space to itself. The composition that sends ${\bf X}$ to $c(G \cdot {\bf X})$ is also not an identity because, for any two ${\bf X}$ and ${\bf X}'$ lying in the same orbit, we have $c(G\cdot{\bf X}) = c(G\cdot{\bf X}')$.
>
> ---
>
> **Q3. Also, $V_0^{\oplus d}$ is not explicitly defined early enough. What is $d$? Does it relate to the maximal rotation order $L$?**
>
> **Reply:** We appreciate your feedback. However, we have introduced $d$ in the first sentence of the statement of Lemma 1, where we mention, "Let $V$ be a $d$-dimensional $G$-steerable vector space…". The parameter $d$ represents the dimension of the $G$-steerable vector space $V$ and is not directly related to the maximal rotation order $L$.
>
> Regarding $V_0$, we acknowledge that we overlooked mentioning in Lemma 1 that we use $V_0$ to denote the trivial 1-dimensional representation of an arbitrary group $G$. We have made a revision to address this.

---

> ### Author Response · Authors · 2023-11-16
> **Response to Reviewer 3HqL (2/6)**
>
> ---
>
> **Q4. You prove that the decomposition of Lemma 1 is unique. But, concretely, what is lambda? Isn't it almost always simply the norm of the irrep? Like, for a vector, its norm, or for a higher-order tensor, also its norm? And then $\rho(g_X)$ is just the usual (matrix) representation of $g_X$. Can it be otherwise? Or is it more complicated? Am I missing the point? If in practice things are simple, in most of the usual settings, saying so is helpful to the reader. OR, maybe $g_X$ simply encodes the orientation of $X$ (say if it's a vector)?**
>
> **Reply:** There seem to be a few points of misunderstanding.
>
> Firstly, the output of the G-invariant function $\lambda$ is a vector of invariant scalars, while the norm is just a scalar. They cannot be the same, especially when the dimension $d$ is greater than $1$. Concretely, $\lambda({\bf X}) = f(c(G \cdot{\bf X}))$ is equal to evaluating the equivariant function $f$ at the representative $c(G \cdot{\bf X})$. Further details can be found in the proof of Lemma 1 (refer to Appendix C in the revised version).
>
> Regarding $\rho$, it represents the group representation on the steerable vector space $V$. On the other hand, $g_{{\bf X}}$ is a group element that satisfies $g_{{\bf X}}\cdot c(G \cdot {\bf X})={\bf X}$. It is true that $g_{{\bf X}}$ encodes the orientation of ${\bf X}$ in the sense that applying rotations or reflections to ${\bf X}$ may lead to a change in $g_{{\bf X}}$ while $c(G\cdot{\bf X})$ remains unchanged. We have provided additional clarification on this point in Remark 1.
>
> It is worth mentioning that $g_{\bf X}$ and $\lambda({\bf X})$ share similarities with local frames and the scalarization proposed in [1], respectively. To provide a more comprehensive understanding, we have included an additional section on related works to discuss the similarities and distinctions between them. Please refer to Appendix A in the revised paper for details.
>
> [1] Du et al. SE(3) Equivariant Graph Neural Networks with Complete Local Frames, ICML 2022.
>
>
> ---
>
> **Q5. In ref [16], page 3, it says: At initialisation, we assign to each node $i\in V$ a scalar node color $c_i\in C’$ and an auxiliary object $g_i$ containing the geometric information associated with it i.e. they already separate (factorize) the scalar (feature only) contributions from the geometric ones (vectorial one v + previous color). In this respect, it is not obvious how contribution (1) (your lemma 1) is new. Furthermore, I lack an intuitive view of what may concretely go into your scalar function lambda(X).**
>
> **Reply:** We believe this is a misunderstanding. In the initialization of the GWL test in [16], the scalar node color only considers the node features. As iterations progress, the color $c_i^{(t)}$ and the auxiliary object $g_i^{(t)}$ incorporate not only the node features but also the geometric information and the information about the graph structure. In contrast, Lemma 1 exclusively focuses on the coordinates (geometric information) and remains independent of node features and the graph structure. In other words, both  $\rho(g_{{\bf X}})$ and $c(G \cdot {\bf X})$ are unaffected by node features and the graph structure.
>
> The separating technique used in Lemma 1 separates ${\bf X}$ into the representative $c(G \cdot{\bf X})$ and the associated group element $g_{{\bf X}}$. Using this technique, we can effectively separate any steerable feature $f({\bf X})$ into its equivariant part $\rho(g_{{\bf X}})$ and its invariant part $\lambda({\bf X})$. It provides us with a framework to investigate steerable features by examining their corresponding invariant features. While we acknowledge that this separating technique might share similarities with the one you mentioned, it is important to emphasize that our framework is more flexible -- it is not limited to GNNs but can be applied to other neural networks or algorithms.
>
> Concretely, $\lambda({\bf X}) = f(c(G \cdot{\bf X}))$ is equal to evaluating the equivariant function $f$ at the representative $c(G \cdot{\bf X})$. Further details can be found in the proof of Lemma 1 (refer to Appendix C in the revised version).

---

> ### Author Response · Authors · 2023-11-16
> **Response to Reviewer 3HqL (3/6)**
>
> ---
>
> **Q6. gothic{g} appears in the r.h.s. of Eq (4) or in the last line of the previous est of equations (let me call this last line Eq. 3b is) (actually the gothic-style has been forgotten in Eq. (4), this is a typo I believe). How can we identify the right term in the r.h.s. of Eq. 3b is as a lambda (invariant), when $g^-1$ appears in it? I don't see why $g^{-1}\cdot x_{ij}$ should be invariant, on the contrary, it seems to me like an equivariant feature.**
>
> **Reply:** Thank you for pointing out the typo. Notice that $g$ is a group element satisfying $g\cdot c(G \cdot{\bf X})={\bf X}$. In other words, we have $ g^{-1}\cdot{\bf X}=c(G\cdot {\bf X})$. It is crucial to note that $c(G \cdot{\bf X})$ is invariant as it only depends on the orbit. The term $g^{-1}\cdot x_{ij}$ is a linear combination of the columns in $g^{-1}\cdot{\bf X}= c(G\cdot{\bf X})$. From this, we deduce its invariance.
>
> ---
>
> **Q7. In Eq. 7, first line, why isn't it $2l+2$ instead of $2l+1$ (in the r.h.s.)? I understood that representing O(3) required one additional component? Could you provide an intuitive explanation?**
>
> **Reply:** The construction of O(3) representations has been thoroughly documented in existing literature, and we have provided a concise review in Section 2. Notably, the irreducible representations of O(3) arise from the product of the irreducible representations of $SO(3)$ and those of ${\pm {\bf I}}$ (rather than a direct sum). Hence, the dimension remains $2l+1$.
>
> ---
>
> **Q8. You write, in page 6: Does any $(2l+1)$-dimensional invariant feature $\lambda({\bf X})$ correspond to a type-l steerable feature $f({\bf X})$? Which sounds like an interesting question (although ideally I'd like to know the mapping, not just know about its existence). Note that this question is essentially asking whether the space of all type-l steerable features $f({\bf X})$ has a dimension of $2l+1$ since $D^l (gX)$ is invertible. But that sounds like a known fact: using Spherical Harmonics, it is known that type-l steerable features need components to be represented. That is, they need $2l+1$ numbers (dimensions)**
>
> **Reply:** We believe the reviewer has a misunderstanding regarding the definition of steerable features in our context. It is true that SO(3)-steerable vector spaces are spanned by Spherical Harmonics, and thus, a type-$l$ steerable vector space $V_l$ has a known dimension of $2l+1$. However, the space of steerable features is a subspace of $V_l$, consisting of type-$l$ steerable vectors that are generated by some equivariant function from the input ${\bf X}$. More formally, it can be expressed as {$ f({\bf X}) \mid \text{ G-equivariant }f:\mathbb{R}^{3\times m}\to V_l $}. In Fig 4 of the revised version, we provide an illustrative example to emphasize that the space of type-$1$ features can be 1-dimensional rather than 3-dimensional, showing that they might not be equal.
>
> The map sending $f({\bf X})$ to $\lambda({\bf X})$ is defined by the composition $\lambda({\bf X}):=f(c(G \cdot{\bf X}))$. The function $c$ is highly related to the notion of alignment in reference [1] listed below. We have included relevant discussions in Appendix A of the revised version.
>
> [1] Winter, R. et al. Unsupervised learning of group invariant and equivariant representations. NeurIPS 2022.

---

> ### Author Response · Authors · 2023-11-16
> **Response to Reviewer 3HqL (4/6)**
>
> **Q9. The remark below corollary 3 is a very strong, original conclusion. I think I understand Corollary 3 (although I feel like in practice lambda is the norm of the representation, and in that case it's kind of trivial..). In any case I do not see how the remark "the expressiveness of learning steerable features is primarily characterized by the feature dimension -- independent of the highest type utilized" follows from corollary 3.**
>
> **Reply:** We have clarified the non-triviality of $\lambda$ in the above responses. Here we would like to address your concern about the implication of Corollary 3. First, we would like to acknowledge that there are two main aspects related to the concept of expressiveness: the capacity of features to carry information and the model's ability to extract that information -- the latter is commonly referred to as universality.
>
> Our discussion on the utilization of different types of steerable features focuses on the former, given that the latter is subject to the model’s architecture. Therefore, asserting that “the expressiveness of learning steerable features is primarily characterized by the feature dimension -- independent of the highest type utilized” might lack precision without considering the model's universality. We have refined the assertion to state that: “the invariant features carried by steerable features are primarily characterized by the feature dimension -- independent of the highest type utilized.”
>
> Additionally, we have rephrased the relevant sections and included a new section in Appendix B of the revised version to delve into universality and potential methods. We believe these revisions address your concerns and provide more precise implications of our results.
>
> ---
>
> **Q10. In table I, I can do the maths for line 1 and 2, but not line 3. (5+3+1)\*256 = 2304 (5+3+1)\*824 = 7416 but then, (1+3+5+7+9+11+13)\*256 =12544, not 7424**
>
> **Reply:** Two hyperparameters are used to control the degree $l$ and order $m$ of the spherical harmonics when generating steerable features. The degree $l$ is directly determined by the steerable feature type $L$. However, the maximum absolute order $m$ is set to $|m|\leq 2$ by default [1] in both eSCN and EquiformerV2. The order $m$ of the spherical harmonics can range from $-l$ to $l$, and when $|m|\leq l$ it serves to restrict the feature space. When $|m|\leq 2$ the equation becomes (1+3+5+5+5+5+5)*256 =7424.
>
> [1] https://github.com/Open-Catalyst-Project/ocp/tree/main
>
> ---
>
> **Q11. experiments are a bit disappointing. 7a. First, Table 2 shows a number of red (interesting) cases, but they turn out to be related to eSCN being SO(3)-equivariant, when the task is O(3). This is not making the point of the paper, and instead is rather confusing, at first sight. Most importantly, it's not clear to me in which sense table 2 is different from table 1 of ref [16] (which by the way seemed more readable and dense). Please clarify this.**
>
> **Reply:** The inclusion of eSCN in Table 2 is used to provide a robust experiment section for all models considered. However, to avoid confusion with the limitations of eSCN, namely the O(3)-equivariance and the quasi-equivarianc, we have included additional discussion in Section 5 and Appendix F.1.
>
> Table 1 of ref [16] considers only chain lengths of $k=4$. This makes it difficult to discern the impact of incorporating $k$-hop information. This is not a concern for the authors of [16] because they do not provide a theoretical understanding of models like ComENet and LEFTNet. Table 2 and Table 5 incorporate a more robust analysis of different models. To improve the clarity of the paper, we have included a more detailed description of the figure in the caption.

---

> ### Author Response · Authors · 2023-11-16
> **Response to Reviewer 3HqL (5/6)**
>
> **Q12. IS2RE. Table 3. Here I enjoyed a lot the idea of comparing models with comparable feature dimension. However, several points: 1) why not report also the total number of parameters? They do not grow as fast as feature dimension since weights are shared for a given $L$ (type order). 2) although I understand it's somewhat hardware dependent, please also report the training time for these models. Maybe even the memory footprint (on GPU). 3) Figure 3 gives an idea of how things perform over a reasonable number of epochs (and I salute the attempt to not go to an exceedingly large number of epochs), but it seems that a more thorough study of variations with $L$ and $c$, reporting only the epoch=12 validation performances, would be useful to the reader (I did not have time to look at the content of ref [5])**
>
> **Reply:**
>
> Thank you for the suggestion. Details for # Param, Run Time, Memory, and validation results for $L=1,3,5$ have been incorporated into Appendix F.2 in Table 6 and Table 7. Below is a reproduction of these tables for the reviewer's convenience.
>
> | Model     	| L |   c | Feat. Dim. | # Param. | Loss                 	| Energy MAE [meV]      	| EwT [%]            	|
> |---------------|---|-----|------------|----------|--------------------------|---------------------------|------------------------|
> | eSCN      	| 1 | 464 | 1856   	| 11M  	| 0.380 ± 0.006        	| 865 ± 14              	| 1.91 ± 0.09        	|
> | eSCN      	| 2 | 206 | 1854   	| 10M  	| 0.369 ± 0.006        	| 842 ± 13              	| 1.94 ± 0.12        	|
> | eSCN      	| 3 | 133 | 1862   	| 9M   	| 0.397 ± 0.001        	| 904 ± 3               	| 1.85 ± .12         	|
> | eSCN      	| 4 | 98  | 1862   	| 9M   	| 0.408 ± 0.006        	| 929 ± 15              	| 1.74 ± 0.12        	|
> | eSCN      	| 5 | 77  | 1848   	| 8M   	| 0.409 ± 0.003        	| 933 ± 7               	| 1.61 ± .12         	|
> | eSCN      	| 6 | 64  | 1856   	| 8M   	| 0.3836 ± 0.003       	| 872 ± 6               	| 1.91 ± 0.19        	|
> | EquiformerV2  | 1 | 77  | 304    	| 7M  	| OOM                  	| OOM                   	| OOM                	|
> | EquiformerV2  | 2 | 34  | 306    	| 9M   	| 0.369 ± 0.009        	| 841 ± 21              	| 2.02 ± 0.14        	|
> | EquiformerV2  | 3 | 22  | 306    	| 12M  	| 0.363 ± 0.009        	| 828 ± 21              	| 1.94 ± 0.08        	|
> | EquiformerV2  | 4 | 16  | 304    	| 15M  	| 0.364 ± 0.005        	| 832 ± 11              	| 2.03 ± 0.14        	|
>
>
> | Model     	| L |   c | Run Time (min)   	| Memory  |
> |---------------|---|-----|----------------------|---------|
> | eSCN      	| 1 | 464 | 151 ± 1          	| 12.3GB  |
> | eSCN      	| 2 | 206 | 297 ± 5          	| 9.1GB   |
> | eSCN      	| 3 | 133 | 207 ± 3          	| 9.0GB   |
> | eSCN      	| 4 |  98 | 347 ± 4          	| 9.2GB   |
> | eSCN      	| 5 |  77 | 246 ± 3          	| 10.9GB  |
> | eSCN      	| 6 |  64 | 429 ± 8          	| 11.5GB  |
> | EquiformerV2  | 1 |  77 | OOM              	| 18.6GB  |
> | EquiformerV2  | 2 |  34 | 284 ± 6          	| 12.6GB  |
> | EquiformerV2  | 3 |  22 | 256 ± 1          	| 11.1GB  |
> | EquiformerV2  | 4 |  16 | 298 ± 2          	| 11.5GB  |
>
>
> Note that for EquiformerV2 L=1, the model will fit onto the GPU.  However, during training the data and the model will exceed the GPU memory capabilities. We denote the out of memory phenomenon OOM in the tables above and report the model parameters and size.
>
> ---
>
> **Q13. S2EF. Table 4 is commented, but no conclusion stands out. What is the point of that table ?**
>
> **Reply:** Table 4 is used to support our theory that no model stands out under a suitable ablation study on $L$ and $c$. This is directly motivated by the analysis in Table 1. Both eSCN[L=6,c=256] and eSCN[L=2,c=824] achieve strong results in comparison to eSCN[L=2,c=256]. However, the fact that eSCN[L=6,c=256] and eSCN[L=2,c=824] themselves are indistinguishable provides empirical support for our theory. In particular, there is no significant gain by increasing $L$ when $c$ is proportionately increased.
>
> ---

---

> ### Author Response · Authors · 2023-11-16
> **Response to Reviewer 3HqL (6/6)**
>
> **Q14. Conclusion. Two key findings are claimed: (1) Propagating steerable features of type $\geq 1$ equips geometric GNNs with the inherent ability to automatically capture the geometry between local structures. (2) When the feature dimension remains constant, increasing the use of steerable features up to type-L cannot essentially improve the performance of equivariant GNNs But I think that (1) is vague and already known. For instance from (16), which clearly distinguishes invariant ($L=0$) from equivariant $L\geq 1$ networks. And that (2) is known as oversquashing and was known before. I note that the term oversquashing is absent from the manuscript, although it is mentioned several times in [16]. I think the term should appear.**
>
> **Reply:** We respectfully disagree with these comments and fear that there are substantial misunderstandings of our contribution. Please allow us to clarify our contribution below.
>
> Concern about (1), we would like to clarify our results not only ***cover cases like ComENet -- as outlined in the limitation section of [16]*** but also emphasize ***the significance of faithfulness in effectively capturing geometry between local neighborhoods during message passing***. It's noteworthy that [16] suggests that incorporating pre-computing non-local features, as in ComENet, may offer a straightforward approach to overcoming the limitations of invariant GNNs. However, our results, both theoretical and empirical, indicate that this approach may not be as effective as suggested. Please refer to Theorem 1 and Table 2. Moreover, [16] asserts that their setup can be extended to higher-order tensors. However, we found that similar results for this extension might not hold if the geometric objects, which correspond to the steerable features, do not lie on faithful representations. In particular, our framework highlights that significant geometric information between local neighborhoods could be lost during message passing if the representations lack faithfulness. Please refer to Theorem 1 and Appendix B in the revised paper for details.
>
> Concern about (2), we did not include a discussion of over-squashing since it is not the problem we are studying and unrelated to our contributions. [16] does mention the over-squashing phenomenon for equivariant features in geometric GNNs, using the example of EGNN, which utilizes a single vector feature to aggregate and propagate geometric information. Specifically, these models are constrained to learn growing information within a fixed-length vector, potentially failing to propagate long-range information and learning only short-range signals from the training data [1].
>
> Let us clarify the empirical evidence supporting the two contributing components of our work. Table 2 presents empirical evidence for **first** part of our theory involving $k$-hop models like ComENet – this was previously unsupported in [16]. In Table 2, we observe the over-squashing in some models as in [16] and **we have included a discussion on over-squashing in Appendix F.2.**
>
> The ablation studies in Figure 2, Table 3, and Table 4 support the **second** part of our contribution which is aimed at answering: **When the feature dimension remains constant, does the performance of equivariant GNNs employing steerable features up to type-$L$ increase as $L$ grows?**
>
> We study the information-carrying ability of different types of steerable features by investigating their corresponding invariant features. In Table 3 and Table 4 we fix the steerable feature dimension while varying the steerable feature type $L$. We select the steerable feature dimension based on the best-performing model (see Table 1). We find that the performance may not necessarily increase when increasing $L$. We do not investigate the phenomenon of over-squashing in this ablation study and we do not observe any effects of over-squashing in our results. Moreover, our theory on the information-carrying ability of steerable features is not limited to GNNs.
>
> We believe that Section 5.2 in [16] is more relevant to the problem we study. As discussed in Section 4, [16] has conducted a comparison of the utilization of different types of steerable features. However, that comparison is not explicitly designed for invariant classification.
>
> In summary, our paper addresses the limitation mentioned in [16] and is the first research highlighting the significance of faithful representations in capturing geometry between local neighborhoods during message passing. Moreover, we provide theoretical reasons supporting the idea that when the feature dimension remains constant, the performance of equivariant GNNs employing steerable features up to type-$L$ may not increase as $L$ grows.
>
> [1] Alon, U., and Yahav, E. On the bottleneck of graph neural networks and its practical implications. ICLR 2021.
>
> [16] Joshi, C. K., Bodnar, C., Mathis, S. V., Cohen, T., and Lio, P. On the expressive power of geometric graph neural networks. ICML 2023.

---

> > ### Comment · Reviewer_3HqL · 2023-11-22
> > **Brief answer**
> >
> > Thank you for your thorough answers. It seems indeed that some of the theoretical subtleties of your work have failed to reach me. I may have read too fast, but this also indicates a relative difficulty to read your messages, for someone who has been working with equivariant networks for over a year.
> > Probably the writing could be further improved.
> >
> > I do not have the time to assess each of your answers individually. However, about over-squashing, I don't understand your answer. Your key empirical finding, that is that increasing L at fixed feature dimension (i.e. reducing the number of channels appropriately) does not lead to increased performance. But isn't that related to a form of oversquashing of the geometrical information ? Meaning, if there are too high order L to be carried, and not enough channels to carry them, then the benefit of these additional L is lost (or even deteriorates performance) ?
> >
> > I will further review your answers later today and most likely increase my rating.

---

> > > ### Author Response · Authors · 2023-11-22
> > > **Further Response to Reviewer 3HqL**
> > >
> > > Dear Reviewer 3HqL,
> > >
> > > Thank you for spending time reviewing our work and rebuttal and providing invaluable feedback. We also thank the reviewer for intending to increase the rating of our paper. We greatly appreciate your valuable insights, which have significantly contributed to the refinement of our work. We acknowledge that some theoretical subtleties in the initial version of our paper required further clarification. Your comments, along with those from other reviewers, have played a crucial role in enhancing the quality and clarity of our paper.
> > >
> > > Concerning the issue of over-squashing, we appreciate your observation of its potential impact on our empirical results. We have incorporated a discussion on this aspect into Section 5 to offer a clearer understanding of the potential factors influencing our empirical results.
> > >
> > > While various factors, including over-squashing, may influence our empirical results, our theoretical contributions to the utilization of different types of steerable features remain robust. Our theoretical framework accommodates the selection of a sufficiently large hidden dimension, ensuring the model has an adequate number of channels to carry different types of steerable features. Furthermore, we highlight the consideration of fully connected graphs, where over-squashing may not occur. Theorem 4 in our revision demonstrates that geometric GNNs learning features on a fixed-dimension steerable vector space can consistently learn the same invariant features at each layer. This result, though sharing similarities with the expressive power equivalence between IGWL and GWL in fully connected graphs (as presented in [20]), distinguishes itself by focusing specifically on the exact features of geometric GNNs without necessitating the injectivity of the update and aggregate functions.
> > >
> > > We are grateful for the work done by the authors of [20], and we aim to provide complementary insights that enrich the ongoing discourse on the expressive power of geometric GNNs.
> > >
> > > Your commitment to reassessing our responses is appreciated, and we remain dedicated to addressing any additional concerns you may have. Once again, we express our gratitude for your valuable feedback and the opportunity to enhance the quality of our paper.
> > >
> > > [20] Joshi, C. K., Bodnar, C., Mathis, S. V., Cohen, T., and Lio, P. On the expressive power of geometric graph neural networks. ICML 2023.

---

### Official Review · Reviewer_BvXD · 2023-10-28

**Soundness:** 3 good
**Presentation:** 2 fair
**Contribution:** 3 good
**Rating:** 6
**Confidence:** 3

**Summary:**

This paper introduces a theoretical analysis of the expressive power of steerable equivariant GNNs. In particular, the authors show that every type-$L$ steerable feature can be analyzed through its corresponding invariant features. The authors use this lemma to study $k$-hop invariant GNNs and show limited expressive power. Then, the authors argue that any type-$L$ steerable feature is as expressive as its dimension. Specifically, there is a one-to-one correspondence between steerable features and $d $ invariants. Hence, increasing $L$, when accounting for the increase of feature dimension, does not improve the GNN's performance. The authors test their findings on several numerical experiments.

**Strengths:**

The writing is of high quality and clarity. Recent works have started analyzing the expressive power of steerable graph NNs, but little consensus has been reached. Hence I'd say this is a timely and significant work.

**Weaknesses:**

See Questions for clarifications.

- I think a highly related work is Villar et al. (2022) ("Scalars are universal...") with several related and perhaps even identical results: please include this in your related work section.
- The notations sometimes could be clarified a bit further.
  - For example, the logic above Lemma 1 could be laid out a bit clearer.
  - It's not clear to me how $\rho(g_X)$ acts on $\lambda(X) \in V_0^{\oplus d}$. First off, what is the action of $\rho(g_X)$? Further, $\lambda(X)$ is a tuple of scalars, so are they all left-invariant? Or are you considering them as a vector in $V$? This is also how you define $f$ in Lemma 1.
  - In general, how can an *equivariant* function be described by *invariants*? I guess I'm not following some notations here.
- I find it slightly confusing what the exact message is that the paper is trying to convey, especially towards the end of the paper.
  - It is claimed that one needs more than invariants to capture the global symmetries, yet Lemma 1 states that one can use invariants to represent equivariant functions (see also my previous comments).
  - I find it slightly confusing that according to the paper, one doesn't need more than $ L=1 $, but many experiments (Table 1, 2, 3) use $ L=2$ and higher.
  - In the conclusion, it is claimed that $L \geq 1$ captures local structures, but one doesn't need more than type $L$. Why don't the authors claim that $L=1$ is enough?

**Questions:**

- Can you contrast your results with Villar et al. (2022)? How do your results differ/improve upon theirs?
Let's take a basis vector $e_1 \in \mathbb{R}^{3 \times 1}$ and $f: \mathbb{R}^{3 \times 1} \to \mathbb{R}^3$ with $e_1 \mapsto \alpha_1 e_1, \alpha \in \mathbb{R}$. This is clearly an equivariant function. What would the corresponding invariant function be such that Lemma 1 holds?
- Let's take three basis vectors $e_1, e_2, e_3 \in \mathbb{R}^{3\times3}$ and $f: \mathbb{R}^{3 \times 3} \to \mathbb{R}^3$ with $e_1, e_2, e_3 \mapsto \alpha_1 e_1 + \alpha_2 e_2 + \alpha_3 e_3, \alpha \in \mathbb{R}$.
  - Isn't the set stabilizer now only the trivial rotation?
  - What would, in this case, be the corresponding unique $\lambda$?
- Why did you use $L=2$ and higher in your experiments in Tables 1, 2, and 3?
- Could you elaborate on my last comments in the previous (weaknesses) section?
- Do you know if your results were somehow already known in (geometric) invariant theory?

---

> ### Author Response · Authors · 2023-11-16
> **Response to Reviewer BvXD (1/3)**
>
> Thank you for your thoughtful review and valuable feedback. In what follows, we provide point-by-point responses to your comments.
>
> ---
>
> **Q1. Can you contrast your results with Villar et al. (2022)? How do your results differ/improve upon theirs? Let's take a basis vector $e_1 \in \mathbb{R}^{3 \times 1} $ and $f: \mathbb{R}^{3 \times 1} \rightarrow \mathbb{R}^3$ with $e_1 \mapsto \alpha e_1, \alpha \in \mathbb{R}$. This is clearly an equivariant function. What would the corresponding invariant function be such that Lemma 1 holds?**
>
> **Reply:** We have integrated the work by Villar et al. (2022) into our related work, including a discussion in Appendix A of the revision on the distinctions between their research and ours. Here, we provide a brief discussion.
>
> Our results exhibit clear distinctions from theirs, mainly due to our consideration of steerable features of higher types $l>1$ and the different approaches we have taken. Instead of investigating the spanning space, we leverage the concept of "representatives" of orbits and apply the equivariance of the function to derive Lemma 1.
>
> Our form also differs from Villar et al. (2022) in terms of the number of basis vectors and the uniqueness of invariants. In our form, the number of basis vectors aligns with the dimension of (steerable) features, rather than the number of input coordinates. Taking your example into account, where the function $f$ sends $e_1$ to $\alpha_1 e_1$. In the form provided in Villar et al. (2022), $f$ would be expressed as a linear combination of $e_1$, resulting in one basis vector. However, in our form, the number of basis vectors corresponds to the number of column vectors in $\rho(g_{{\bf X}})$ (which is three in this case). Moreover, $\lambda({\bf X})$ is directly determined by $\rho(g_{{\bf X}})^{-1}f({\bf X})$, using the invertibility of $\rho(g_{{\bf X}})$. This not only ensures the uniqueness of $\lambda$ but also highlights a key distinction. In Villar et al. (2022)’s framework, the invariants may not be unique, especially if the number of input coordinates exceeds the rank of their spanning space.
>
> Let’s consider the example you provided, where the function $f$ sends $e_1$ to $\alpha_1 e_1$. In this case, the input embedding contains only one vector, namely $e_1$. There exists a rotation that transforms $e_1$ to the vector $(||e_1||, 0, 0)$, which lies on the $x$-axis. Here, we choose the representative $c(G\cdot{\bf X})$ to be the vector $(||e_1||, 0, 0)$ and the associated group element $g_{{\bf X}}$ to be the transformation that rotates $(||e_1||, 0, 0)$ to $e_1$. Then, $\lambda({\bf X}) = f(c(G\cdot{\bf X})) = f((||e_1||, 0, 0)) =(\alpha_1 ||e_1||, 0, 0)$ is equal to evaluating $f$ at the representative.
>
>
> Additionally, we have established different correspondences between equivariant functions and invariant functions. Our results also involve a detailed investigation into the message-passing mechanisms of equivariant GNNs and invariant GNNs. We believe our results contribute to a more comprehensive understanding of the benefits offered by learning steerable features in geometric GNNs.
>
> ---
>
> **Q2. Let's take three basis vectors and with $e_1,e_2,e_3\in\mathbb{R}^{3 \times 1}$ and $f: \mathbb{R}^{3 \times 3} \rightarrow\mathbb{R}^3$ with $e_1,e_2,e_3\mapsto \alpha e_1 + \alpha e_2 + \alpha e_3, \alpha \in \mathbb{R}.$ 1) Isn't the set stabilizer now only the trivial rotation? 2) What would, in this case, be the corresponding unique?**
>
> **Reply:** The stabilizer is defined based on a given input embedding ${\bf X}$. Additionally, the size of stabilizers is contingent on the rank of ${\bf X}$, as detailed in the proofs of Theorems 2 and 3. Notably, when the rank of ${\bf X}$ is one, the stabilizer is formed by rotations around the line spanned by the column vectors in ${\bf X}$.
>
> As mentioned earlier, the corresponding $\lambda({\bf X})$ is equal to $\rho(g_{{\bf X}})^{-1}f({\bf X})$, while we define $\lambda({\bf X})$ to be $f(c(G\cdot{\bf X}))$  in the proof of Lemma 1. Therefore, the explicit value of $\lambda({\bf X})$ may vary depending on the choice of the representative $c(G\cdot{\bf X})$ and the group element $g_{\bf X}$. It is challenging to explicitly work through your example due to the difficulty in explicitly constructing $c(G\cdot{\bf X})$ or $g_{{\bf X}}$ for any ${\bf X}$. In particular, this construction is highly related to the notion of alignment in reference [1] listed below, where they apply an autoencoder to learn $c(G\cdot{\bf X})$. (We also added a few comments to discuss this in Appendix B of the revision)
>
> Nevertheless, we have performed computations for the associated group element $c(G\cdot{\bf X})$ and the corresponding invariant function $\lambda({\bf X})$ using another example you provided in Q1. We hope this addresses your concerns.
>
> [1] Winter, R. et al. Unsupervised learning of group invariant and equivariant representations. NeurIPS 2022.

---

> ### Author Response · Authors · 2023-11-16
> **Response to Reviewer BvXD (2/3)**
>
> ---
>
> **Q3. Why did you use $L=2$ and higher in your experiments in Tables 1, 2, and 3?**
>
> **Reply:** Table 1 rows 1&3 report the baseline hyperparameters for the eSCN model. Table 1 row 2 illustrates what we propose as a suitable comparison. In particular, we fix $L=2$ to maintain the steerable feature type in the baseline, while adjusting the hidden channels $c$ in order to maintain the dimension of the steerable feature. Table 2 reports results for different models which use $L=0,1,2$. We report results for $L=2$ to demonstrate the effects of $k$-Chains when using higher-order steerable features. Table 3 reports results for $L=2,4,6$ for eSCN and $L=2,4$ for equiformerV2. This serves as an ablation study over the steerable feature dimension $L$. We have enhanced the robustness of this study in the revision which now includes results for $L=1,3,5$.
>
> ---
>
> **Q4. Concern about exact message in the paper. In particular, elaborate on the last comments in the weaknesses section?**
>
> **Reply:** We would like to address each of your concerns individually.
>
> For your first concern, we want to clarify that there is no inconsistency. While Lemma 1 allows us to study equivariant features by examining their corresponding invariant features, it is essential to consider the message-passing mechanisms, especially the local aggregation, when investigating geometric GNNs. We observe that, due to the faithfulness of representations, equivariant GNNs can effectively capture the geometry between local neighborhoods. However, hindered by trivial representations, invariant GNNs struggle in this regard significantly affecting their ability to precisely capture global geometry and obtain global invariant features from local invariant features. Basically, there are scenarios where equivariant GNNs can learn certain global invariant features that invariant GNNs cannot.
> In Appendix B of the revision, we have added a detailed explanation of why faithfulness is important.
>
> The response to your second concern is contained in our response to Q3.
>
> For the last concern, we would like to acknowledge that there are two main aspects related to the concept of expressiveness: the capacity of features to carry information and the model's ability to extract that information -- the latter is commonly referred to as universality. Our discussion on the utilization of different types of steerable features focuses on the former, as the latter is subject to the given model architecture. It is not sufficient to conclude that $L=1$ is enough without considering universality. We have added a section in Appendix B of the revised version to discuss universality and potential methods. Nevertheless, our results still show that when the feature dimension remains constant, the performance of equivariant GNNs employing steerable features up to type-$L$ may not increase as $L$ grows.
>
> We have revised the relevant sections in our paper to enhance clarity and eliminate potential misunderstandings. We believe these revisions address your concerns and provide more precise implications of our results.
>
> ---
>
> **Q5. Do you know if your results were somehow already known in (geometric) invariant theory?**
>
> **Reply:** We have included an additional section in the revised version to discuss the similarities and distinctions between Lemma 1 and existing results. While we have conducted an extensive literature review, we have not identified an exact identical result in the existing literature.
>
> However, if there is the same result already presented in the existing literature, we would appreciate it if the reviewer could provide a specific reference. This information will be crucial for us to acknowledge and appropriately credit prior contributions.

---

> ### Author Response · Authors · 2023-11-16
> **Response to Reviewer BvXD (3/3)**
>
> **Q6. Concern about notations mentioned in weakness?**
>
> **Reply:** Thank you for your feedback. We have improved the presentation of the argument above Lemma 1.
>
> Regarding your concern about the group action in Lemma 1, we have provided further clarification in Remark 1, particularly emphasizing that the group action is absorbed by $\rho(g_{\bf X})$. An alternative perspective is to consider the product $\rho(g_{\bf X})\lambda({\bf X})$ as a linear combination of the column vectors of $\rho(g_{\bf X})$ with the coefficients given by the scalars in $\lambda({\bf X})$. Induced by the definition of group homomorphism $\rho(h\cdot g) = \rho(h)\cdot \rho(g)$, the group action then acts on the column vectors of $\rho(g_{\bf X})$.
>
> Furthermore, the well-defined mapping in Lemma 1, which associates equivariant functions with invariant functions, allows us to use the corresponding invariant function $\lambda({\bf X})$ to understand an equivariant functions $f({\bf X})$. In the proof of Lemma 1, we define $\lambda({\bf X})$ as $f(c(G\cdot{\bf X}))$, evaluating $f$ at the representative $c(G\cdot{\bf X})$. The key reason $\lambda({\bf X})$ sufficiently represents $f({\bf X})$ is that we can use the associated group element $g_{\bf X}$ to recover $f$ from $\lambda({\bf X})$ by computing $\rho(g_{\bf X})\lambda({\bf X})$. It is essential to note that $g_{\bf X}$ is pre-selected and is independent of $f$. In fact, $g_{\bf X}$ shares similarities with local frames in reference [1] listed below. To provide a more comprehensive understanding, we have included an additional section on related works to discuss the similarities and distinctions between $g_{\bf X}$ and local frames. Please refer to Appendix A in the revised paper for details.
>
>
> [1] Du et al. SE(3) Equivariant Graph Neural Networks with Complete Local Frames, ICML 2022.
>
> ---
>
> We have updated our submission based on the reviewer's feedback, with the revision highlighted in blue. We are happy to address further questions on our paper. Thank you for considering our rebuttal.

---

> > ### Comment · Reviewer_BvXD · 2023-11-22
> >
> > Thanks for your detailed feedback. The authors have made considerable effort to improve the paper based on the extensive feedback. In light of these improvements, I have decided to promote my score slightly because I think it is worth publishing this paper rather than not. However, I still agree with many of the issues reviewers 3HqL and WxNP present, and encourage the authors to provide further clarifications.

---

> > > ### Author Response · Authors · 2023-11-22
> > > **Further Response to Reviewer BvXD**
> > >
> > > We appreciate your positive response and the promotion of our score. In addressing the concerns raised by reviewers 3HqL and WxNP, we have provided detailed responses and implemented improvements based on their feedback. We are open to addressing any further questions or concerns that may arise from these issues and are committed to ensuring the clarity and quality of our paper.

---

### Official Review · Reviewer_WxNP · 2023-10-30

**Soundness:** 3 good
**Presentation:** 2 fair
**Contribution:** 3 good
**Rating:** 6
**Confidence:** 5

**Summary:**

In recent years there has been growing interest in various forms of `geometric graph neural networks', where the input are graphs whose node have attributes in R^3, and the tasks discussed are equivariant to both relabeling of the nodes and applying a global rigid motion to all node coordinates.

There is a zoo of different methods for these problems, which differ among others in the use of invariant vs equivariant features and in the type of GNN used as a backbone. This paper attempts to understand the importance  of these various different choices. It  discusses k-hop GNNs with invariant/equivariant features. Its main claims are:
(a) Invariant features are less expressive than equivariant features.
(b) equivariant features of order 1 are as expressive as higher order features- the main issue is the dimension of the features and not the equivariant order

**Strengths:**

The paper has some observations I believe are novel and interesting:
(a) an interesting correspondence between equivariant functions and invariant functions, and between equivariant functions spaces of different order.
(b) the conjecture that high dimensional representations do not matter, once the number of features is evened out, is interesting and recieves some empirical and theoretical evidence in the paper.

**Weaknesses:**

* Regarding (a): This correspondence has an important limitation which I believe the authors should mention:  Update and Aggregation functions which are actually useful for learning are reasonably `nice': for example, differentiable almost everywhere, continuous, etc. The correspondence suggested by lemma 1 uses a mapping of every orbit to a canonical element which indeed exists, but is not in general continuous or very well behaved. As a result an equivariant `nice' aggregation function will correspond to a `wild' invariant aggregation function.

* The paper's stand on invariant vs equivariant features seems inconsistent. On the one hand the paper maintains that there is a one-to-one correspondence between invariant and equivariant features, and that "propagation of steerable features can be effectively understood as propagation of invariant features, therefore analyzing the message passing using the corresponding invariant features is a reasonable approach (page 4)" on the other hand once this analysis is carried out the paper maintains that it points to an advantage of equivariant over invariant.

*I have some issues with the writing, which could be addressed in a revision towards a camera ready version.

**Questions:**

Dear authors: please relate to the first two weaknesses above- what your opinion is about them, and how you intend to address them if you agree they should be addressed.

Detailed remarks about writing
* Page 4: Steerable features: It would make more sense to me to define f to be the steerable features. What does it mean to say that a vector is a steerable feature? Say you have a vector x=[1,2,5] in R^3. Is it a steerable feature or not?
* Page 5: "without loss of generality we may assume the group reps are all the same" why?
* Remark 3 is vague and out of context. Corollary 3 seems related, is much more formal and accurate, but does not require the action to be faithful. Can you formally explain why faithfulness is important? This could be the criitical missing link explaining why in terms of the irreducible order $\ell$ we claim to have 0<1=2=3=4=... which is the message you are trying to convey but I don't think the theory currently supports.
* Page 6: "This question is essentially asking whether the space of all type-ell features has a dimension of 2\ell +1" This is well defined once f is fixed I believe and X is free? Perhaps clarifying this will also help me understand how you define steerable features in general.
* Page 6: Could you specify v=f(x_1,x_2) or whatever the relation between v and x_1,x_2 is?
* Also I think a similar observation appears in [Villar, Lemma 3] I suggest you cite them
* Table 1: what are the significance of the specific three values of (L,c) chosen in Table 1? Presumably you can run eSCN with any choice of the c-s?
* In Table 2 the invariant equiformerV2 actually does reasonably well, seemingly contradicting your theorem?

small comments that should not be addressed in rebuttal:
* Page 2: the stabilizer is not only a set but also a group, perhaps you would prefer to say 'group'
* Page 3: I didn't understand the definition of k-hop distinct. Possibly a typo somewhere there?
* in Lemma 1 and elsewhere: in the formula f(X)=... the output of lambda is in $V_0^{d}$ and then you apply $\rho(g)$ and get something in $V$. I understand what you mean but technically the group action should go from V to V.
* Page 4: "for simplicity we represent g as g_X" I believe here and a line or two above you accidently swapped between g and g_X
* In theorem 2 has V_l,aug been previouisly defined? Couldn't easily find the definition.
  [Villar et al. Scalars are universal: Equivariant machine learning, structured like classical physics. Lemma numbering from arxiv version]
 *Page 9: `our work has evoked' I would suggest different wording focusing on what the works does rather than what it caused the reader to do.

---

> ### Author Response · Authors · 2023-11-16
> **Response to Reviewer WxNP (1/3)**
>
> Thank you for your thoughtful review and valuable feedback. In what follows, we provide point-by-point responses to your comments.
>
> ---
>
> **Q1. Regarding (a): This correspondence has an important limitation which I believe the authors should mention: Update and Aggregation functions which are actually useful for learning are reasonably ‘nice’: for example, differentiable almost everywhere, continuous, etc. The correspondence suggested by lemma 1 uses a mapping of every orbit to a canonical element which indeed exists, but is not in general continuous or very well behaved. As a result an equivariant ‘nice’ aggregation function will correspond to a ‘wild’ invariant aggregation function.**
>
> **Reply:** The point you raised is accurate. In the revised version, we have provided additional explanations and potential methods to address this concern. For detailed information, please refer to Appendix B.
>
> ---
>
> **Q2. The paper's stand on invariant vs equivariant features seems inconsistent. On the one hand the paper maintains that there is a one-to-one correspondence between invariant and equivariant features, and that “propagation of steerable features can be effectively understood as propagation of invariant features, therefore analyzing the message passing using the corresponding invariant features is a reasonable approach (page 4)” on the other hand once this analysis is carried out the paper maintains that it points to an advantage of equivariant over invariant.**
>
> **Reply:** We appreciate the reviewer's careful examination of our paper. We want to clarify that there is no inconsistency in our stance. Corollary 1 establishes one-to-one correspondences between invariant and equivariant features. However, when investigating geometric GNNs, it is essential to consider the message-passing mechanisms. We observe that, due to the faithfulness of representations, equivariant GNNs can effectively capture the geometry between local neighborhoods, while invariant GNNs -- hindered by trivial representations -- struggle in this regard. This distinction significantly affects their ability to precisely capture global geometry and derive global invariant features from local invariant features. **There are scenarios where equivariant GNNs can learn certain global invariant features that invariant GNNs cannot.**
>
> To enhance clarity and mitigate potential misunderstandings, we have revised all relevant contexts in our paper. We believe these revisions address your concerns and provide more precise implications of our results.
>
>
>  ---
>
> **Q3. Page 4: Steerable features: It would make more sense to me to define f to be the steerable features. What does it mean to say that a vector is a steerable feature? Say you have a vector $x=[1,2,5]$ in $\mathbb{R}^3$. Is it a steerable feature or not?**
>
>
> **Reply:** Features, in the context of our discussion, refer to values obtained through a parameterized function applied to the input. That's why we refer to the steerable vector $f({\bf X})$, generated by the equivariant function $f$ from the input ${\bf X}$, as a steerable feature. It's important to note that not every arbitrary (steerable) vector, like $x=[1,2,5]$, qualifies as a steerable feature. There may not be an equivariant function such that $f({\bf X}) = [1,2,5]$ for some ${\bf X}$ with a rank less than 3. We have provided an example illustrating this in Figure 4 of the revision.
>
> ---
>
> **Q4. Page 5: "without loss of generality we may assume the group reps are all the same" why?**
>
>
> **Reply:** To enhance clarity, we have replaced the phrase “without loss of generality” with “in practice.” In practice, models are typically designed to update hidden features in the same space before the last layer. Consequently, we can reasonably assume that the group representations are all the same.

---

> ### Author Response · Authors · 2023-11-16
> **Response to Reviewer WxNP (2/3)**
>
> **Q5. Remark 3 is vague and out of context. Corollary 3 seems related, is much more formal and accurate, but does not require the action to be faithful. Can you formally explain why faithfulness is important? This could be the critical missing link explaining why in terms of the irreducible order $\ell$ we claim to have $0<1=2=3=4=\ldots$ which is the message you are trying to convey but I don't think the theory currently supports.**
>
>
> **Reply:** We have added a detailed explanation of why faithfulness is important in Appendix B of the revised paper. Here, we elaborate on the significance of the faithfulness of representations to address your concern. The faithfulness of representations ensures that the data $\rho\big((\mathfrak{g}^{(t+1)}_i)^{-1}\mathfrak{g}^{(t)}_j\big)$ must originate from $(\mathfrak{g}^{(t+1)}_i)^{-1}\mathfrak{g}^{(t)}_j$. Without faithfulness, $\rho\big((\mathfrak{g}^{(t+1)}_i)^{-1}\mathfrak{g}^{(t)}_j\big)$ might stem from different group elements. For instance, consider the worst case where the representation is trivial. Then different $(\mathfrak{g}^{(t+1)}_i)^{-1}\mathfrak{g}^{(t)}_j$ are all mapped to the same matrix $\rho\big((\mathfrak{g}^{(t+1)}_i)^{-1}\mathfrak{g}^{(t)}_j\big) = {\bf I}$, the identity matrix. In other words, the faithfulness of representations ensures that our models do not lose this geometric information.
>
> In practice, equivariant GNNs are designed to learn steerable features up to type $L>0$, including the learning of type-1 steerable features, which lie on faithful representations. Therefore, as we have explained in Remark 3, they exhibit the same ability to capture the geometry between local neighborhoods even as $L$ varies.
>
> ---
>
> **Q6. Page 6: “This question is essentially asking whether the space of all type-$l$ features has a dimension of $2l+1$” This is well defined once $f$ is fixed I believe and $X$ is free? Perhaps clarifying this will also help me understand how you define steerable features in general.**
>
>
> **Reply:** In this question, ${\bf X}$ is a fixed input and $f$ represents any equivariant function. Therefore, we are examining the space {$ f({\bf X}) \mid \text{ G-equivariant } f:\mathbb{R}^{3\times m}\to V_l $}, consisting of all possible type-$l$ steerable features generated from ${\bf X}$.
>
> ---
>
> **Q7. Page 6: Could you specify v=f(x_1,x_2) or whatever the relation between v and x_1,x_2 is?**
>
>
> **Reply:** According to our definition of steerable features, $v = f(x_1, x_2)$ is a type-1 steerable vector that is generated by an equivariant function $f$ from the input coordinates $x_1,x_2$. The relation between $v$ and $x_1,x_2$ can be explained as follows: applying a transformation $g$ to $x_1$ and $x_2$ will result in a changed vector $g\cdot v$ due to the equivariance of $f$.
>
> In the revision, this example has been moved to Appendix G.
>
> ---
>
> **Q8. I think a similar observation appears in [Villar, Lemma 3] I suggest you cite them**
>
>
> **Reply:** Thank you for bringing this to our attention. We have incorporated [Villar] into the additional related work section and provided a thorough discussion outlining the differences between their work and ours. For more details, please refer to Appendix A in the revised version.
>
> ---
>
> **Q9. Table 1: what are the significance of the specific three values of (L,c) chosen in Table 1? Presumably you can run eSCN with any choice of the c-s?**
>
>
> **Reply:** Table 1 demonstrates that existing comparisons made between feature type L and channels c are unfair due to the significant decrease in the dimension of the hidden features and the number of parameters. The existing ablation study uses rows 1&3, only adjusting the feature type. This results in vastly different model sizes. By contrast, comparing rows 1&2 would be a more suitable comparison in which case the number of parameters and the feature dimension are similar. In particular, maintaining the feature dimension is crucial for providing empirical evidence supporting our results.
>
> To clarify this we have added the following sentence to the revision at the end of the section “Numerical comparisons in steerable feature types.”: The ablation study in [23] compares rows 1&3, while rows 2&3 provide a more suitable comparison.

---

> ### Author Response · Authors · 2023-11-16
> **Response to Reviewer WxNP (3/3)**
>
> ---
>
> **Q10. In Table 2 the invariant equiformerV2 actually does reasonably well, seemingly contradicting your theorem?**
>
>
> **Reply:** While the equiformerV2 model obtains improved accuracy over other invariant architectures in Table 2, it is difficult to assume that this is a desirable performance feature. Due to the enhanced robustness of our dataset, which is obtained by augmenting the data with rotations, reflections, and translations, it is difficult to determine if the improvement in accuracy is due to sensitivity to the numerical precision in our augmentation. In particular, we observe that the best model is still three standard deviations away from perfectly classifying the $k$-chains. For this reason, we do not draw any strong conclusions about the invariant equiformerV2 architecture.
>
>
> ---
> Small Comment List:
>
> We have clarified that stabilizers are subgroups.
>
> We have corrected the typo in the definition of k-hop distinct.
>
> In response to your concern about group action in Lemma 1, we have added further explanation in Remark 1, particularly emphasizing that the group action is absorbed by $\rho(g_{\bf X})$.
>
> The definition of $V_{l, aug}$ is presented in Section 2, where we introduce $O(3)$-steerable vector spaces.
>
> We have refined our phrasing to improve clarity and precision.
>
>
> ---
>
> We have updated our submission based on the reviewer's feedback, with the revision highlighted in blue. We are happy to address further questions on our paper. Thank you for considering our rebuttal.

---

> > ### Comment · Reviewer_WxNP · 2023-11-20
> >
> > Thanks for your answers. Here’s an initial response hope to discuss in more detail in a later time.
> >
> > I feel like the theoretical message is still a little confusing. In particular one thing which could help imho is showing that:
> > * equivariant networks can Separate geometric graphs from theorem 1 which invariant networks cannot
> > * all equivariant networks with faithful reps can separate geometric graphs equally well
> >
> > can you prove these two statements. How far is the second statement from remark3?

---

> ### Author Response · Authors · 2023-11-20
> **Further Response to Reviewer WxNP**
>
> Thank you for your continued engagement and feedback. We appreciate your interest in clarifying the theoretical aspects.
>
> - Regarding the 1st statement, demonstrating that equivariant networks can effectively distinguish geometric graphs require strong assumptions about the injectivity of the update function (UPD), the aggregation function (AGG), and the graph-level readout function, as highlighted in Prosition 2 in [20]. The work in [20] has already proven, under these assumptions, that equivariant networks have the same expressive power as the geometric Weisfeiler-Lehman (GWL) test, enabling them to distinguish any two $k$-distinct geometric graphs ($k$ is arbitrary). Please refer to Propositions 2 and 3 in Appendix D of our revised paper for a quick review. It's worth noting that using the GWL test to establish distinguishability is preferable due to its direct connection with the task at hand.  However, as the distinguishability of equivariant GNNs has been extensively explored in [20], we tend to focus on $k$-hop invariant GNNs, which cover the cases outlined in the limitation section of [20]. ***Our framework enables us to identify the underlying issues, specifically non-faithful representations, that limit the distinguishability of $k$-hop invariant GNNs.***
>
>
> - About the 2nd statement, it is easier to show that equivariant GNNs, assuming faithful representations and the injectivity of the update function, aggregation function, and graph-level readout function, have the same capability as the GWL test. Indeed, the proof of Proposition 2 in [20] can be directly applied to demonstrate this. We emphasize the significance of faithfulness in our work because the proof provided in [20] does not hold for equivariant GNNs learning features on non-faithful representations. Please refer to Appendix B in our revision for details.
>
> We hope these responses address your concerns, and we are happy to address further questions regarding our paper.
>
> [20] Joshi, C. K., Bodnar, C., Mathis, S. V., Cohen, T., and Lio, P. On the expressive power of geometric graph neural networks. ICML 2023.

---

> > ### Comment · Reviewer_WxNP · 2023-11-20
> >
> > * So what's the bottom line? The two claims I suggested above are correct? But you do not wish to state them in the revision?
> > I would be inclined to raise my score if you would, or would provide some other precise theorems explaining why different faithful Equivariant networks are equivalent, but invariant networks are not.
> >
> > Let me fill in on some more details which I didn't get to earlier today. First of all thanks for the revision and taking my comment seriously, I appreciate it. Remarks:
> > * As I see it, and this is a matter of taste of course, the main interesting claim in the paper is that faithful representations of any kind should be equivalent. This is demonstrated in experiments but on the theory there is only remark 3 which seems like hand-waving. If there is something precise to say (e.g. the second claim I suggested) I suggest you say it *in the main text*.
> >
> > * Corollary 3 applies to any representation, also non-faithful. It applies even for the trivial representation. So while interesting it is also a bit confusing as to the takeaway message. I'm not saying it isn't a good statement to have. But it is difficult to say that it justifies your main claims in the paper.
> >
> > * I find the k-hop contribution new but less interesting since it seems natural given the result in [20] re 1-hop. Also most geometric GNNs I believe, use 1-hop. In particular in the appendix where you discuss this all the methods you discuss are 1-hop methods.
> >
> > * Re invariant equiformer2: From reading the paper without going into the detail of this method, it seems to be two orders of magnitude better than the 50% I would expect from your theorems.  I think you should add this to the discussion and explain why you think this is (numerical errors, whatever).

---

> > > ### Author Response · Authors · 2023-11-21
> > > **Further Response to Reviewer WxNP (1/2)**
> > >
> > > Dear Reviewer WxNP,
> > >
> > > Thank you for your further feedback and invaluable comments and suggestions, which have significantly contributed to enhancing the clarity of our paper. We are sincerely sorry for not being able to update the revised paper instantaneously in response to your previous comment since it took us a while to restructure the paper.
> > >
> > > Following your suggestion, we have modified the discussion on the faithfulness of representations and the disparity between Corollary 3 and Section 3.1. In particular, we have established Theorem 2 and Theorem 4 and have included additional explanations to address your concerns on our theoretical contribution. We appreciate the reviewer for being inclined to raise the rating score.
> > >
> > > ---
> > >
> > > **Q1. So what's the bottom line? The two claims I suggested above are correct? But you do not wish to state them in the revision? I would be inclined to raise my score if you would, or would provide some other precise theorems explaining why different faithful Equivariant networks are equivalent, but invariant networks are not.**
> > >
> > > **Reply:** These two claims are correct if we assume the update function and aggregate function satisfy certain injectivity conditions. The first claim for equivariant GNNs learning steerable features up to type $1$ has been proved in [20]; refer to Proposition 7 in Appendix D of our revised paper.
> > >
> > > Now we include Theorem 2 to demonstrate that with the same assumption on the injectivity of  the update function and aggregate function, equivariant GNNs learning steerable features on faithful representations can distinguish any two $k$-hop distinct geometric graphs with a sufficient number of iterations.
> > >
> > > Following your suggestions, we have included the statement of Theorem 2 in the main text. We appreciate your suggestions, which indeed significantly improve our paper.
> > >
> > >
> > > ---
> > >
> > > **Q2. As I see it, and this is a matter of taste of course, the main interesting claim in the paper is that faithful representations of any kind should be equivalent. This is demonstrated in experiments but on the theory there is only remark 3 which seems like hand-waving. If there is something precise to say (e.g. the second claim I suggested) I suggest you say it in the main text.**
> > >
> > > **Reply:** We sincerely thank you for your comment again. In the revised paper, we have introduced Theorem 2 to demonstrate that under certain assumptions on the injectivity of the update function and aggregate function, equivariant GNNs learning steerable features on faithful representations can distinguish any two $k$-hop distinct geometric graphs with a sufficient number of iterations.
> > >
> > > ---
> > >
> > > **Q3. Corollary 3 applies to any representation, also non-faithful. It applies even for the trivial representation. So while interesting it is also a bit confusing as to the takeaway message. I'm not saying it isn't a good statement to have. But it is difficult to say that it justifies your main claims in the paper.**
> > >
> > > **Reply:** To address this confusion, we have introduced Theorem 4 in the revision and have provided additional explanations to clarify the distinction between Corollary 3 and the necessity of faithfulness.
> > >
> > > In particular, we employed a similar approach used in the proof of Corollary 3 to establish the equivalence of geometric GNNs on fully connected graphs. This suggests that geometric GNNs with the same hidden dimension can learn the same corresponding invariant features on fully connected graphs, even without relying on the faithfulness of representations.
> > >
> > > However, we have emphasized that this proof depends on the understanding of how geometric GNNs capture global geometry. When dealing with fully connected graphs, they exhibit the same capacity to capture global geometry. When dealing with non-fully connected graphs, each node can only capture global geometry through message-passing. The faithfulness of representations then influences the model's ability to capture global geometry through message passing, as highlighted in Section 3.1.
> > >
> > > Proving an analogous result to Theorem 4 for non-fully connected graphs under the assumption of the faithfulness of representations is more challenging and complicated. We believe it may be more feasible to demonstrate this by considering specific architectures with a better understanding of how they obtain global geometry from local information.
> > >
> > > Nevertheless, as we have demonstrated in Theorem 2 and Remark 3 of the revised paper, equivariant GNNs learning steerable features up to type-$L$ exhibit the same capacity to capture global geometry. Consequently, we assert that the performance of equivariant GNNs employing steerable features up to type-$L$ may not increase as $L$ grows.
> > >
> > > ---

---

> ### Author Response · Authors · 2023-11-21
> **Further Response to Reviewer WxNP (2/2)**
>
> **Q4. I find the $k$-hop contribution new but less interesting since it seems natural given the result in [20] re 1-hop. Also most geometric GNNs I believe, use 1-hop. In particular in the appendix where you discuss this all the methods you discuss are 1-hop methods.**
>
> **Reply:** While the findings in [20] extend easily from 1-hop to $k$-hop, it is crucial to note that [20] identified these works as limitations and suggested that integrating non-local features, like ComENet, might overcome the limitation of propagating invariant features.
>
> However, our work fills this gap and emphasizes the significance of incorporating global features. Notably, our results suggest that the integration of non-local features may not adequately address the limitations of invariant GNNs. Our analysis indicates that considering global features is necessary, as evidenced by the performance of ComENet and LEFTNet, both classified as 2-hop methods.
>
> It is noteworthy that designing multi-hop methods is more challenging than 1-hop methods, and they have been considered a potential solution to mitigate the limitations of invariant GNNs. Nevertheless, our work emphasizes that relying solely on multi-hop methods might not be as effective a solution as previously anticipated.
>
> ---
>
> **Q5. Re invariant equiformer2: From reading the paper without going into the detail of this method, it seems to be two orders of magnitude better than the 50% I would expect from your theorems. I think you should add this to the discussion and explain why you think this is (numerical errors, whatever).**
>
> **Reply:** The EquiformerV2 and eSCN architectures both rely on the spherical activation function introduced in eSCN which is not truly equivariant but quasi-equivariant. The details of the quasi-equivariance introduced by the architectures are discussed in Appendix D of [1] listed below. These quasi-equivariant architectures introduce error into the equivariance. This error is larger when the grid size of the spherical activation function is small as evidenced in Figure 10 of [1]. The grid size scales with $L$ and $m$ as $(2L+1)\times(2m+1)$, therefore the error is largest and the grid is smallest when $L=m=0$. We have provided more discussion on the error introduced by quasi-equivariant architectures; see Section 5 and Appendix F.1 of the revised paper for details.
>
> [1] Saro Passaro and C. Lawrence Zitnick. Reducing SO(3) Convolutions to SO(2) for Efficient Equivariant GNNs. ICML 2023.
>
> ---
>
> We hope these responses address your concerns and we are happy to address further questions regarding our paper. Thank you for considering our rebuttal.

---

> > ### Comment · Reviewer_WxNP · 2023-11-22
> > **Revised evaluation**
> >
> > Thanks for addressing my remarks,  I’ve raised my score to 6.
> >
> > To summarize, I think the main nice result in the paper is the theoretical and empirical results suggesting that for geometric gnn there may not be a need to use high dimensional features. This is an important point.
> >
> > Disadvantages of the paper are that the proof of theorem 2 is really already known from [20] and more importantly that the results are not explained so well. The authors have made a considerable effort to fix this but I would strongly encourage them to thoroughly go over the paper so that it is clear for readers who have no previous acquaintance with [20]

---

> ### Author Response · Authors · 2023-11-22
> **Further Response to Reviewer WxNP**
>
> Dear Reviewer WxNP,
>
> Thank you for your positive response to our contribution and the increased score. We sincerely appreciate your recognition of our efforts. Regarding the proof of Theorem 2, we acknowledge its origin in [20]. We would also like to take this opportunity to commend the work done by [20], which has undoubtedly laid a solid foundation in this domain.
>
> However, it is crucial to emphasize that the faithfulness of representations ensures the existence of injective functions, an aspect not explored in [20]. Additionally, as highlighted in our contribution, we would like to stress the novelty of Theorem 1, which conflicts with certain claims in [20], instead of that of Theorem 2.
>
> In response to your valuable feedback, we have added a sentence at the beginning of Section 2 in the revised version to guide readers to the review of the Geometric Weisfeiler-Lehman test (GWL) and its relevant results in Appendix D. We believe this addition enhances the accessibility of the paper, especially for those unfamiliar with [20].
>
> We are open to addressing any further questions regarding our contributions and remain committed to improving the comprehensibility of the background and the overall quality of our paper.

---

### Official Review · Reviewer_nbPE · 2023-11-01

**Soundness:** 3 good
**Presentation:** 4 excellent
**Contribution:** 4 excellent
**Rating:** 8
**Confidence:** 3

**Summary:**

This paper studies the expressivity of geometric graph networks which are “multi-hop”, or aggregate information at each message passing step based on all nodes at most some number of steps from each node. They use the notion of equivariant moving frames to connect invariant features with equivariant features, illustrating that any equivariant activation space can be converted to an invariant activation space of equal dimension, so long as the frame is tracked (via an input-dependent group element). With this perspective, they show that k-hop invariant GNNs lose some geometric information relative to equivariant GNNs, but that the particular representations internal to equivariant GNNs matter less (in some sense) than the dimensions of the representations.

**Strengths:**

The problem studied in this paper is topical and practically meaningful, as geometric GNNs are quite widespread in application. Moreover, the insights presented in the paper are prescriptive, and give a rule of thumb for selecting hyper parameters such as the highest “L” (indexing the irrupts of O(3) or SO(3)) to include, and how many channels to use. The use of equivariant moving frames to convert to invariants is uncommon elsewhere in the literature, and its application to expressivity considerations is creative. The paper is generally clear and well-written. The authors’ theoretical claims, and even some less rigorous intuitions, are backed up with large-scale experiments on cutting edge architectures for the OpenCatalyst dataset.

**Weaknesses:**

1. My main qualm is with Corollary 3 and similar statements. In this Corollary, the authors note that given an equivariant function on a k-dimensional input representation space, it can be converted to an equivariant function on any other k-dimensional input representation space. As a result, they claim that the choice of representation to which the internal activations of a neural network transform is irrelevant (modulo its dimension). However, the conversion mapping from one equivariant vector space to a different (equal-dimensional) equivariant vector space may be arbitrarily complex or difficult to compute, and indeed, it may or may not be computable with a given architecture. In the case of rotations, one needs to recover g from a faithful representation (e.g. l=1), and then evaluate another representation at g — but computing a Wigner D matrix of high L may take many layers/parameters in the GNN. This is in the same spirit as universality results, which e.g. require increasing numbers of layers to approximate polynomials of increasing degree (see e.g. the appendix of Bogatskiy et al 2022, Lorentz Group Equivariant Neural Network for Particle Physics). In other words, there is a critical distinction between whether the **information** to compute a particular mapping is available, and whether a given architecture can actually **efficiently compute** that mapping. The authors of this work seem to focus more on the former perspective; namely, whether or not there is information loss from a certain architecture, which precludes one from ever computing a particular function (by any means). This has been a fruitful perspective for existing impossibility results on invariant GNNs — since these results roughly establish that, by only working with invariants, some neighborhoods are indistinguishable, and so **no** architecture can distinguish between them. This is a strong notion, but the converse does not hold: when the information is **not** lost, this does not imply that any architecture can actually compute the mapping. All of this is to say that, in my opinion, Corollary 3 does not sufficiently imply that the choice of equivariant representation is irrelevant up to dimension. I suspect even that the choice of representation may affect the universality of a fixed architecture family.
2. On a related note, many of the paper’s claims are informal — e.g. “there is no fundamental distinction in message-passing mechanisms arising from different values of L” in Remark 3, or “analyzing the message passing using the corresponding invariant features is a reasonable approach” on page 4. It would be very helpful and important to make these precise.
3. There are a few very related lines of work which aren’t cited, but probably should be. For example, the paper’s reasoning relies heavily on the concept of “equivariant moving frames,” which are a classical notion (dating back to Elie Cartan); yet this term does not appear in the paper, nor does a citation to e.g. Puny et al’s frame averaging paper, which is a more recent machine learning paper that harnesses the concept of frames. A small section (the start of Section 3.2 on page 6) in this paper also notes that the output symmetry of an equivariant function must be at least the input symmetry; this is a well-established fact about equivariant functions, e.g. see Smidt et al 2021. Finally, and perhaps most significantly, related ideas were discussed in ClofNet (Du et al 2022) and its follow-up LeftNet (Du et al 2023), both of which use the precise idea of moving frames and invariantization to obtain equivariants from invariants. The latter work in particular includes an expressivity result for two-hop geometric GNNs.


References:
* Frame Averaging for Invariant and Equivariant Network Design by Omri Puny, Matan Atzmon, Heli Ben-Hamu, Ishan Misra, Aditya Grover, Edward J. Smith, and Yaron Lipman
Finding symmetry breaking order parameters with Euclidean neural networks by Tess E. Smidt, Mario Geiger, and Benjamin Kurt Miller
* SE(3) Equivariant Graph Neural Networks with Complete Local Frames by Weitao Du, He Zhang, Yuanqi Du, Qi Meng, Wei Chen, Bin Shao, and Tie-Yan Liu
* A new perspective on building efficient and expressive 3D equivariant graph neural networks by Weitao Du, Yuanqi Du, Limei Wang, Dieqiao Feng, Guifeng Wang, Shuiwang Ji, Carla Gomes, and Zhi-Ming Ma


Here are a few minor typos and writing notes:
* On page 3, in the definition of geometric graphs, in the last two sentences: g inconsistently has a subscript $g_i$. Also, the last equality should presumably be an inequality (k-hop distinct if for all isomorphism, there is a node I, such that for any g, that equation does NOT hold).
* Page 3, k-hop geometry GNNs: “Given a geometric graph G=(V,E,F,X).” is not a full sentence; perhaps “Given” should have been “Consider”.
* In Lemma 1, was $V_0$ defined somewhere?
* As a very minor nitpick, the calligraphic g the authors have chosen for the group element is used almost universally to refer to an element of a Lie algebra, not of a Lie group. I would recommend sticking to regular $g$.

**Questions:**

1. Intuitively, where is the multi-hop aspect used in this work? The intuition about invariant features and moving frames seems true for 1-hop networks too. What is the takeaway regarding “multi-hop” architectures from this work, in contrast to 1-hop networks?
2. eSCN is an architecture evaluated in the experiments section. But is this a multi-hop architecture?
3. Could the authors comment further on Corollary 3, regarding point (1) from the Weaknesses section? For example, doesn’t the choice of internal representation affect the universality a given architecture (beyond just the dimension)?

---

> ### Author Response · Authors · 2023-11-16
> **Response to Reviewer nbPE (1/3)**
>
> Thank you for your thoughtful review, valuable feedback, and endorsement. In what follows, we provide point-by-point responses to your comments.
>
> ---
>
> **Q1. Questions related to a few minor typos and writing note.**
>
>
> **Reply:** We appreciate your feedback, and we have made revisions to address the noted typos and have provided the requested clarifications. Specifically,  we have added the definition of $V_0$, representing the 1-dimensional trivial representation of a given group $G$. Regarding the use of the calligraphic g, we acknowledge the convention that it is commonly associated with the Lie algebra. In this context, we have chosen it to distinguish from the regular g, which is used as a function in the proof of Lemma 2.
>
> ---
>
> **Q2. Answer the questions raised by weakness (1) and the question: Could the authors comment further on Corollary 3, regarding point (1) from the Weaknesses section? For example, doesn’t the choice of internal representation affect the universality of a given architecture (beyond just the dimension)?**
> .**
>
>
> **Reply:** We acknowledge two main aspects related to the concept of expressiveness: the capacity of features to carry information and the model's ability to extract that information. We intended to focus on the first aspect for the exact reason you mentioned in the passage on weakness (1) -- the latter is subject to the architecture of the model given, while the former is not. We want to clarify that our intention is not to directly conclude that the choice of equivariant representation is irrelevant up to dimension. We appreciate your effort in pointing out the lack of precision of some claims in the paper, especially the misleading conclusion derived from Corollary 3. We have carefully revised our statements to convey the implications of our results more precisely. Additionally, we have added a section in Appendix B to discuss the universality and potential methods. Here, we present a part of that discussion.
>
> The relationship between internal representation and universality is delicate. At first glance, the topological properties of the Lie group $G=SO(3)$ (or $G=O(3)$) prevent us from constructing a continuous function ${\bf X}\mapsto g_{\bf X}$. That is, the poor regularity of $\lambda({\bf X})$ is inevitable, making any general analytical tool that relies on regularity assumption unsuitable for studying universality. However, a potential solution to this regularity issue is to extend the function ${\bf X}\mapsto g_{\bf X}$ to a function from a suitable covering space of the spatial embedding to $G$. To put it simply, lifting the spatial embedding to the covering space makes it possible to select a continuous extension of ${\bf X}\mapsto g_{\bf X}$. Once we have that, we naturally obtain an extended version of Corollary 3 with well-behaved correspondence between two representations on the covering space.
> As a result, we suspect that under any architecture with universality, changing the internal representation while fixing the dimension does not affect the class of function the intrinsic invariant feature $\lambda({\bf X})$ can approximate. However, we cannot deny the possibility that certain choices of representation could benefit the model, allowing it to approximate the target function with fewer layers. To demonstrate this idea will require extensive work, and given the page limitation, we leave this to future works.

---

> ### Author Response · Authors · 2023-11-16
> **Response to Reviewer nbPE (2/3)**
>
> **Q3. Questions raised by weaknesses (2) and (3).**
>
>
> **Reply:** We appreciate the reviewer's feedback and concerns. In response, we have made several revisions to enhance the precision and clarity of our statements. Additionally, we have incorporated a new section (refer to Appendix A in the revised paper) to discuss additional related works, including Puny et al.'s frame averaging paper and the works by Du et al. on ClofNet and LEFTNet, and the distinctions between these existing notions and Lemma 1.
>
> Indeed, within our framework, the function ${\bf X}\mapsto g_{\bf X}$ is exactly a singleton-valued equivariant moving frame defined in [1], if we restrict the function’s domain to where the action of $G$ is free. While Puny et al.'s frame averaging paper studies set-valued equivariant moving frames, the domain of these equivariant moving frames is still restricted to guarantee the equivariance. In contrast, our defining $g_{\bf X}$ remains well-defined even in cases where the action of $G$ is not free, making it not necessarily equivariant.
>
> The scalarization proposed in Du et al.'s papers, while similar to how we define $\lambda({\bf X})\coloneqq \rho(g_{\bf X})^{-1} \cdot f({\bf X})$, differs in that the local frames used for scalarization are generated from two coordinates rather than all the coordinates in ${\bf X}$. In contrast, our $g_{\bf X}$ involves a consideration of all input coordinates in ${\bf X}$ since it is related to the orbit $G\cdot{\bf X}$.
>
> To augment the paper's comprehensiveness, we have included the empirical results of ClofNet and LEFTNet on $k$-chains. In particular, we believe the invariant design for LEFTNet belongs to the class of 2-hop invariant GNNs. So it's important to highlight that LEFTNet may still face the issues discussed in the last part of Section 3.1. Please refer to the table below for further details.
>
> Lastly, we have cited Smidt et al. (2021) to acknowledge and reference the well-established fact about the output symmetry of an equivariant function being at least the input symmetry, as noted at the beginning of Section 3.2. We believe these revisions address your concerns and provide a more precise and well-referenced discussion of related works and concepts.
>
> ---
>
> **K=2 & K=3**
>
> |        	| 1 Layer       	| 2 Layers     	| 3 Layers     	| 1 Layer       	| 2 Layers     	| 3 Layers     	| 4 Layers     	|
> |------------|-------------------|-------------------|-------------------|-------------------|-------------------|-------------------|-------------------|
> |        	| k-hop chain   	| k = 2         	| k = 2         	| k = 3         	| k = 3         	| k = 3         	| k = 3         	|
> | ClofNet	| 50.0 ± 0.0    	| 50.0 ± 0.0    	| 100.0 ± 0.0   	| 50.0 ± 0.0    	| 50.0 ± 0.0    	| 100.0 ± 0.0    	| 100.0 ± 0.0   	|
>
> ---
>
> **K=4**
> |                	| 2 Layers      	| 3 Layers      	| 4 Layers      	| 5 Layers      	| 6 Layers      	|
> |--------------------|-------------------|-------------------|-------------------|-------------------|-------------------
> | ClofNet        	| 50.0 ± 0.0    	| 100.0 ± 0.0   	| 95.0 ± 15.0   	| 95.0 ± 15.0   	| 95.0 ± 15.0   	|
> | LEFTNet        	| 50.0 ± 0.0    	| 50.0 ± 0.0    	| 50.0 ± 0.0    	| 50.0 ± 0.0    	| 50.0 ± 0.0 |100.0 ± 0.0   	|               	|
>
>
> [1] Olver, P. J. (2009). Lectures on moving frames.
>
> ---

---

> ### Author Response · Authors · 2023-11-16
> **Response to Reviewer nbPE (3/3)**
>
> ---
>
> **Q4. Intuitively, where is the multi-hop aspect used in this work? The intuition about invariant features and moving frames seems true for 1-hop networks too. What is the takeaway regarding “multi-hop” architectures from this work, in contrast to 1-hop networks?**
>
>
> **Reply:** The concept of "multi-hop" in our work specifically refers to how far each message-passing step can propagate features. We present the architecture of $k$-hop geometric GNNs in Eq (1), where the variable $k$ denotes the size of neighborhoods from which a node aggregates information. In other words, it represents the maximum distance over which a node aggregates information. Additionally, we reference ComENet as an instance of 2-hop invariant GNNs in Section 1.1 since it utilizes dihedral angles.
>
> Importantly, $k$-hop geometric GNNs are not equivalent to regular 1-hop geometric GNNs. As demonstrated in Appendix B (refer to Appendix D in the revised paper), under certain assumptions on UPD and AGG, $k$-hop invariant GNNs can distinguish any two $k$-hop distinct geometric graphs but fail to distinguish any two k-hop identical geometric graphs. This indicates that 1-hop invariant GNNs might be less powerful than their 2-hop counterparts, and this trend continues.
>
> The reason lies in the fact that the value of $k$ influences the size of local neighborhoods, consequently affecting the input of the aggregation process. Utilizing a smaller $k$ may cause $k$-hop invariant GNNs to overlook changes in geometry between small local neighborhoods, whereas using a larger k can capture these variations.
>
> However, it's noteworthy that, for any finite $k$, we can always construct two $k$-hop identical but $(k+1)$-hop distinct geometric graphs. In this scenario, $k$-hop invariant GNNs cannot differentiate them even with thousands of layers, while 1-hop equivariant GNNs, due to the faithfulness of features, can achieve this with $k+1$ layers. This finding sheds light on the advantages of learning steerable features up to type-$L\geq1$. We have substantiated this point through empirical results for ComENet and the addition of the LEFTNet model in Table 5 of Appendix F.1 in the revision.
>
> To clarify this implication, we have refined the concluding remarks. In particular, we have added the following remark: relying solely on propagating invariant features confined to specific $k$-hop neighborhoods is insufficient for enabling geometric GNNs to capture the global information of geometric graphs precisely.
>
> ---
>
> **Q5. eSCN is an architecture evaluated in the experiments section. But is this a multi-hop architecture?**
>
> **Reply:** No, the eSCN architecture is a 1-hop architecture. However, to avoid confusion, we have limited the eSCN experiments to performing the ablation studies on the steerable feature type $L$ and dimension $c$.
>
> ---
>
> We have updated our submission based on the reviewer's feedback, with the revision highlighted in blue. We are happy to address further questions on our paper. Thank you for considering our rebuttal.

---

> > ### Comment · Reviewer_nbPE · 2023-11-21
> > **Thanks for the thoughtful response**
> >
> > Many thanks to the authors for their thoughtful response. Two follow-up questions:
> >
> > * First, the authors said, "Once we have that, we naturally obtain an extended version of Corollary 3 with well-behaved correspondence between two representations on the covering space.". Where is this extended version of Corollary 3?
> > * I am also not sure what "lifting the spatial embedding to the covering space" means, and Appendix B (unless I am looking in the wrong place) did not further clarify. What is the covering space -- could the authors give a concrete example?
> >
> > Generally speaking, I stand by my endorsement of the paper, but I agree with reviewer WxNP that it is rather important to make these central claims formal theorems for a final version (including those added newly to Appendix B).

---

> ### Author Response · Authors · 2023-11-22
> **Further Response to Reviewer nbPE**
>
> Dear Reviewer nbPE,
>
> Thank you for your further feedback and invaluable comments. We highly appreciate your endorsement and have taken your suggestions into careful consideration.
>
> Regarding the extended version of Corollary 3 and the concept of lifting the spatial embedding to the covering space, we have included the relevant details in Appendix B of the revision. We aim to provide a clear understanding of the key concepts involved, and we hope the additional information addresses your questions.
> It's important to note that while we believe this extension can be proven, we have not delved into all the details and solved all technical challenges. As such, we have identified it as a potential direction for future work. We appreciate your understanding and are open to any further questions you may have.
>
> ---
>
> **Q1. First, the authors said, "Once we have that, we naturally obtain an extended version of Corollary 3 with well-behaved correspondence between two representations on the covering space.". Where is this extended version of Corollary 3? I am also not sure what "lifting the spatial embedding to the covering space" means, and Appendix B (unless I am looking in the wrong place) did not further clarify. What is the covering space -- could the authors give a concrete example?**
>
> **Reply:** We apologize for the lack of precision in our previous statement. In response to your suggestion, we have incorporated Question 1 and Question 2 into Appendix B of the revised version. Notably, Question 2 complements Corollary 3 by considering the universality aspect. Let us briefly explain it as follows.
> Consider two steerable vector spaces, $V$ and $W$, of the same dimension. Suppose we have a continuous $\mathfrak{G}$-equivariant function $f_V:\mathbb{X}_3 \to V$. According to Corollary 3, which implies the existence of a $\mathfrak{G}$-equivariant function $f_W:\mathbb{X}_3\to W$ with the same corresponding invariant functions as $f_V$. However, the continuity of $f_W$ cannot be guaranteed, and hence it may not be approximated by a given architecture. Therefore, in Question 2, we ask if there exists a continuous $\mathfrak{G}$-equivariant function $f'_W:\mathbb{X}_3\to W$ such that $f'_W$ and $f_W$ coincide on any compact set except an $\epsilon$-small measurable subset. By applying the universality of the given architectures, we can approximate $f'_W$, which has a corresponding invariant function identical to that of $f_V$ on any compact set except an $\epsilon$-small measurable subset.
> Then we delve into a detailed explanation of a potential method to address Question 2 in Appendix B. In particular, we describe how we adopt a similar approach used to obtain the equivariant moving frame in Puny et al.'s frame-averaging paper to construct the concrete covering space. Intuitively, the covering space $\widetilde{\mathbb{X}_3}$ of a space $\mathbb{X}_3$ is a topological space that locally looks like multiple copies of $\mathbb{X}_3$. We show that we have the desired continuity and equivariance of the function $g_X$ on these copies. Leveraging these properties, we should be able to construct the desired $f'_W$ for any given $f_W$. For a more in-depth understanding, please refer to the details provided in Appendix B.
>
> ---
>
> **Q2. Generally speaking, I stand by my endorsement of the paper, but I agree with reviewer WxNP that it is rather important to make these central claims formal theorems for a final version (including those added newly to Appendix B)**
>
> **Reply:** We appreciate your valuable feedback. To enhance the formality of our claims, we have introduced formal results, namely, Theorem 2 and Proposition 8 (originated from Remark 3 in the old version), to elucidate the significance of faithful representations. Additionally, we have presented formal statements, namely, Questions 1 and 2, to delve into the universality aspect and have proposed potential methods to address Question 2. It's important to note that we've used the term "Question" instead of "Theorem" or "Proposition" as we have not yet verified all the details and have decided to leave this as future work. Nevertheless, we hope these improvements in the formality of our claims and statements address your concerns.
>
> ---
>
> We hope these responses address your concerns, and we are happy to address further questions regarding our paper. Thank you for considering our rebuttal.

---

### Author Response · Authors · 2023-11-16
**General Response**

Dear reviewers and AC,

Thank you for your thoughtful reviews and valuable feedback, which have helped us significantly improve the paper. We appreciate the reviewers’ acknowledgment of the strengths of our paper. In this general response, we aim to address some common comments and summarize our major revision.

---

**Contributions and novelty**

Existing works have attempted to make theoretical and empirical comparisons to assess the expressive power and performance of invariant and equivariant GNNs. However, there is currently no theoretical comparison between $k$-hop invariant GNNs and equivariant GNNs or theoretical analysis on the performance of equivariant GNNs learning steerable features up to different type-$L$ when maintaining the feature dimension. Our work aims to bridge these gaps, and we are enthusiastic that the reviewers find our work **“topical and practically meaningful”**.

The key contributions of our work are twofold: (1) pointing out that the expressive power of $k$-hop invariant GNNs is limited compared to equivariant GNNs, (2) showing that when maintaining the feature dimension, the performance of equivariant GNNs employing steerable features up to type-$L$ may not be improved by increasing $L$.

The practical implications of our work can be summarized as (1) To attain equivalent expressiveness in invariant GNNs as in equivariant GNNs, it is essential to incorporate global features, i.e., features beyond $k$-hop neighborhoods for a fixed $k$. (2) When maintaining a constant feature dimension, achieving better performance may not be guaranteed by using higher-type steerable features in equivariant GNNs, and this choice may come with extra computational costs.

---

**Addressing common comments**

***Concern about Lemma 1:*** We have introduced an additional section in Appendix A dedicated to addressing the references raised by the reviewers related to Lemma 1. This new section delves into the distinctions between Lemma 1 and existing results, including equivariant moving frames, local frames, and explicit forms of equivariant functions. The introduction of Lemma 1 has also been rephrased, and additional explanations on  Lemma 1 have been incorporated into Remark 1.

***Concern about the importance of faithfulness:*** A new section has been included in Appendix B of the revised paper to provide a detailed discussion on the significance of faithfulness. In brief, we highlight that faithfulness ensures that features, during propagation, will not lose geometric information between local neighborhoods.

***Concern about the universality:*** We have addressed the precision of the implications of Corollary 3. We now explicitly discuss that the concept of expressiveness includes two crucial aspects: the capacity of features to carry information and the ability of a model to extract it (referred to as universality). We clarify that our primary focus lies on the former, as it remains independent of the model's architecture. We acknowledge that the exploration of the latter aspect is deferred for future research; however, we have introduced a new discussion on potential methods in Appendix B of the revised paper.


---

**Summary of the major revision**

Incorporating the comments and suggestions from all reviewers, besides fixing typos and reformating the paper, we have made the following major changes in the revision:

- Significant revisions throughout to enhance the clarity and precision of our results and their implications.

- Providing an extended discussion of additional related works, the significance of faithful representations, and limitations; see Appendix A and B for details.

- Providing more robust experiments for $k$-chains and the type $L$ ablation studies. See Tables 2, 5, 6, and 7 as well as Appendix F.1 for additional details.

-----

Thank you for considering our rebuttal.

---

### Meta-Review · Area_Chair_BjNL · 2023-12-12

**Metareview:**

The paper studies equivariant graph neural networks, which apply to geometric graphs which have both a connectivity structure and geometric features, and which are designed to be equivariant to both reordering of the vertices and euclidean transformations of the features. A variety of network structures have been proposed, including steerable features of various orders, and propagation across one or more hops on the connectivity graph.

The paper studies these structures mathematically, developing a correspondence between steerable features and invariant features. It uses this relationship to study the information content of higher order steerable features, arguing that increasing the order may not improve performance. It also develops a separation between multi-hop invariant and equivariant GNNs, arguing that equivariant networks carry more information about local graph structure.

The initial review produced a mixed evaluation of the paper, with reviewers praising the paper’s effort to clarify the advantages of various geometric GNN structures, and especially the result that higher dimensional steerable features may not carry additional information. At the same time, reviewers raised questions about the paper's main messages, and distinctions in the technical analysis vis. reference [20]. The author response addressed many of these issues, clarifying both the contributions and exposition in the submission, and highlighting the role of faithfulness in the paper's analysis.

**Justification For Why Not Higher Score:**

The paper has sufficient technical novelty, while building off of the technical foundation of reference [20] (for single hop GNNs). The writing style makes its results most accessible to specialists in the area.

**Justification For Why Not Lower Score:**

The paper provides theoretical results that help to clarify the benefits (or not) of geometric GNN structures, arguing that multi-hop equivariant GNNs capture local graph information that their invariant counterparts do not, arguing that higher dimensional steerable filters do not carry additional information, and developing a general relationship between invariant and equvariant / steerable architectures. These results could be valuable for clarifying aspects of geometric GNN design (what operations, and what feature dimension).

---

### Decision · Program_Chairs · 2024-01-16

Accept (poster)